# Hepatic thyroid hormone signalling modulates glucose homeostasis through the regulation of GLP-1 production via bile acid-mediated FXR antagonism

Ying Yan[1], Zhoumin Niu[1], Chao Sun[1], Peng Li[1], Siyi Shen[1], Shengnan Liu[1], Yuting Wu[1], Chuyu Yun[2], Tingying Jiao[3], Sheng Jia[4], Yuying Li[1], Zhong-Ze Fang[5], Lin Zhao[6], Jiqiu Wang [1][4], Cen Xie [1][3], Changtao Jiang [1][2], Yan Li [1][7], Xiaoyun Feng[8], Cheng Hu[9], Jingjing Jiang [1][6] ✉ & Hao Ying [1][1,10] ✉

Thyroid hormones (TH) regulate systemic glucose metabolism through incompletely understood mechanisms. Here, we show that improved glucose metabolism in hypothyroid mice after T3 treatment is accompanied with increased glucagon-like peptide-1 (GLP-1) production and insulin secretion, while co-treatment with a GLP-1 receptor antagonist attenuates the effects of T3 on insulin and glucose levels. By using mice lacking hepatic TH receptor β (TRβ) and a liver-specific TRβ-selective agonist, we demonstrate that TRβ-mediated hepatic TH signalling is required for both the regulation of GLP-1 production and the insulinotropic and glucose-lowering effects of T3. Moreover, administration of a liver-targeted TRβ-selective agonist increases GLP-1 and insulin levels and alleviates hyperglycemia in diet-induced obesity. Mechanistically, T3 suppresses *Cyp8b1* expression, resulting in increased the levels of Farnesoid X receptor (FXR)-antagonistic bile acids, thereby potentiating GLP-1 production and insulin secretion by repressing intestinal FXR signalling. T3 correlates with both plasma GLP-1 and fecal FXR-antagonistic bile acid levels in people with normal thyroid function. Thus, our study reveals a role for hepatic TH signalling in glucose homeostasis through the regulation of GLP-1 production via bile acid-mediated FXR antagonism.

Type 2 diabetes, a complex endocrine and metabolic disorder characterized by hyperglycemia arising from a deficient insulin secretion in the context of insulin resistance, has reached epidemic proportions over the past 50 years. As type 2 diabetes has a multifactorial pathogenesis, there is an urgent need for differently acting pharmacological compounds at different stages of the disease. The incretin hormone glucagon-like peptide-1 (GLP-1) is secreted from the intestinal L cells in response to the nutrient ingestion and potentiates glucose-dependent insulin secretion in healthy individuals[1,2]. As the insulinotropic capacity of GLP-1 is partially retained in diabetic humans, GLP-1-based treatments have been successfully developed. Recent studies suggest that GLP-1 also mediates the metabolic effects of metformin, gut microbiota, and bariatric surgery[3–6]. A better understanding of the modulation of GLP-1 release may underpin the development of glucose-lowering treatments.

Bile acids (BAs) are synthesized and conjugated in liver through a multistep process, stored in gallbladder, and released into duodenum upon nutrient ingestion[7]. BAs are essential endocrine molecules that

control metabolic homeostasis via nuclear and membrane-bound receptors, Farnesoid X receptor (FXR) and Takeda G protein-coupled receptor 5 (TGR5), respectively. Cholic acid (CA) and chenodeoxy-cholic acid (CDCA) are the major primary BAs in humans, whereas the major primary BAs in rodents are CA and muricholic acids (MCAs), the latter of which are CDCA derivatives. CA is less hydrophobic than CDCA, while MCAs are the most hydrophilic BAs[8]. In general, most hydrophobic BAs are considered as FXR agonists, while hydrophilic non-12α-OH TMCA, UDCA, and T(G)UDCA serve as endogenous FXR antagonists[4,9,10]. Since inhibition of FXR in intestinal L cells is able to promote GLP-1 release[11], changes in BA composition or profiles are likely to affect glucose homeostasis by controlling insulin secretion via modulating GLP-1 production.

The role of thyroid hormone (TH) in glucose metabolism is profound. Glucose tolerance and insulin secretion have been investigated in patients with different thyroid status and experimental hypo- and hyperthyroid animal models, but not yielding a consistent explanation, which is likely due to the severity or duration of thyrotoxicosis or hormone treatment. Glucose intolerance in thyrotoxicosis is usually accompanied with an increase in insulin secretion, which is often considered to be a compensatory response[12,13]. On the other hand, the glucose-lowering effect observed for TH mimetics is believed to be secondary to the beneficial effects of weight loss on insulin sensitivity[14]. Consistently, in a population-based study of euthyroid human study participants, decreasing levels of free thyroxine (T4) were associated with elevated markers of insulin resistance[15], suggesting that TH may exert beneficial effects on glucose homeostasis at least within the normal range of thyroid function. It is also worth noting that a close association between the feeding state and hepatic TH metabolism has been noticed early[16], however, the regulatory role of hepatic TH signaling in response to feeding is largely unknown.

Here, to explore the regulatory role of TH in glucose metabolism, we compared hypothyroid mice to 3,5,3′-triiodo-L-thyronine (T3)-treated hypothyroid mice with serum T3 concentrations close to those of euthyroid mice. Interestingly, we found that improved glucose metabolism in hypothyroid mice after T3 treatment was accompanied with increased GLP-1 production and insulin secretion. Our data also suggest that liver might be the major site responsible for the glucose-lowering effect and insulinotropic action of T3. Mechanistically, T3 acts to alter the BA composition by targeting CYP8B1, a sterol 12α-hydroxylase required for CA synthesis, and then the subsequently altered BA profiles, particularly the increased non-12α-OH FXR-antagonistic BAs, potentiates the GLP-1 production and insulin secretion by repressing FXR signaling.

## Results

### T3 treatment is able to improve the glucose metabolism in hypothyroid mice

To clarify the regulatory effect of TH on glucose homeostasis, we employed an animal treatment protocol as previously reported. To achieve maximal responses to TH treatment, mice were rendered hypothyroid first by methimazole (MMI) treatment and then received daily injection of T3, the active form of TH with a short half-life in serum, for five days. Twenty-four hours after last injection, these T3-treated hypothyroid mice with normal serum T3 levels (MMI + T3-5d mice) were compared to the hypothyroid mice with serum T3 values decreased by approximately 60% (MMI mice) to minimize the effects of potential confounding factors (Supplementary Fig. 1a). We found that the five days of T3 treatment markedly reduced the blood glucose levels in MMI mice (Fig. 1a). In contrast, the glucose-lowering effect of T3 treatment could not be observed in the hypothyroid mice four hours after injection of a single dose of T3 (MMI + T3-4h mice), although these mice displayed very high levels of circulating T3 (Supplementary Fig. 1a, b).

The regulatory effect of TH on glucose homeostasis was also evaluated by using intraperitoneal glucose tolerance test (GTT). We found that the blood glucose levels after overnight fasting were lower in MMI + T3-5d mice than those in MMI mice. Moreover, five days of T3 treatment resulted in an improvement of glucose tolerance in MMI mice (Fig. 1b). By contrast, a single dose of T3 treatment did not alter the fasting glucose levels and glucose tolerance in MMI mice (Supplementary Fig. 1c). Based on above results, we speculated that the change in glucose homeostasis observed after T3 treatment in hypothyroid mice is likely attributed to the chronic rather than acute effect of T3.

### T3 treatment is capable of increasing insulin and GLP-1 levels in hypothyroid mice

The effect of T3 treatment for five days on the blood glucose levels observed in MMI mice prompted us to determine the insulin levels. We found that five days of T3 treatment increased the insulin levels in MMI mice, while a single dose of T3 injection had no effect on the insulin levels in MMI mice (Fig. 1c). Based on these data, we speculated that T3 might have an insulinotropic effect, which contributed to its glucose-lowering action. We also inferred that the regulation of insulin levels by T3 treatment is also likely attributed to the chronic but not acute effect of T3. In line with these results, improved oral glucose tolerance was observed in MMI + T3-5d mice but not MMI + T3-4h mice compared to MMI mice (Fig. 1d and Supplementary Fig. 1d).

As the incretin effect of GLP-1 is greatly responsible for the insulin secretion after an oral glucose load, we determined the GLP-1 levels after the oral glucose challenge. In agreement with the improved oral glucose tolerance, elevated GLP-1 and insulin levels were observed after an oral glucose load in MMI + T3-5d mice compared to MMI mice (Fig. 1e and Supplementary Fig. 1e, f). Consistently, increased levels of GLP-1 and insulin could be observed in MMI + T3-5d mice compared to MMI mice even without oral glucose load (Fig. 1c, f). We then evaluated the T3 effect on the transcription of *Gcg* and found an increase in *Gcg* mRNA levels in the ileum of MMI + T3-5d mice compared to MMI mice (Fig. 1f). Accordingly, increased staining of GLP-1 was observed in the ileum of MMI + T3-5d mice (Fig. 1g and Supplementary Fig. 1g). In contrast, we could not observe any changes in the levels of GLP-1 and insulin, the mRNA expression of ileal *Gcg* as well as the immunostaining of ileal GLP-1 in MMI + T3-4h mice (Fig. 1c, f and Supplementary Fig. 1h). Collectively, these results suggest that increased GLP-1 and insulin production contribute to the glucose-lowering effect of T3. Similar results were observed either in female mice or when T4 was used instead of T3 for the treatment (Supplementary Fig. 1i, j). Consistently, elevated glucose levels, impaired glucose tolerance, reduced insulin levels, decreased plasma levels of GLP-1, and downregulated mRNA expression of ileal *Gcg* were observed in MMI mice compared to untreated control mice (CT mice) (Supplementary Fig. 1k–n). Notably, unlike GLP-1, we did not detect any alterations in the levels of glucose-dependent insulinotropic polypeptide (GIP), another incretin hormone, and its ileal expression in MMI mice after T3 treatment (Supplementary Fig. 1o).

### GLP-1 mediates the insulinotropic and glucose-lowering effects of T3

We also found that GLP-1 treatment could elevate insulin levels, decrease glucose levels, and improve oral glucose tolerance in MMI mice (Fig. 1h–j and Supplementary Fig. 1p). To test whether the incretin hormone GLP-1 was the predominant contributor to the observed improvement of glucose homeostasis, we employed exendin 9−39 (Ex-9), a GLP-1 receptor antagonist. We found that Ex-9 treatment had no effect on the plasma GLP-1 levels in MMI mice either with or without five days of T3 treatment (Fig. 1k). Interestingly, Ex-9 treatment abolished the insulinotropic effect of T3 in MMI mice (Fig. 1l). Accordingly, the glucose-lowering effect of T3 was also diminished by Ex-9

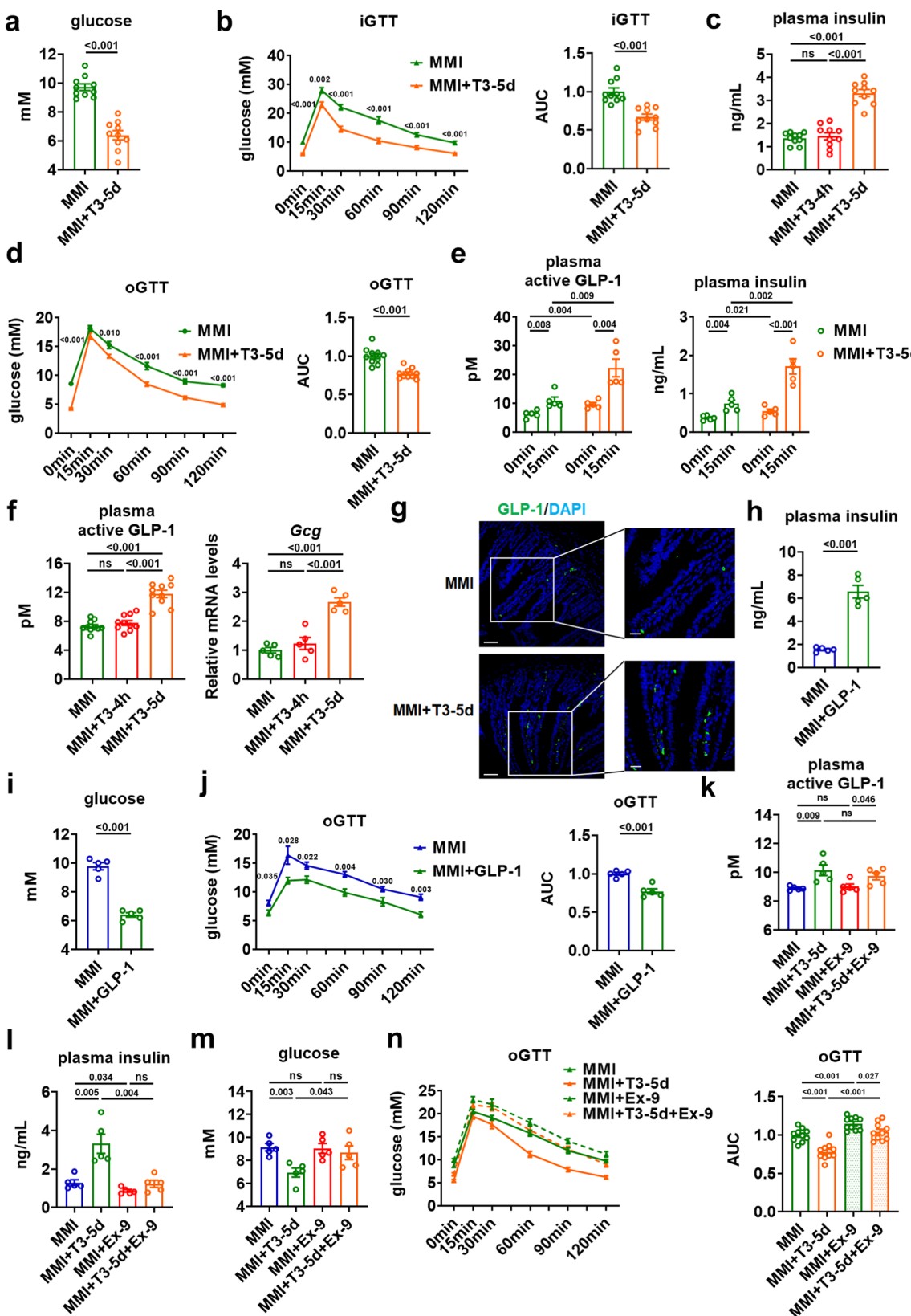

treatment in MMI mice (Fig. 1m). In line with these findings, Ex-9 administration could attenuate the effect of T3 on oral glucose tolerance in MMI mice (Fig. 1n). Thus, based on our data, we hypothesized that GLP-1 is an important contributor that mediates the insulinotropic and glucose-lowering effects of T3. Interestingly, T3 treatment had no effect on the GLP-1 levels in the culture medium of enteroendocrine

STC-1 cells, NCI-H716 cells, and mouse intestinal organoids (Supplementary Fig. 2a–c). Accordingly, the *Gcg* and *Fxr* mRNA expression were also not changed after T3 addition in these enteroendocrine cells and intestinal organoids (Supplementary Fig. 2a–c). Thus, we speculated that T3 might regulate GLP-1 production in a cell-nonautonomous manner.

**Fig. 1 | GLP-1 mediates the insulinotropic and glucose-lowering effects of T3.**
**a** Blood glucose levels in MMI and MMI + T3-5d mice (*n* = 10). **b** iGTT for MMI and
MMI + T3-5d mice (left) and the AUC (right) (*n* = 10). **c** Plasma insulin levels in MMI,
MMI + T3-4h and MMI + T3-5d mice (*n* = 10). **d** oGTT for MMI and MMI + T3-5d mice
(left) and the AUC (right) (*n* = 10). **e** Plasma active GLP-1 (left) and plasma insulin
(right) levels in fasted MMI and MMI + T3-5d mice after oral glucose challenge
(*n* = 5). **f** Plasma active GLP-1 (*n* = 10) and relative *Gcg* mRNA levels in the ileum
(*n* = 5) of MMI, MMI + T3-4h and MMI + T3-5d mice. **g** Representative IF staining of
ileal GLP-1 (green) and DAPI staining (blue) in MMI and MMI + T3-5d mice. Scale bar
(original: 50 μm; enlarged: 20 μm). **h, i** Plasma insulin (**h**) and blood glucose levels
(**i**) in MMI mice and MMI mice treated with GLP-1 for 5 days (*n* = 5). **j** oGTT for MMI
mice and MMI mice treated with GLP-1 for 5 days (left) and the AUC for oGTT (right)
(*n* = 5). **k–m** Plasma active GLP-1 (**k**), plasma insulin (**l**), and blood glucose (**m**) levels
in MMI and MMI + T3-5d mice treated with Ex-9 for 5 days (*n* = 5). **n** oGTT for MMI
and MMI + T3-5d mice treated with Ex-9 for 5 days and the AUC (right) (*n* = 10). iGTT
intraperitoneal GTT, oGTT oral GTT. Means ± SEM are shown. *P* values were cal-
culated by two-tailed unpaired Student's *t* test. ns not significant. Source data are
provided as a Source Data file.

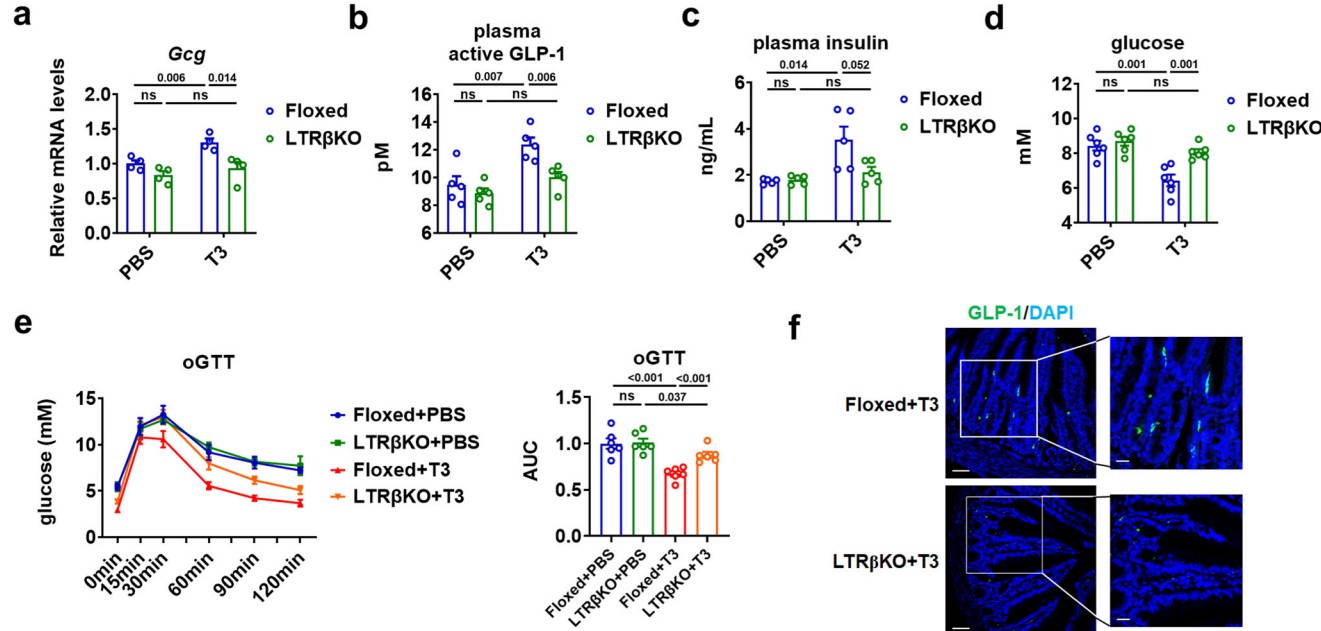

**Fig. 2 | The hepatic TRβ is required for the regulatory effect of T3 on glucose**
**metabolism. a** Relative *Gcg* mRNA levels in the ileum of Floxed and LTRβKO mice
treated with PBS or T3 for 5 days as indicated (*n* = 4). **b–d** Plasma active GLP-1
(**b**, *n* = 5), plasma insulin (**c**, *n* = 5), and blood glucose levels (**d**, *n* = 6) in Floxed and
LTRβKO mice treated with PBS or T3 for 5 days as indicated. **e** oGTT for Floxed and
LTRβKO mice treated with PBS or T3 for 5 days and the AUC for oGTT (right) (*n* = 6).
**f** Representative IF staining of ileal GLP-1 (green) and DAPI staining (blue) in Floxed
and LTRβKO mice receiving 5 days of T3 injection. oGTT, oral GTT. Means ± SEM are
shown. *P* values were calculated by two-tailed unpaired Student's *t* test. ns not
significant. Source data are provided as a Source Data file.

## Hepatic TH signaling governs the GLP-1 production and glucose homeostasis

As the glucose-lowering effect has been reported for either TRβ-
selective agonist or liver-specific TRβ-selective agonist in obese
mice[17,18], we then employed mice lacking TRβ in the liver (LTRβKO
mice) we developed recently (Supplementary Fig. 2d, e) to test whe-
ther the hepatic TH signaling was responsible for the elevation of GLP-1
and insulin levels observed in above experiments. The specificity and
efficiency of TRβ knockout was confirmed, as evident from the sig-
nificantly decreased mRNA levels of *Thrb* in the liver but not in the
adipose tissues, muscle, heart, and ileum of LTRβKO mice (Supple-
mentary Fig. 2f). LTRβKO mice were born at the expected frequency
and developed normally. There was no significant difference in body
weight and food intake between LTRβKO and TRβ flox/flox (Floxed)
mice for both gender (Supplementary Fig. 2g). We found that T3
treatment for five days stimulated the *Gcg* mRNA expression in the
ileum (Fig. 2a), increased the plasma GLP-1 and insulin levels,
decreased the blood glucose levels, as well as improved the oral glu-
cose tolerance in TRβ flox/flox (Floxed) mice (Fig. 2b–e), suggesting
that the metabolic effect of T3 could be also observed in euthyroid
state. Interestingly, the T3 effects on the *Gcg* mRNA expression, the
GLP-1 immunostaining in ileum, the plasma GLP-1 and insulin levels,
the blood glucose levels, as well as the oral glucose tolerance were all
attenuated in LTRβKO mice (Fig. 2a–f), suggesting that hepatic TRβ is

required for the observed metabolic effect of T3. Similar results could
be observed in female LTRβKO mice treated with T3 (Supplementary
Fig. 2h, i).

## Pharmacological activation of hepatic TRβ potentiates the GLP-1 production

Consistently, we found that pharmacological activation of hepatic
TRβ by daily injection of MB07811, a liver-targeted TRβ-selective
agonist, for five days could increase the plasma levels of GLP-1 and
insulin, decrease the blood glucose levels, and as well as improve
the oral glucose tolerance in MMI mice (Fig. 3a–d), which
mimicked the metabolic effect of T3 observed in MMI + T3-5d mice
(Fig. 1a, c, d, f). It is worth noting that the improved glucose
metabolism observed after MB07811 treatment was not accom-
panied by any changes in body weight (BW) and food intake
(Supplementary Fig. 3a), suggesting that the glucose-lowering
effect is independent of weight loss. As expected, the MB07811
effects on the circulating levels of GLP-1, insulin, and glucose were
all diminished in LTRβKO mice (Fig. 3e–g), not only supporting the
notion that MB07811 is a bona fide liver-targeted TRβ-selective
agonist, but also substantiate our hypothesis that hepatic TRβ-
mediated TH signaling is responsible for the GLP-1-mediated
insulinotropic and glucose-lowering effects of T3. Similar results
could be observed when TRβ[+/+], Alb-cre[+] mice were used as

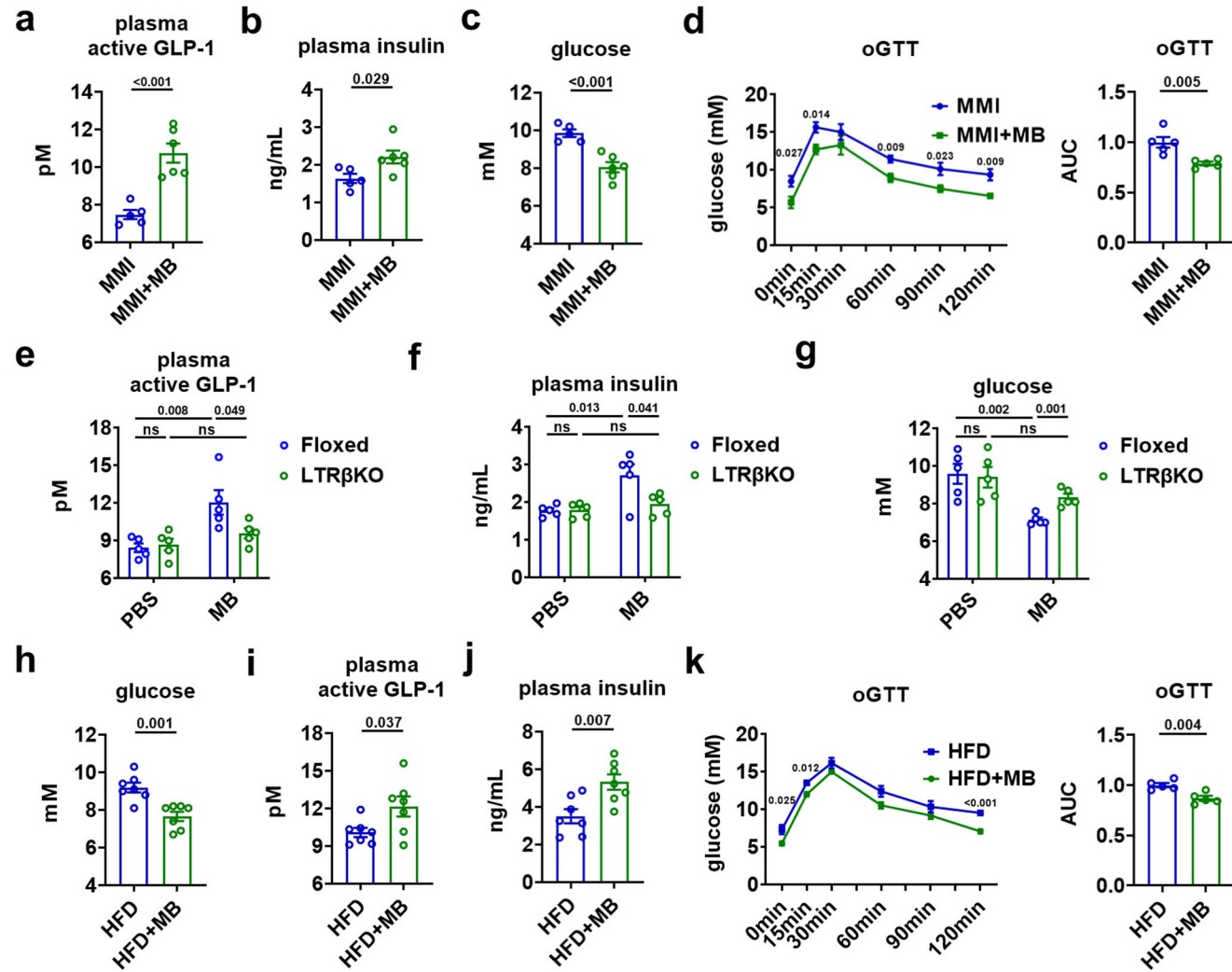

**Fig. 3 | Pharmacological activation of the hepatic TRβ potentiates the GLP-1 production. a–c** Plasma active GLP-1 (**a**), plasma insulin (**b**), and blood glucose (**c**) levels in MMI mice and MMI mice treated with MB for 5 days ($n = 5$–6). **d** oGTT for MMI mice and MMI mice treated with MB for 5 days (left) and the AUC for oGTT (right) ($n = 5$). **e–g** Plasma active GLP-1 (**e**), plasma insulin (**f**), and blood glucose (**g**) levels in Floxed and LTRβKO mice treated with PBS or MB for 5 days as indicated

($n = 5$). **h–j** Blood glucose (**h**), plasma active GLP-1 (**i**), and plasma insulin (**j**) levels in HFD mice treated with or without MB for 14 days ($n = 7$). **k** oGTT for HFD treated with or without MB for 14 days (left) and the AUC for oGTT (right) ($n = 5$). oGTT, oral GTT. Means ± SEM are shown. $P$ values were calculated by two-tailed unpaired Student's $t$ test. ns not significant. Source data are provided as a Source Data file.

controls for LTRβKO mice treated with MB07811 (Supplementary Fig. 3b). Accordingly, we did not observe any changes in the levels of GIP and its expression in the ileum of MMI mice after MB07811 treatment (Supplementary Fig. 1o).

As the insulinotropic properties of GLP-1 are retained in patients with diabetes, to test whether activation of hepatic TH signaling has potential therapeutic implications by inducing GLP-1 production, we treated high fat-diet (HFD)-fed mice with MB07811 for two weeks. As expected, we found that MB07811 treatment not only reduced the body weight, fat weight, liver weight without changing food intake, but also lowered the blood glucose levels in HFD-fed mice (Fig. 3h and Supplementary Fig. 3c–g). In agreement with our notion, the plasma levels of GLP-1 and insulin were both elevated after MB07811 treatment in HFD-fed mice (Fig. 3i, j). Moreover, the oral glucose tolerance was also improved in these HFD-fed mice after MB07811 treatment (Fig. 3k). These results indicate that the elevated plasma GLP-1 levels contribute to the glucose-lowering effect of the activation of hepatic TRβ-mediated TH signaling in HFD-fed mice. Based on above results, we propose that hepatic TRβ could serve as an exploitable therapeutic target for treating diabetes.

## TH modulates the BA composition by targeting CYP8B1 in the liver

To understand the molecular mechanism underlying the glucose-lowering effect of hepatic TH signaling, we performed RNA-seq followed by KEGG pathway analysis to identify hepatic genes or pathways regulated by T3 (Fig. 4a). We found that 3251 genes were differentially expressed in the liver of MMI + T3-5d mice compared to MMI mice (GSE184261) (Fig. 4b). KEGG pathway analysis of these differentially expressed genes (DEGs) revealed that 72 pathways were regulated by T3 treatment, including primary BA synthetic pathway (Fig. 4b). Consistent with previous reports and our knowledge[19,20], further analysis showed that many genes involved in BA synthesis and transport were regulated by T3 (Supplementary Fig. 4a). Given that GLP-1 is secreted postprandially and GLP-1-mediated incretin effect contributes to the glucose-lowering effect of T3, we decided to compare the T3-reuglated pathways with those pathways altered in response to oral intake of nutrients. By performing microarray and KEGG pathway analysis, we identified 1961 DEGs in the liver of fasted mice and fasted mice upon feeding and 42 pathways regulated by feeding (GSE184055) (Fig. 4b). Interestingly, after comparing the KEGG pathways identified above, we found 10 pathways regulated by both T3 administration and nutrition

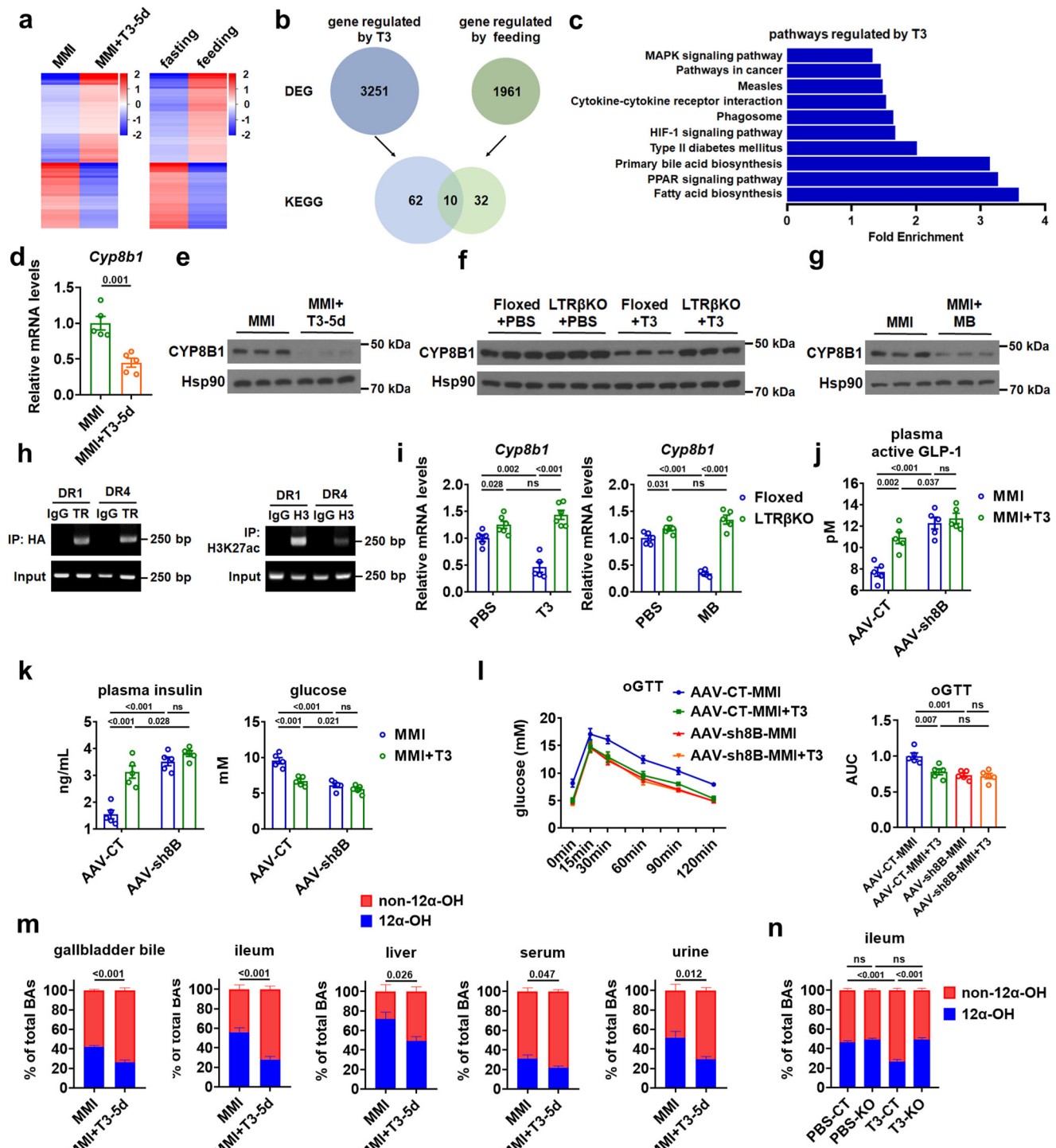

**Fig. 4 | TH shapes the BA composition by targeting CYP8B1 in the liver.**
**a** Heatmaps of all DEGs identified by RNA-seq analysis in the liver of MMI mice treated with T3 for 5 days (left, $n = 3$) and microarray analysis in the liver of mice upon feeding (right, $n = 3$). **b** KEGG analysis of these DEGs in response to T3 treatment and food ingestion, respectively. Venn diagram illustrating commonalities and differences in enriched KEGG pathways regulated by T3 and feeding. **c** KEGG pathways regulated by T3, which are also regulated by feeding, are ranked by fold enrichment. **d, e** Relative mRNA levels (**d**, $n = 5$) and western blot analysis (**e**, $n = 3$) of CYP8B1 in the liver of MMI and MMI + T3-5d mice. **f** Western blot analysis of CYP8B1 in the liver of Floxed and LTRβKO mice treated with PBS or T3 for 5 days ($n = 3$). **g** Western blot analysis of CYP8B1 in the liver of MMI mice and MMI mice treated with MB for 5 days ($n = 3$). **h** ChIP-PCR analysis of HA-TRβ recruitment (left) and H3K27 acetylation (right) in the predicted super-enhancer region containing two putative TRβ binding sites (DR1 and DR4) in primary hepatocytes. **i** Relative mRNA levels of *Cyp8b1* in the liver of Floxed and LTRβKO mice treated with T3 (left) or MB (right) for 5 days ($n = 5–6$). **j, k** Plasma active GLP-1 (**j**), plasma insulin and blood glucose levels (**k**) in MMI and MMI + T3-5d mice administered with AAV-CT or AAV-sh*Cyp8b1* (AAV-sh8B) ($n = 5$). **l** oGTT for MMI and MMI + T3-5d mice administered with AAV-CT or AAV-sh8B (left) and the AUC for oGTT (right) ($n = 5$). **m** Relative levels of 12α-OH (blue) and non-12α-OH (red) BAs in the gallbladder bile, ileum, liver, serum, and urine of MMI mice and MMI + T3-5d mice ($n = 5$). **n** Relative levels of 12α-OH (blue) and non-12α-OH (red) BAs in the ileum of Floxed and LTRβKO mice treated with PBS or T3 for 5 days ($n = 5$). oGTT oral GTT. Means ± SEM are shown. *P* values were calculated by two-tailed unpaired Student's *t* test. ns not significant. Source data are provided as a Source Data file.

ingestion, which also include the primary BA synthetic pathway (Fig. 4c). We further analyzed the overlapped gene sets in these 10 pathways and identified 22 genes (Supplementary Fig. 4b), including *Cyp8b1* which belongs to the primary BA synthetic pathway. As CYP8B1 has been implicated in the regulation of GLP-1 secretion and glucose homeostasis[21]. We then speculated that CYP8B1 might be involved in the regulation of glucose metabolism by T3.

The downregulation of hepatic *Cyp8b1* mRNA expression in MMI mice after T3 treatment was further confirmed by qPCR analysis (Fig. 4d). Accordingly, T3 treatment could also reduce the protein levels of CYP8B1 in the liver (Fig. 4e and Supplementary Fig. 4c). Notably, the regulation of CPY8B1 expression could be observed only after daily T3 injection for five days but not four hours after one T3 injection (Supplementary Fig. 4d). As expected, T3 failed to repress the CYP8B1 expression in the liver of LTRβKO mice (Fig. 4f, i and Supplementary Fig. 4e), while pharmacological activation of hepatic TRβ by MB07811 could reduce the CYP8B1 protein levels in MMI mice (Fig. 4g and Supplementary Fig. 4f). These results further support the notion that CYP8B1 is regulated by T3, which requires hepatic TRβ. As the previous study failed to define known thyroid hormone response elements (TREs) for *Cyp8b1*[22], whether it is transcriptionally by TRβ has been a longstanding mystery. Our knowledge of the transcriptional regulation mediated by T3 and its receptor TR has been greatly expanded recently by taking advantage of the genome-wide ChIP-seq analysis and expression analysis after acute and chronic TH treatment as well as during TH state transition[23–28]. We analyzed the ChIP-seq data reported recently[26] and identified a super-enhancer encompassing the mouse *Cyp8b1* (Supplementary Fig. 4g). We then performed ChIP analysis and confirmed the recruitment of TRβ to the super-enhancer region containing two putative TRβ binding sites (Fig. 4h and Supplementary Fig. 4g)[26]. These results suggest that TRβ might transcriptionally controls the *Cyp8b1* expression through a super-enhancer-mediated mechanism. Consistently, we found that the either T3 or MB07811 effect on the *Cyp8b1* mRNA expression could be attenuated in the liver of LTRβKO mice (Fig. 4i). To test whether hepatic CYP8B1 plays a primary role in the regulation of glucose metabolism by T3, we performed a CYP8B1-knockdown experiment using adeno-associated virus (AAV) expressing shRNA specific for *Cyp8b1*. We found that AAV-mediated knockdown of hepatic CYP8B1 could block the T3 effects on the GLP-1 production, insulin and glucose levels, and oral glucose tolerance in either hypothyroid or euthyroid mice (Fig. 4j–l and Supplementary Fig. 4h, i), suggesting that CYP8B1 might be the primary regulator that mediates the metabolic effect of T3 observed in this study.

As CYP8B1 acts to catalyze 12α-hydroxylation and is required for CA synthesis, it serves as a master regulator for the BA composition. To test whether the regulation of CYP8B1 by T3 would have any effect on BA composition, we performed UFLC-Triple-time of flight/MS-based metabolite profiling to quantitate the BA levels and found that the percentage of 12α-hydroxylated (12α-OH) BAs was reduced, while the proportion of non-12α-hydroxylated (non-12α-OH) BAs was increased in the gallbladder bile, ileum, liver, serum, and urine after five days of T3 treatment in MMI mice (Fig. 4m and Supplementary Data 1), suggesting that a decrease in CYP8B1 expression did translate into a decrease in 12α-OH to non-12α-OH BA ratio. Accordingly, the T3 effect on the percentage of either 12α-OH BAs or non-12α-OH BAs was diminished in the ileum of LTRβKO mice (Fig. 4n and Supplementary Data 2). Consistently, the mRNA and protein levels of hepatic CYP8B1 were elevated, while the percentage of 12α-OH BAs was increased and the proportion of non-12α-OH BAs was reduced in the gallbladder bile, ileum, liver, serum, and urine in MMI mice compared to untreated control mice (CT mice) (Supplementary Fig. 4j–l and Supplementary Data 1). In agreement with our notion, the percentage of either 12α-OH or non-12α-OH BAs was not altered four hours after one T3 injection (Supplementary Fig. 4m and Supplementary Data 1).

## TH potentiates the production of non-12α-OH FXR-antagonistic BAs

Recent evidence suggests that a couple of non-12α-OH BAs, including T(α/β)MCA, (T/G)UDCA, and (T)HDCA, can exert FXR-antagonistic effects[4,9,10,29], while intestinal FXR inhibition is able to stimulate GLP-1 production and secretion in L cell[11]. We then tested whether the levels of FXR-antagonistic BAs were altered after T3 treatment. In agreement with the role of CYP8B1 in the regulation of BA composition, we found that the percentage of TβMCA, the most abundant non-12α-OH BA (MCA), was increased significantly in the gallbladder bile of MMI + T3-5d mice compared to MMI mice (Fig. 5a and Supplementary Data 1). Importantly, a significant increase in the percentage of TβMCA was observed in the ileum of MMI + T3-5d mice compared to MMI mice (Fig. 5b and Supplementary Data 1). Increased percentage of TβMCA could also be observed in the feces of MMI + T3-5d mice compared to MMI mice (Fig. 5c and Supplementary Data 1). Consistently, the T3 effect on the percentage of TβMCA was abolished in the ileum of LTRβKO mice (Fig. 5d, Supplementary Fig. 5a and Supplementary Data 2). Given that TβMCA can promote the GLP-1 production as a natural FXR antagonist[30], we speculated that TβMCA is a major contributor to the T3 effect on GLP-1 and insulin production and glucose homeostasis.

After combining the amounts measured for the known FXR-antagonistic BAs, we also showed that the percentage of T(α/β)MCA, (T/G)UDCA, and (T)HDCA in the gallbladder bile was significantly increased in MMI + T3-5d mice compared to MMI mice (Fig. 5e). Increases in the percentage of T(α/β)MCA, (T/G)UDCA, and (T)HDCA were also observed in either the ileum or the feces of MMI + T3-5d mice compared to MMI mice (Fig. 5f, g). In line with above data, the T3 action on the percentage of FXR-antagonistic BAs was diminished in the ileum of LTRβKO mice (Fig. 5h). These results indicate that the increased amount of non-12α-OH FXR-antagonistic BAs consequent to the decrease in CYP8B1 expression after T3 treatment potentiates GLP-1 production by deactivating the intestinal FXR, thereby improving glucose homeostasis via the augmentation of insulin secretion. Consistently, the percentage of TβMCA and total percentage of FXR-antagonistic BAs were both reduced in the gallbladder bile, ileum, and feces in MMI mice compared to untreated control mice (CT mice) (Supplementary Fig. 5b–e and Supplementary Data 1). Again, in agreement with our hypothesis, the percentage of TβMCA and total percentage of FXR-antagonistic BAs were not altered four hours after one T3 injection (Supplementary Fig. 5f, g).

As TGR5-mediated BA signaling also plays a role in the regulation of GLP-1 production, we analyzed the percentage of TGR5-agonistic BAs in the ileum of MMI mice before and after T3 treatment. We found that T3 treatment decreased the total percentage of BAs with TGR5 agonist activities in the ileum of MMI mice, while the percentage of DCA, which is abundant in ileum and exhibits potent TGR5 agonist activity, was not altered (Supplementary Fig. 5h). Consistently, T3 treatment had similar effect on total percentage of TGR5-agonistic BAs in the ileum of mice in the euthyroid state, which could be abolished in LTRβKO mice (Supplementary Fig. 5i). However, the ileum cAMP levels were not affected by either T3 or MB07811 treatment in MMI mice (Supplementary Fig. 5j). Accordingly, T3 had no effect on the cAMP levels in STC-1 cells, NCI-H716 cells, and mouse intestinal organoids (Supplementary Fig. 5k). As the mRNA expression of *Fxr* and *Tgr5* was not altered by either T3 or MB07811 (Supplementary Fig. 5l), the above in vitro results together with the findings that T3 could not change GLP-1 production in cultured enteroendocrine cells or intestinal organoids (Supplementary Fig. 2a–c) indicate that BAs, probably the FXR-antagonistic BAs, might contribute to the cell-nonautonomous regulation of GLP-1 production by T3. Moreover, given that we detected an increases in the levels of FXR-antagonistic BAs, but did not observe an elevation in the levels of TGR5-agonistic BAs and cAMP, we then speculate that FXR-mediated pathway rather than

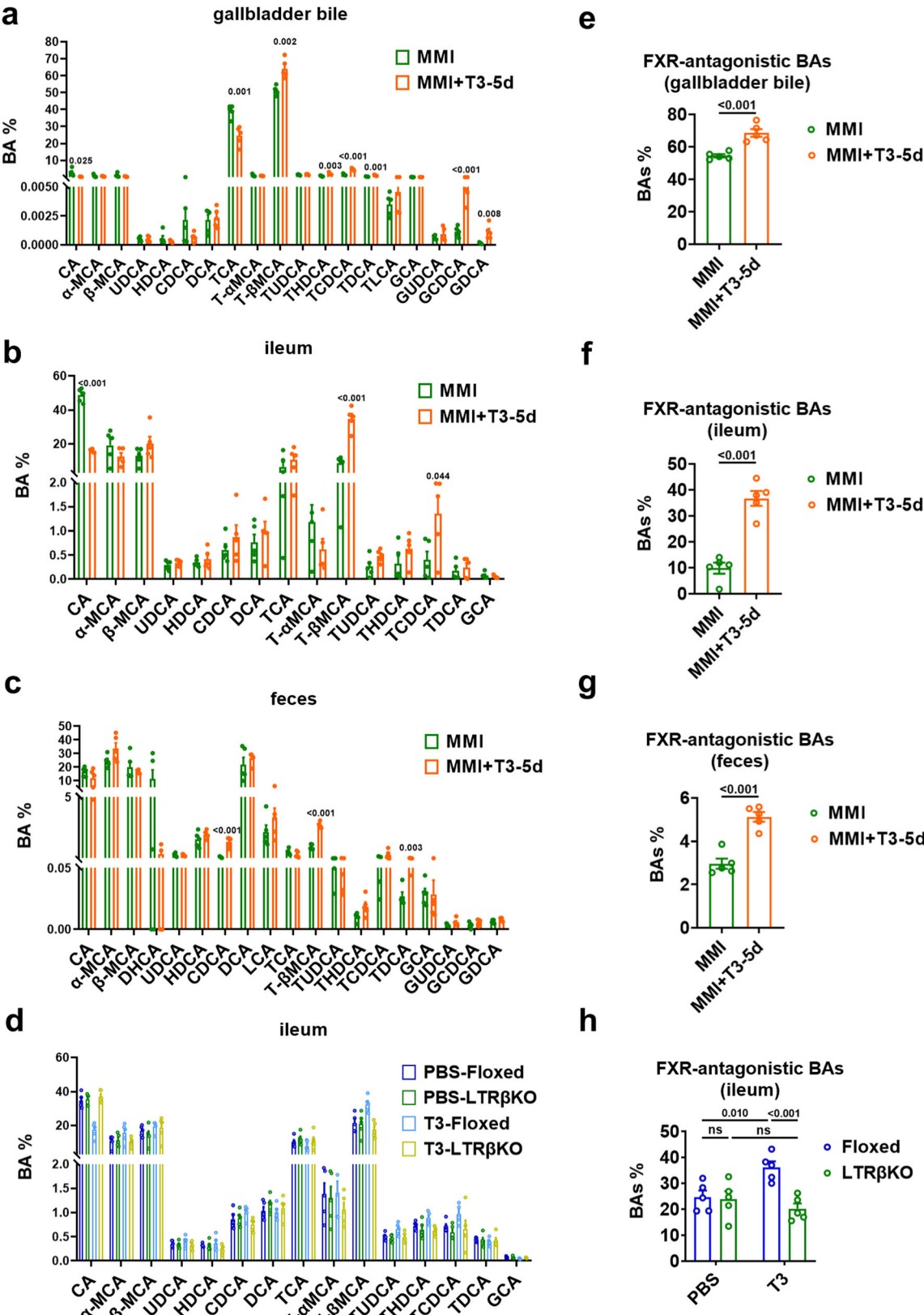

**Fig. 5 | TH potentiates the production of non-12α-OH FXR-antagonistic BAs.**
**a–c** The percentage of individual BA in the gallbladder bile (**a**), ileum (**b**) and feces (**c**) of MMI and MMI + T3-5d mice (*n* = 5). **d** The percentage of individual BA in the ileum of Floxed and LTRβKO mice treated with PBS or T3 for 5 days (*n* = 5). **e–g** The percentage of non-12α-OH FXR-antagonistic BAs, including T(α/β)MCA, (T/G) UDCA, and (T)HDCA in the gallbladder bile (**e**), ileum (**f**) and feces (**g**) of MMI and MMI + T3-5d mice (*n* = 5). **h** The percentage of non-12α-OH FXR-antagonistic BAs in the ileum of Floxed and LTRβKO mice treated with PBS or T3 for 5 days (*n* = 5). Means ± SEM are shown. *P* values were calculated by two-tailed unpaired Student's *t* test. ns not significant. Source data are provided as a Source Data file.

TGR5-mediated pathway was critically involved in the regulation of GLP-1 by T3 treatment in mice.

## Non-12α-OH FXR-antagonistic BAs mediates the T3 effect on glucose homeostasis

To dissect the contribution of the augmentation of non-12α-OH FXR-antagonistic BAs to the improved glucose metabolism after T3 daily injection, MMI mice were fed a diet supplemented with TβMCA, the most abundant murine FXR-antagonistic BA, for five days. We found that TβMCA administration increased the mRNA levels of *Gcg* in the ileum of MMI mice (Fig. 6a). Substantially elevated GLP-1 and insulin levels along with reduced blood glucose levels were observed in these mice after TβMCA treatment (Fig. 6b). Moreover, TβMCA treatment improved the oral glucose tolerance in these mice (Fig. 6c). No significant differences in BW were observed between TβMCA-treated and control MMI mice (Supplementary Fig. 6a). These results suggest that TβMCA treatment could result in a phenotype similar to that observed in MMI mice after T3 treatment (Fig. 1a, c, d, f). Consistently, we found that administration of CA, an endogenous FXR agonist, could not only

significantly attenuate the T3-induced elevation of GLP-1 and insulin levels but also attenuate the glucose-lowering effect of T3 in MMI mice (Fig. 6d and Supplementary Fig. 6b).

In line with previous reports[11] and above in vivo findings, we found that TβMCA treatment could inhibit the mRNA expression of *Shp*, an FXR downstream target, and increase the mRNA levels of *Gcg* and GLP-1 secretion in the STC-1 cells, NCI-H716 cells, and mouse intestinal organoids (Fig. 6e and Supplementary Fig. 6c–e). We also noticed that TβMCA treatment could increase the immunostaining of GLP-1 in these mouse intestinal organoids without affecting their gross morphology (Fig. 6f, g). Importantly, we found that addition of FXR-antagonistic BAs, including TβMCA, HDCA or UDCA, could antagonize the effect of CDCA, a potent FXR agonist, on the mRNA expression of *Shp* and GLP-1 secretion in NCI-H716 cells (Fig. 6h and Supplementary Fig. 6f). In agreement with the notion that GLP-1 promotes L-cells differentiation and development via a paracrine mechanism, we found that TβMCA treatment could elevate the mRNA expression of differentiation markers of L-cells, such as *Ngn3*, *Nd1*, and *Arx* (Supplementary Fig. 6g). Unlike the

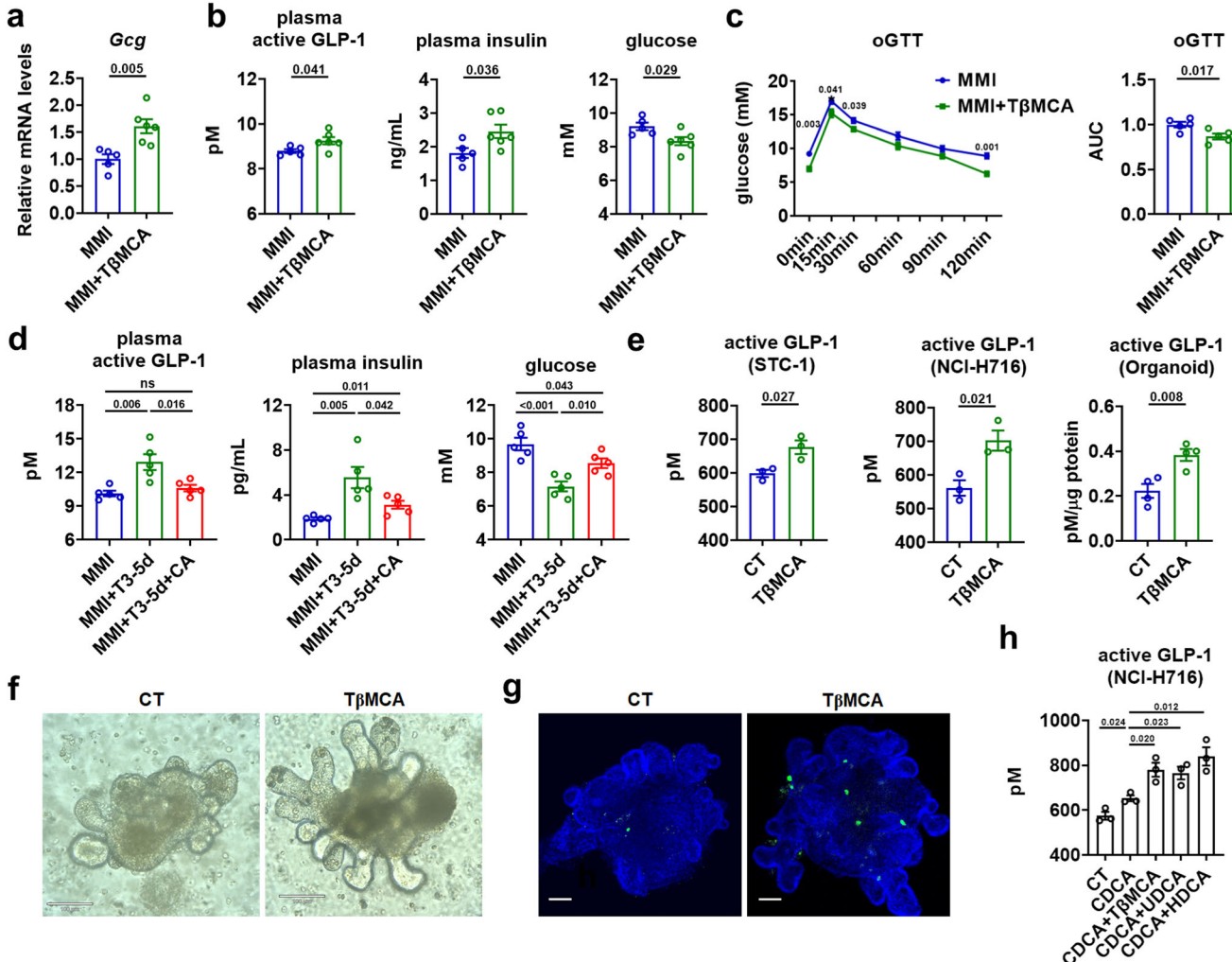

**Fig. 6 | Non-12α-OH FXR-antagonistic BAs mediates the T3 effect on glucose homeostasis. a–c** Relative mRNA levels of ileal *Gcg* (**a**, *n* = 5–6), plasma active GLP-1, plasma insulin and blood glucose levels (**b**, *n* = 5–6), oGTT (left) and the AUC for oGTT (**c**, *n* = 5) in MMI mice with or without 5 days of TβMCA treatment as indicated. **d** Plasma active GLP-1, plasma insulin and blood glucose levels in MMI mice treated with T3 and CA for 5 days as indicated (*n* = 5). **e** GLP-1 concentration in the supernatants of STC-1 cells (*n* = 3), NCI-H716 cells (*n* = 3) and mouse intestinal organoids (*n* = 4) were measured after TβMCA treatment. **f** Images of CT and

TβMCA-treated organoids. Scale bar: 100 μm. **g** Representative IF staining of GLP-1 (green) and DAPI (blue) staining of mouse intestinal organoids treated with or without TβMCA. Scale bar: 70 μm. **h** GLP-1 concentration in the supernatants of NCI-H716 cells treated with CT, or CDCA alone or together with other FXR-antagonistic BAs as indicated (*n* = 3 biologically independent experiments). oGTT oral GTT. Means ± SEM are shown. *P* values were calculated by two-tailed unpaired Student's *t* test. ns not significant. Source data are provided as a Source Data file.

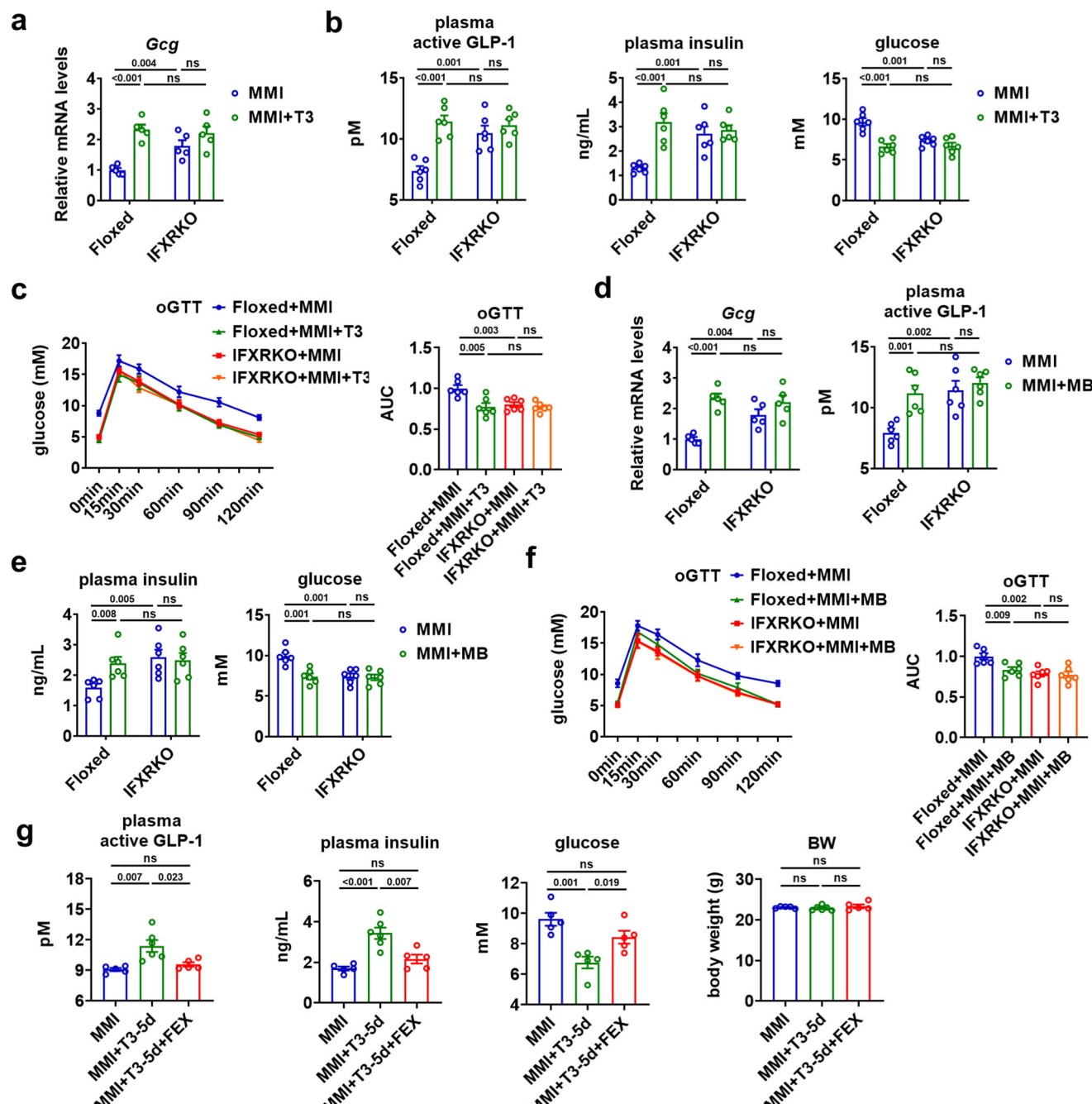

**Fig. 7 | Intestinal FXR mediates the hepatic T3 effect on the GLP-1 production and glucose homeostasis. a** Relative mRNA levels of *Gcg* in the ileum of Floxed and IFXRKO mice treated with MMI or both MMI and 5 days of T3 (*n* = 5). **b** Plasma active GLP-1, plasma insulin, and blood glucose levels in Floxed and IFXRKO mice treated with MMI or both MMI and 5 days of T3 (*n* = 6). **c** oGTT for Floxed and IFXRKO mice treated with MMI or MMI and 5 days of T3 (left) and the AUC for oGTT (right) (*n* = 6). **d** Relative mRNA levels of *Gcg* in the ileum and plasma active GLP-1 levels of Floxed and IFXRKO mice treated with MMI or MMI and 5 days of MB (*n* = 5). **e** Plasma insulin, and blood glucose levels in Floxed and IFXRKO mice treated with MMI or MMI and 5 days of MB (*n* = 6). **f** oGTT for Floxed and IFXRKO mice treated with MMI or MMI and 5 days of MB (left) and the corresponding AUC for oGTT (right) (*n* = 6). **g** Plasma active GLP-1, plasma insulin, blood glucose levels and BW of MMI mice treated with T3 and FEX for 5 days as indicated (*n* = 5–6). Means ± SEM are shown. *P* values were calculated by two-tailed unpaired Student's *t* test. ns, not significant. Source data are provided as a Source Data file.

potent TGR5 agonists, DCA and LCA, we could not detect any effects of TβMCA on cAMP levels in the STC-1 cells, NCI-H716 cells, and mouse intestinal organoids, suggesting that the increased ileal TβMCA observed in the mice after T3 treatment might not act via TGR5-mediated pathway. Thus, we hypothesized that the elevated non-12α-OH FXR-antagonistic BA levels contribute to the glucose-lowering effect of T3 through the regulation of GLP-1 and insulin production via inhibiting intestinal FXR signaling.

## Intestinal FXR mediates the hepatic T3 action on GLP-1 and glucose homeostasis

To test whether intestinal FXR signaling was essential for the T3 action observed in this study, intestine-specific FXR-null mice (IFXRKO) were employed[31]. We found that the T3 effects on the mRNA expression of ileal *Gcg*, GLP-1, insulin, and glucose levels, and oral glucose tolerance were all abolished in the MMI-treated IFXRKO mice (Fig. 7a–c and Supplementary Fig. 7a). Similar results were observed when MB07811

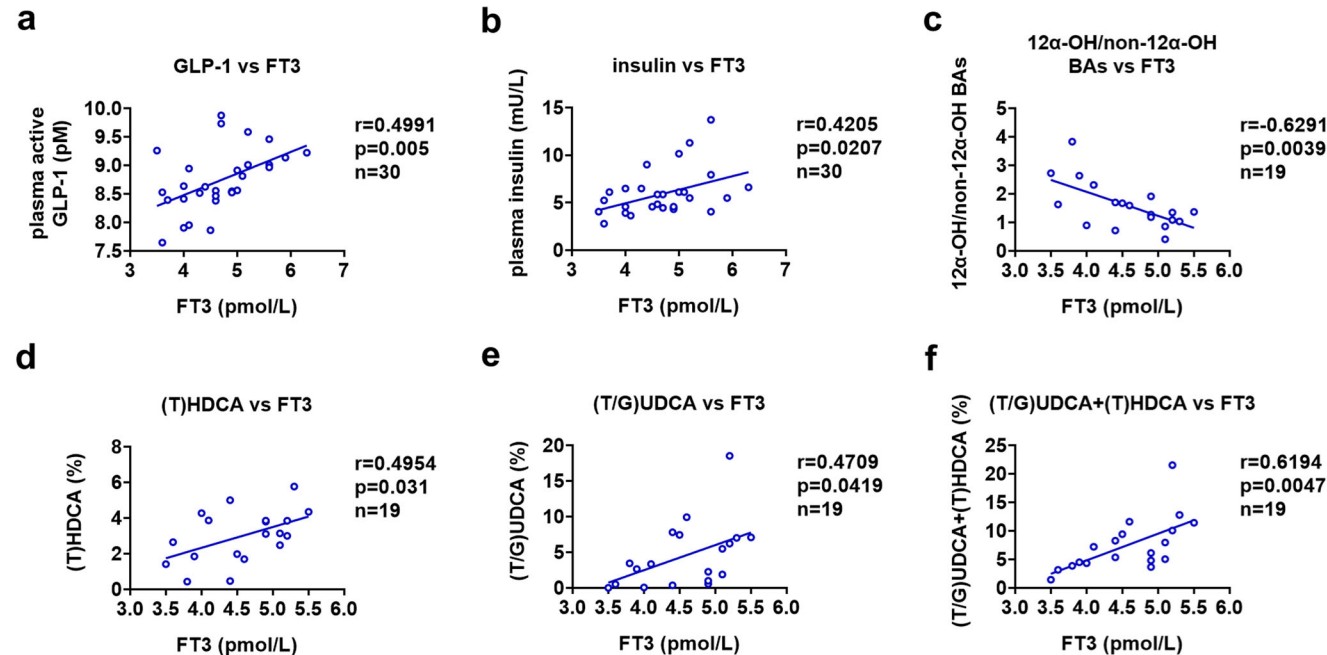

**Fig. 8 | T3 levels are associated with GLP-1 and non-12α-OH FXR-antagonistic BA levels in humans. a**, **b** Correlation between free T3 levels and plasma active GLP-1 levels (**a**) or plasma insulin levels (**b**) in a cohort of euthyroid human study participants (*n* = 30). **c**–**f** Association analysis of free T3 levels and fecal BA levels *in* another cohort of euthyroid human study participants (*n* = 19). Correlation of free T3 levels with 12α-OH/non-12α-OH ratios (**c**), (T)HDCA levels (**d**), (T/G)UDCA levels (**e**), and the levels of both (T)HDCA and (T/G)UDCA (**f**) in the feces of these human study participants. Pearson correlation and two-tailed *t*-test was performed for correlation analysis. Source data are provided as a Source Data file.

was used for treatment in IFXRKO mice in the hypothyroid state (Fig. 7d–f and Supplementary Fig. 7a). We also found that the T3 effects on the mRNA expression of ileal *Gcg*, GLP-1, insulin, and glucose levels, and oral glucose tolerance were all greatly attenuated in euthyroid IFXRKO mice (Supplementary Fig. 7b–d). Similar data were obtained when MB07811 was used for treatment in IFXRKO mice in the euthyroid state (Supplementary Fig. 7b, e, f). These results indicate that intestinal FXR is indispensable for the beneficial effect of either T3 or MB07811 on GLP-1 production and glucose homeostasis. In addition, consistent with our current knowledge of intestinal FXR signaling and our hypothesis, the GLP-1 production was increased, while the glucose metabolism was improved in MMI-treated IFXRKO mice as compared to MMI-treated Floxed mice (Fig. 7a–f). Notably, in line with the previous findings that intestinal FXR was not required for maintaining the normal glucose homeostasis[32], the glucose metabolism was normal in IFXRKO mice in euthyroid state, suggesting that compensatory mechanisms may exist (Supplementary Fig. 7c–f). Furthermore, fexaramine (FEX), a gut-restricted FXR agonist[33], was administered daily via oral gavage for five days to further substantiate our hypothesis. We found that oral administration of FEX could lower the levels of GLP-1 and insulin and increase the blood glucose levels without changing BW in MMI mice receiving T3 treatment (Fig. 7g). Collectively, these results indicate that the inhibition of intestinal FXR signaling by non-12α-OH FXR-antagonistic BAs is critically involved in the regulation of GLP-1 production, insulin secretion, and glucose homeostasis by T3.

### T3 levels are associated with GLP-1 and FXR-antagonistic BA levels in humans

To assess the physiological relevance of our above findings in this study, we analyzed the relationship between the levels of GLP-1 and T3 in human study participants with normal thyroid function. In a cohort of 30 euthyroid human study participants, a negative correlation between free T3 levels and total cholesterol levels was observed (Supplementary Fig. 8a). Interestingly, in agreement with our results

obtained in mice, a positive correlation between the plasma GLP-1 levels and the serum free T3 levels were observed in these human human study participants (Fig. 8a). We also found a positive correlation with the plasma insulin levels and serum free T3 levels in this cohort (Fig. 8b). These results indicate that, at least within the normal range of circulating T3 levels, T3 might also control the release of GLP-1 in humans, thereby modulating glucose homeostasis.

We also investigated the relationship between the BA composition and T3 levels in another cohort of euthyroid human study participants. Although significant correlation between free T3 levels and cholesterol levels was not observed in this small cohort (Supplementary Fig. 8b), in line with the data obtained in mice, we found a negative correlation between the T3 levels with the ratios of 12α-OH BAs to non-12α-OH BAs (Fig. 8c and Supplementary Data 3). These data suggest that the serum T3 levels might determine the BA composition at least in euthyroid human study participants. As humans do not produce TβMCA, we then examined the relationship between the levels of fecal non-12α-OH FXR-antagonistic BA species (HDCA and UDCA-related species) and the levels of serum T3 in the same cohort. Intriguingly, consistent with our proposed hypothesis, positive correlations between the levels of T3 and the percentages of either HDCA or UDCA species were observed in this cohort of euthyroid human study participants (Fig. 8d–f). Notably, a negative correlation was observed between free T3 levels and the percentages of total TGR5-agonistic BAs (Supplementary Fig. 8c), meanwhile, positive correlation was also not observed between the free T3 levels and the percentages of potent TGR5-agonistic BAs (LCA and DCA) (Supplementary Fig. 8d), further indicating that TGR5 signaling might not play a considerable role here. Collectively, our data suggest that T3 might also be able to increase the production of non-12α-OH FXR-antagonistic BA species in human likely consequent to the increased synthesis of non-12α-OH BAs, thereby potentiating the GLP-1 and insulin production by suppressing the intestinal FXR signaling (Fig. 9). Thus, we speculated that the findings and the associated mechanisms as detailed in our mouse studies could likely translate to humans.

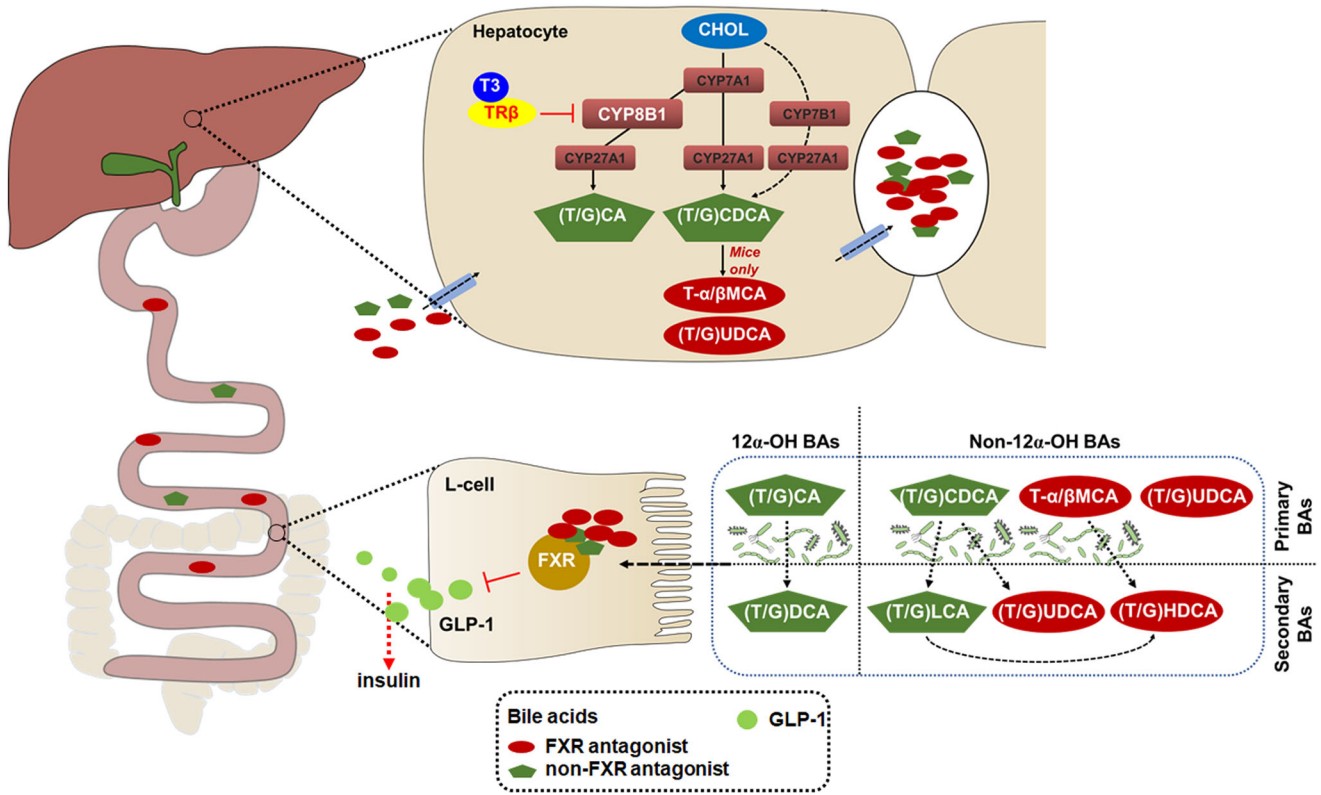

**Fig. 9 | Schematic diagram of the working model.** The role of hepatic TH signaling in the regulation of glucose homeostasis. In hepatocytes, activation of TRβ by T3 shapes the BA composition and increases the levels of non-12α-OH FXR-antagonistic BAs, which are shown in red, through suppressing the CYP8B1 expression. In L-cells, inactivation of FXR signaling by non-12α-OH FXR-antagonistic BAs increases the GLP-1 production, thereby potentiating insulin secretion. CHOL, cholesterol.

## Discussion

Normal glucose homeostasis requires a fine-tuned regulation of insulin secretion by islet β cells. Incretin effect is essential for the postprandial glucose controls, as it is responsible for 50–70% of insulin secretion after oral glucose intake in non-diabetic subjects[1]. GLP-1 accounts for most of the incretin effect after an oral glucose load and has emerged as a viable therapeutic target for the treatment of type 2 diabetes mellitus[34]. Growing evidence suggest that the elevation of GLP-1 is also critically involved in the effects of metformin, gut microbiota, and bariatric surgery on metabolic homeostasis[3–6]. In this study, we demonstrate that GLP-1 is able to mediate the insulinotropic effect of T3, thereby endowing the T3 with the ability to modulate glucose homeostasis. Our study not only refines our understanding of the regulation of GLP-1 secretion but also provides insights into the physiology of TH.

TH has a profound effect on carbohydrate metabolism; however, the underlying mechanisms are uncertain, due to the fact that some TH actions on glucose tolerance and insulin sensitivity in different tissues may have divergent influences on blood glucose levels[35,36]. TH was early described to be diabetogenic as an antagonism of the insulin effect by TH was observed, however, this concept has been challenged. Indeed, increased insulin secretion has been noticed in thyrotoxicosis in some early studies, which was usually considered as a compensatory response to lower the elevated blood glucose levels[37,38]. Here, we provide experimental evidence to demonstrate that T3 has an insulinotropic effect by acting on the GLP-1 production. We speculate that TH is capable of controlling blood glucose levels by modulating the insulin secretion. Our study also suggests that the glucose-lowering property of TH could be further exploited for the development of therapeutic strategies to treat type 2 diabetes. We would also like to point out that hyperthyroidism can increase renal blood flow and

glomerular filtration rate[39], while kidney is thought to be a major site of GLP-1 extraction[40]. Thus, the dynamics of GLP-1 needs to be carefully investigated in hyperthyroid mouse models in the future, which will help to understand the glucose metabolism in patients with clinical hyperthyroidism.

TH has a profound effect on carbohydrate metabolism; however, the underlying mechanisms are uncertain, due to the fact that some TH actions on glucose tolerance and insulin sensitivity in different tissues may have divergent influences on blood glucose levels[35]. TH was early described to be diabetogenic as an antagonism of the insulin effect by TH was observed, however, this concept has been challenged. Indeed, increased insulin secretion has been noticed in thyrotoxicosis in some early studies, which was usually considered as a compensatory response to lower the elevated blood glucose levels[37,38]. Here, we provide experimental evidence to demonstrate that T3 has an insulinotropic effect by acting on the GLP-1 production. We speculate that TH is capable of controlling blood glucose levels by modulating the insulin secretion. Our study also suggests that the glucose-lowering property of TH could be further exploited for the development of therapeutic strategies to treat type 2 diabetes.

Notably, TH mimics have already attracted a large amount of interest as they have cholesterol-lowering and weight loss properties[14]. The regulatory role of TH in glucose homeostasis has been noticed in obese mice treated with TRβ agonists. The TH mimics-induced improvement in glucose homeostasis was usually considered to be secondary to weight loss[17]. Here, our data demonstrate that, similar to T3 (Figs. 1a–d, 7g and Supplementary Fig. 6b), MB07811, a liver-targeted TRβ agonist could enhance insulin secretion and reduce glucose levels independent of weight loss in non-obese mice (Fig. 3b–d and Supplementary Fig. 3a), suggesting that hepatic activation of T3 signaling is sufficient to promote the insulin secretion and lower the

glucose levels. Consistently, the insulinotropic and glucose-lowering effect of T3 was lost in mice lacking hepatic TRβ. Furthermore, hepatic deficiency of TRβ could abolish the beneficial effect of MB07811 observed in this study (Fig. 3e–g and Supplementary Fig. 3b). Accordingly, MB07811 treatment could increase GLP-1 production and thus improve glucose metabolism in obese mice (Fig. 3h–k). Collectively, we uncovered a link between hepatic T3 action and glucose homeostasis through GLP-1, and provided evidence to show that the promising results observed for the liver-targeted TRβ agonist were not secondary to the known beneficial effects of weight loss on insulin sensitivity. We propose that liver-targeted TRβ agonists that are capable of uncoupling beneficial effects from deleterious effects could be tested in future clinical studies for the treatment of type 2 diabetes.

The physiological importance and consequence of altered BA metabolism after TH treatment remain underexplored. Although redressing the BA composition has proved to be beneficial in mouse models, it was unlikely to translate to human pathophysiology, as changes in BA composition in mice hardly reflect similar changes in BA profiles in humans due to the differences in BA composition[41]. However, recent evidence suggests that non-12α-OH FXR-antagonistic BAs in humans exert similar metabolic effects observed for those non-12α-OH FXR-antagonistic BAs in rodents[29]. Here, we found that the percentage of TβMCA, the most abundant FXR antagonist in mice, was increased after T3 treatment. As humans do not produce TβMCA, we examined the levels of non-12α-OH FXR-antagonistic BA species in humans, and found that the levels of non-12α-OH FXR-antagonistic BAs (HDCA and UDCA-related species) were positively correlated with T3 levels in euthyroid human study participants (Fig. 8d–f). We also found that the FXR-antagonistic BAs, including TβMCA, HDCA or UDCA, antagonized the effect of CDCA, a potent FXR agonist, on the *Shp* mRNA expression and GLP-1 secretion in NCI-H716 cells (Fig. 6h and Supplementary Fig. 6f). Thus, we speculate that T3 can increase the production of non-12α-OH FXR-antagonistic BAs in human likely attributed to the same mechanisms in mice, thereby promoting GLP-1 and insulin production by suppressing intestinal FXR signaling. Our findings point to activation of hepatic T3 signaling as an exploitable opportunity for treating type 2 diabetes.

T3 levels are diet sensitive and have a positive correlation with basal metabolic rate[42]. As the hepatic T4 to T3 conversion is stimulated in response to feeding[16], the role of hepatic TH signaling in the metabolic response to nutrient ingestion has been of interest. It is not surprising that hepatic T3 signaling has a metabolic effect similar to those observed after nutrition ingestion on incretin GLP-1 through the regulation of CYP8B1 and non-12α-OH FXR-antagonistic BAs. Here, we identified a previously unrecognized role of hepatic TH action in modulating the GLP-1 and insulin secretion, which are important for the maintenance of systemic glucose homeostasis. Our data elucidate the importance of hepatic CYP8B1 downregulation (Fig. 4j–l), intestinal FXR inhibition (Fig. 7a–f), and enhanced GLP-1 production (Fig. 1h–n) on the regulation of insulin secretion and glucose metabolism by T3. Based on our data, we propose that T3 has a regulatory role in insulin-controlled glucose metabolism, and T3 exerts this function by increasing non-12α-OH FXR-antagonistic BA levels via downregulating the expression of CYP8B1, which potentiates GLP-1 production via suppressing the intestinal FXR signaling. Our discovery can be integrated into a model for the role of T3 in regulating multiple aspects of glucose metabolism.

Taken together, we discover a link from TH signaling to glucose homeostasis via GLP-1. We propose that hepatic activation of TH signaling has insulinotropic and glucose-lowering effects, as it is able to increase non-12α-OH FXR-antagonistic BA levels through shaping the BA composition by repressing hepatic CYP8B1, thereby enhancing the GLP-1 production via suppressing intestinal FXR signaling. Our study also indicates that liver-targeted TRβ agonists could be tested for the treatment of type 2 diabetes in the future.

## Methods

### Animal study

Experiments involving mice were all in accordance with institutional guidelines for the care and use of animals. Animal welfare and experimental procedures were performed in accordance with the current guide of the Ethics Committee of Shanghai Institute of Nutrition and Health, Chinese Academy of Sciences. Animal protocols were approved by the Animal Care Committee (SIBS-2019-YH-1, SINH-2020-YH-1). Male C57BL/6 J mice and transgenic mice on C57BL/6 background aged about 8–12 weeks were used. Age-matched mice or mice from same litters were randomly assigned to each group. Mice were housed under 12-h/12-h light/dark cycle, humidity 50–60% and 18–22 °C ambient temperature with free access to food (chow diet, MD17121, Mediscience Ltd, China) and water. Diet-induced obese mice were fed with 60 kcal% fat diet (D12492, Research Diets) at age 6–8 weeks for 3 months. Mice with loxP sites flanking the fifth TRβ exon (TRβ Floxed mice), developed by Shanghai Model Organism Center, Inc., were fertile and appeared indistinguishable from TRβ[+/+] mice. TRβ Floxed exhibited normal body weight, food intake, glucose, and insulin levels as compared to TRβ[+/+] mice. The Liver-selective TRβ knockout mice (LTRβKO) were generated by cross-breeding of TRβ Floxed mice with mice harboring Cre-recombinase under the control of the albumin promoter (Alb-Cre mice). Intestine-specific FXR-null mice (IFXRKO) were provided by Prof. Changtao Jiang (Peking University) and described previously[31]. All animals were maintained on a C57BL/6 background.

To explore the T3 action in vivo, mice were rendered hypothyroid (MMI mice) by addition of 0.1% methimazole (M8506, Sigma) and 1% NaClO₄ (410241, Sigma) in their drinking water for 20 days as indicated[43]. MMI mice received 5 daily T3 injection at a dose of 0.25 µg per gram BW were euthanized 24 h after last T3 injection (MMI + T3-5d mice). MMI mice received a single T3 injection at a dose of 0.25 µg per gram BW were sacrificed 4 h later (MMI + T3-4h mice). To activate hepatic TRβ, mice were injected with MB07811 at a dose of 5 mg/kg/day. For T4 treatment, MMI mice received daily T4 injection at a dose of 0.6 µg/g/day for 5 days and were then euthanized 24 h after last T4 injection. For TβMCA (20289, Cayman) treatment, mice were fed with a diet supplemented with 0.1% (w/w) TβMCA for 5 days. For CA (C1129, Sigma) treatment, mice were fed with a diet containing 0.1% (w/w) CA for 5 days. Regarding FEX (BCP15784, BioChemPartner) treatment, mice were gavaged with 100 mg/kg FEX every day for 5 days. For GLP-1 antagonist treatment, mice were injected with exendin-(9–39) amide (Ex-9) (2081, R&D Systems) at a dose of 25 nmol/kg/day or vehicle (PBS) for 5 days. For GLP-1 (T3984, TargetMol) treatment, MMI mice received daily GLP-1 injection at a dose of 100 ng/g/day for 5 days and were then euthanized 2 h after last GLP-1 injection. To downregulate CYP8B1 expression level in liver, AAV-sh*Cyp8b1*, which was provided by Prof. Cen Xie (SIMM, CAS), were administered to mice at a dose of 2 × 10¹¹ vector genomes (vg) per mouse through tail vein injection. Mice were killed 4 weeks after the virus injection. For intraperitoneal GTT and oral GTT, blood glucose levels were determined in fasted mice by using a glucose analyzer at 0, 15, 30, 60, 90, and 120 min following an intraperitoneal injection and an oral administration of glucose at a dose of 2 g/kg, respectively.

### Studies with human study participants

The study conforms to the Declaration of Helsinki and Good Clinical Practice guidelines. All patients gave a written informed consent and the human sample collection was approved by the Ethics Committee of Zhongshan Hospital, Fudan University. There was no compensation for participation. For the measurement of GLP-1, 30 human study participants with normal thyroid function were recruited (Supplementary Table 1). Plasma samples were collected from these human study participants and measured following the standard protocol of the hospital. DPP-IV inhibitor (DPP4, Merck/millipore) was added to the blood samples immediately after collection and stored at −80 °C. For

the measurement of fecal BAs, 19 euthyroid human study participants were recruited (Supplementary Table 2). Fecal samples were collected from these human study participants in the sterile fecal containers, and stored in a −80 °C freezer until BA analysis.

## Cell culture and treatment

The STC-1 (CRL-3254™, ATCC) and NCI-H716 (TCHu210, Cell Bank of Type Culture Collection of Chinese Academy of Sciences) cells were cultured in DMEM medium and RPMI 1640 medium, respectively, containing 10% FBS and 1% penicillin-streptomycin. To isolate intestinal organoid, following euthanasia of 8–10-week-old male mice, crypts were isolated from the ileum fragments by EDTA incubation and plated in Matrigel using advanced DMEM/F12 containing 100 U/mL penicillin-streptomycin, 10 mM HEPES, 2 mM GlutaMAX (35050061), supplements N2 (17502048) and B27 (17504044) from ThermoFisher, 50 ng/mL murine EGF (315-09, Peprotech), 1 mmol/L N-acetylcysteine (A9165, Sigma), 100 ng/mL murine Noggin (250-38, Peprotech), and 1000 ng/mL human R-spondin-1 (120-38-20, Peprotech). The STC-1, NCI-H716 cells and mouse intestinal organoids were treated with 100 nM T3 for 2 h and 24 h for cAMP and GLP-1 measurement, respectively, unless otherwise indicated. The cells were treated with BAs (100 μM, 2 h), including TβMCA, LCA (T2202, TargetMol) and DCA (T2965, Target-Mol), for cAMP measurement. The cells were treated with BAs (100 μM, 24 h), including CDCA, TβMCA, HDCA (T2968, TargetMol) and UDCA (T0700, TargetMol), for GLP-1 measurement and RNA isolation.

## Hormone and cAMP measurement

For the measurement of plasma GLP-1, blood samples were collected in ice-cooled BD Microtainer blood collection tubes coated with K2EDTA (36784, BD) followed by addition of 50 μM DPP-IV inhibitor (DPP4, Merck/millipore), while culture medium was collected plus 1% DPP-IV inhibitor and centrifuged for 5 min at 4 °C at $12,000 \times g$. Active GLP-1 levels were determined by using an active GLP-1 ELISA kit (EGLP-35K, Merck/millipore) according to the manufacturer's instructions. For the measurement of insulin, a mouse insulin ELISA kit (EZRMI-13K, Merck/millipore) was used. For the measurement of serum levels of total T3, a T3 (total) (Mouse/Rat) ELISA Kit (KA0925, Abnova) was used. For the measurement of GIP, a Rat/Mouse GIP (total) ELISA kit (EZRMGIP-55K, Merck/millipore) was used. For the measurement of cAMP, cAMP was extracted from frozen intestine tissue samples by lysis buffer and analyzed by a mouse cAMP Assay kit (ab138880, Abcam). Intracellular cAMP was extracted from STC-1, NCI-H716 cells and intestinal organoids, and then analyzed according to the manufacturer's instruction.

## Bile acid analysis

The tissue samples of gallbladder bile, serum, liver and ileum were harvested after mice were fasted for 6 h, while the feces were collected over a 24-h period. The BA concentration in samples were quantified by using ultra-fast liquid chromatography (UFLC)-Triple-time of flight/mass spectrometer[44,45]. The 12a-OH BAs in the tissues of mice were calculated by summing the values of BA species, including CA, DCA, TCA, TDCA, GCA, GDCA. The non-12a-OH BAs in the tissues of mice were calculated by summing the values of BA species, including αMCA, βMCA, UDCA, HDCA, CDCA, LCA, T-αMCA, T-βMCA, TUDCA, THDCA, TCDCA, TLCA, GUDCA, GCDCA, GLCA.

## qPCR, western blot, and morphological studies

RNA was isolated using TRIzol reagent (15596018, ThermoFisher) according to the instructions. Concentration of total RNA was measured by NanoDrop 2000C spectrophotometer followed by the reverse transcription using RT Reagent Kit (RR037B, Takara). Primers for RT-PCR are shown in the Supplementary Table 3. The real-time qPCR reaction was conducted with SYBR Green PCR Master Mix (368735, Roche) and the reaction was accomplished by QuantStudio 6 Flex Real-Time System. Quantification of gene expression was performed using the comparative cycle threshold (Ct) method. An average Ct value was calculated from the duplicate reactions and normalized to the expression of 18S, and the ΔΔCt value was then calculated. Proteins were extracted by using RIPA buffer supplemented with phosphatase and protease inhibitors, quantified by using Bradford method (23236, ThermoFisher). Lysates were resolved on SDS–PAGE, transferred onto nitrocellulose membranes and probed with various antibodies. Chemiluminescent signals were measured by BeyoECL Plus (P0018S, Beyotime). Antibodies were obtained from Abcam (CYP8B1, ab191910; 1:1000) and Cell Signaling Technology (Hsp90, 4874S; 1:1000). Slides were fixed with 4% PFA and permeabilized in ice cold menthol. Heat-mediated antigen retrieval with a 0.01 M citric acid (pH 6.0) was performed for 5 min in a microwave. After blocking with 5% BSA, the sections were incubated with 1:200 anti-GLP-1 (ab23468, Abcam), followed by detection with secondary antibodies Alexa Fluor 488 conjugated goat antibody to mouse IgG (A-11029, ThermoFisher; 1:1000). Slides were then washed in PBS and stained with DAPI. Images were acquired by fluorescence microscopy (Zeiss System). Immunohistochemical staining of paraffin sections was carried out by using 1:1000 anti-GLP-1 antibody (ab26278, Abcam) and detected by ECHO Revolve Microscope.

## Super enhancer analysis and chromatin immunoprecipitation assay

The previously published[44] ChIP-seq dataset data re-used in this study are available in the Gene Expression Omnibus database under accession code GSE159648. Briefly, all sequencing reads of biological replicates were aligned to the mm9 genome using Bowtie2[46]. After removing duplicate reads, peaks were called using MACS2[47]. Super enhancers were identified with a stitching distance of 12,500 bp and assigned to the associated genes by ROSE[48]. ChIP assays of primary hepatocytes were performed using an EZ Magna ChIP G kit (1–409, Merck/millipore). Cells were transfected with HA-TRβ. Immunoprecipitation was performed using an anti-HA antibody (3724S, Cell Signaling Technology; 1:200) or Anti-Histone H3 (acetyl K27) antibody (ab4729, Abcam; 1:200) with rabbit IgG as a control. PCR products were resolved by electrophoresis in a 2% Agarose-gel. Primer sequences for ChIP assay are provided in Supplementary Table 3.

## RNA-seq and microarray data analysis

RNA-seq libraries were prepared using Illumina TruSeq Stranded mRNA Library Prep (Illumina). Samples were pooled for deep sequencing by using Illumina Hiseq Xten (2 × 150) platforms at the CAS-MPG Partner Institute for Computational Biology Omics Core, Shanghai, China. For RNA-seq data analysis, reads of the samples were trimmed for adaptors and low-quality bases using Trimmomatic 0.36 software before alignment with the mouse reference genome (mm10) and the annotated transcripts using STAR 2.5.1. Reads for annotated (Ensembl) genes were counted using HtSeq. Normalization of expression levels for each gene and differential expression analysis were performed using DESeq2 1.30.1. For microarray analysis, Agilent Array platform was employed. The sample preparation and microarray hybridization were performed based on the manufacturer's standard protocols. Briefly, total RNA from each sample was amplified and transcribed into fluorescent cRNA with using the manufacturer's Agilent's Quick Amp Labeling protocol (version 5.7, Agilent Technologies). The labeled cRNAs were hybridized onto the Whole Mouse Genome Oligo Microarray (4x44K, Agilent Technologies). After having washed the slides, the arrays were scanned by the Agilent Scanner G2505C. Agilent Feature Extraction software (version 11.0.1.1) was used to analyze acquired array images. Quantile normalization and subsequent data processing were performed using the GeneSpring GX v11.5 software package (Agilent Technologies). After quantile normalization of the raw data, differentially expressed genes were identified through Fold Change filtering. Genes with a twofold change were

considered differentially expressed. Functional enrichment analysis was performed with DAVID using the pathways related to metabolism from the KEGG database annotation.

## Statistics and Reproducibility

Randomization and blinding strategy were employed whenever possible. Representative images of two independent experiments with similar results were shown in Figs. 1g, 2f, 6f, g, and Supplementary Fig. 1g, h. All remaining experiments were performed at least twice, and representative data are shown. Data are expressed as means ± SEM. GraphPad Prism 8.0.2 and two-tailed unpaired student's t test were applied. A p value of less than 0.05 was considered significant. ZEN 2.3 (blue edition) software was used for acquisition of images from confocal microscope. ImageJ v1.8.0 (National Institutes of Health) was used for Western blot densitometry analysis.

## Reporting summary

Further information on research design is available in the Nature Research Reporting Summary linked to this article.

## Data availability

Data supporting the findings of this study are available within the article and its Supplementary information files or from the corresponding author upon reasonable request. Microarray and RNA-seq data generated in this study have been deposited in the Gene Expression Omnibus under accession code GSE184055 and GSE184261, respectively. The previously published[44] ChIP-seq dataset data re-used in this study are available in the Gene Expression Omnibus database under accession code GSE159648. Mouse reference (mm10) dataset required for Cell Ranger can be downloaded at 10x Genomics. Source data are provided with this paper.

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

## Acknowledgements

This work was supported by the National Key Research and Development Program of China No. 2021YFA1100500 (H.Y.); National NSFC 91957205 (H.Y.), 82070821 (J.J.), 81970748 (L.Z.); Pujiang Talent Program from STCSM 21PJ1416100 (Y.Y.L.); Youth Innovation Promotion Association, CAS 2021261 (Y.Y.L.); NHC Key Laboratory of Food Safety Risk Assessment 2020K02 (Y.W.); Nature Science Foundation of Shanghai 202R1444400 (X.F.), and Collaborative Innovation Center of Food Safety and Quality Control in Jiangsu Province, Jiangnan University (2022-3-1) (Y.L.). The authors would like to thank Zhonghui Weng et al. from ICSTF of SINH, CAS for technical assistance.

## Author contributions

Y.Y. and H.Y. designed the experiments and analyzed the data; Y.Y. carried out most of the experiments and analyzed the data; Z.N., C.S., P.L., S.S., S.L., C.Y., T.J., S.J., Y.L., Z.F., L.Z., J.W., C.X., C.J. Y.W., and J.J. provided the technical assistance; J.W. and C.X. contributed to the interpretation of data; J.W., Y.L., X.F., C.H., Y.Y.L., and J.J. contributed to the discussion and supervised the project; Y.Y, J.J. and H.Y. wrote and revised the manuscript.

## Competing interests

The authors declare no competing interests.

## Additional information

[1]CAS Key Laboratory of Nutrition, Metabolism and Food Safety, Shanghai Institute of Nutrition and Health, University of Chinese Academy of Sciences, Chinese Academy of Sciences, and Shanghai Jiao Tong University Affiliated Sixth People's Hospital, Shanghai, China. [2]Department of Physiology and Pathophysiology, School of Basic Medical Sciences, Peking University, Beijing 100191, China. [3]State Key Laboratory of Drug Research, Shanghai Institute of Materia Medica, Chinese Academy of Sciences, Shanghai 201203, China. [4]Key Laboratory for Endocrine and Metabolic Diseases of Chinese Health Commission, Department of Endocrinology and Metabolism, Ruijin Hospital, Shanghai Jiao Tong University School of Medicine, Shanghai 200025, China. [5]Department of Toxicology and Sanitary Chemistry, School of Public Health, Tianjin Medical University, Tianjin 300070, China. [6]Department of Endocrinology and Metabolism, Zhongshan Hospital, Fudan University, Shanghai 200031, China. [7]State Key Laboratory of Food Science and Technology, School of Food Science and Technology, Jiangnan University, Wuxi 214122, China. [8]Department of Endocrinology and Metabolism, Shanghai General Hospital, School of medicine, Shanghai Jiaotong University, Shanghai 200080, China. [9]Shanghai Diabetes Institute, Shanghai Key Laboratory of Diabetes Mellitus, Shanghai Clinical Centre for Diabetes, Shanghai Jiao Tong University Affiliated Sixth People's Hospital, Shanghai 200233, China. [10]Key Laboratory of Food Safety Risk Assessment, Ministry of Health, Beijing 100021, China. ✉e-mail: jiang.jingjing@zs-hospital.sh.cn; yinghao@sibs.ac.cn

