## [Peer Review File · Nature Communications]

Title: Hepatic thyroid hormone signalling modulates glucose homeostasis through the regulation of GLP-1 production via bile acid-mediated FXR antagonismREVIEWER COMMENTS

Reviewer #1 (Remarks to the Author):

This manuscript proposes a four-organ interaction (a thyroid-liver-ileum-pancreas axis) whereby T3 signaling increases non-12 α -hydroxylated BA production in the liver. This results in increased GLP-1 synthesis and secretion from the ileum due to local reduction in FXR signaling by those non-hydroxylated BAs. Ultimately, this increase in GLP-1 is responsible of the T3-induced improvement in glucose tolerance.

This is an interesting premise that links different cellular mechanisms playing key roles in the control of glucose homeostasis that are susceptible of drug targeting. This undeniably adds interest to the work. However, there also significant limitations, which are explained below:

Most experiments are performed in the context of T3 replacement in pharmacologically (MMI)-induced hypothyroidism and conclusions are drawn based on simple pair-wise comparisons between minimally powered groups. Lack of back-to-back comparison with euthyroid controls limits the potential relevance of the findings a pathophysiological state. There is also some lack of lack of systematic comparability across otherwise similar experiments (e.g. groups or parameters are selectively shown).

Although increased GLP-1 levels are at the center of their hypothesis, conclusions are inferred out of small changes in ambient levels of GLP-1 or Gcg expression. Considering the low number of replicates and the small size of the effect, this is a significant weakness that challenges reproducibility. Sometimes conclusions are derived from overinterpretation of indirect assessments. For example, in Fig1n if the T3-improvement in tolerance were due to solely to increased GLP-1, then the effect of the blockade of GLP1R with Ex9 between T3 (difference between green vs yellow) the untreated groups (difference between blue vs red) should be statistically significant. Given that additional experiments providing a more accurate determination of the GLP-1 secretion kinetics would be desirable.

The T3-induced improvement in glucose tolerance were mimicked by treatment with a liver-biased TRb agonist and were is significantly attenuated in mice lacking TRb expression in the liver (although, and again, critical Albcre+ control mice were not investigated; additional details about the generation including targeting methodology, genomic context and general phenotype of the flox mouse should provided).

Nonetheless, this is an important experiment that compellingly demonstrate the role of hepatic T3 signaling in modulating glucose metabolism. There is no question that this TRb mediates some of the benefits of T3 and MB07811 on glycemic control. However, significant unknowns arise from this experiment that question the main hypothesis. For instance, loss of hepatic TRb3 signaling in liver does not result in dramatic increases in Cyp8b1 levels compared to controls, as it would be expected in light of the effect of MMI treatment. Since changes in BA composition were not investigated either, it is hard to attribute the loss of effect of T3 in the KO to changes in BA composition. Considering this, MMI treatment of liver-specific TRb KO mice, plus minus T3, would shed light on the actual contribution of

hepatic TRb signaling on the benefits seen with the T3 replacement in the MMI treated mice. If worsening in glycemic control (note that the TRb KO exhibit normal glucose tolerance) is seen and that can be normalized with T3 treatment, then other options should be considered.

Administration of TbmCA acid to MMI-treated mice recapitulated a similar effect to that of T3 replacement on glucose tolerance and Gcg expression, GLP-1 and insulin levels. Conversely, treated with CA or the gut-bias FXR agonist Fexeramine reverted some of the effects of T3 replacement in MMI-treated mice that are consistent with enhanced GLP-1 secretion. However, due to the concern manifested above, this interaction between FXR signaling and T3 could be due to independent events acting in parallel with different contributions. T3 treatment of MMI-treated, villin-cre:FXR flox mice would be certainly more convincing. This is not unreasonable considering that other modalities of BA signaling play a meaningful role regulating L-cell differentiation (<https://doi.org/10.2337/db19-0764>)

Reviewer #2 (Remarks to the Author):

In the current study by Yan et al., hepatic activation of T3 signaling is sufficient to promote the insulin secretion and lower the glucose levels. Furthermore, the team identified that this is due to reduction in CYP8B1 and increase in bile acids that are FXR antagonists. Based on one study reported in 2015 that inhibition of FXR in L cells increase GLP1 production and secretion, the authors concluded that the T3-mediated induction in proglucagon (precursor of GLP-1) in L cells is due to intestinal inhibition of FXR. While the conclusion is exciting, there are some concerns:

1. How about the regulation of T3 on other genes involved in bile acid synthesis? What about levels of bile acids that are TGR5 activators?
2. What are the effects of T3 on carbohydrate ingestion and other incretin secretion in addition to GLP-1?
3. Validation of TGR5 activation. How about comparing to report showing that FXR can induce TGR5? How about the activation status in the intestine epithelial cells and also in L cells
4. In humans, inhibition of CYP8B1 will lead to the bile acid pool enriched in CDCA, which is the most FXR endogenous ligand---opposite to rodents.
5. How about using T4 than T3? which is more commonly used in the clinic for the treatment of hypothyroidism.
6. In fig 6, it will be interesting to have additional control groups treated with CA or FEX with or without MMI

Reviewer #3 (Remarks to the Author):

The authors demonstrate that euthyroid animals have improved glucose tolerance and increased circulating insulin and circulating and intestinal GLP-1 levels compared to hypothyroid animals. Using a

wide range of models the authors demonstrate that liver TRB is required for (the majority of) this effect. Intestinal FXR is shown to modulate the beneficial effect of T3 on GLP-1 levels, plasma insulin and plasma glucose in hypothyroid mice. The authors postulate that the link between (hepatic) thyroid hormone signaling and changes in bile acid composition, resulting in increased GLP-1 secretion, is the enzyme CYP8B1 in liver. The data is novel, interesting and clinically relevant given the development of liver specific TRB agonists for the treatment of metabolic disease.

Specific Comments:

1. Is hyperglycemia commonly found in hypothyroid mouse models. To this reviewers knowledge the answer is no. What is the reason for this here. Other examples should be brought forward.
2. The increased levels of GLP-1 described in Figure 1J are not clear.
3. Why was GLP-1 not given directly to hypothyroid animals?
4. Ideally, further experiments should be provided to demonstrate KO of the TR in the liver. ie TH mediated gene expression of classic TR target genes. The model is under-described in all aspects.
5. In Figure 2 and Figure 3 the induction of hypothyroidism leads to a rise in blood glucose in both WT and KO animals and T3 or the analog reduces it only in WT or floxed animals. Why is there an induction of BG in hypothyroidism in the absence of the TR?
6. In Figure 3 there is no data on the high fat diet mice ie food intake, body weight etc. It is impossible to determine the effects of the MB compound in this setting.
7. In Figure 4 where is the RNA-Seq data of the genes regulated by feeding. What does Cyp8b1 do with increased feeding.
8. Also, in Figure 4, T3 is known to regulate many other genes involved in bile acid transport that also regulate bile acid hydrophobicity including cyp27a1 and cyp3a11 and as well bile acid transporters. These genes are not included in the analysis shown. Interestingly, Cyp8b1 has been identified previously as a target of TH signaling: Andersson U, Yang YZ, Bjorkhem I, Einarsson C, Eggertsen G, Gafvels M. Thyroid hormone suppresses hepatic sterol 12 α -hydroxylase (CYP8B1) activity and messenger ribonucleic acid in rat liver: failure to define known thyroid hormone response elements in the gene. *Biochim Biophys Acta*. 1999;1438(2):167–174. Additionally, thyroid hormone signaling has also been shown to alter bile acid composition previously and thus regulate intestinal cholesterol absorption (Astapova et al, *JCI* 2014).
9. The explanation of the data in Figure 7 is not clear. TSH is used to determine TH action. Do the TSH levels inversely correlate with the TH levels. Additionally do the T3 levels correlate with other actions of TH on the liver ie serum cholesterol.
10. Is there a CYP8B1-KO model available or specific inhibitor that would allow you to study the effect of T3 in the absence of CYP8B1?
11. Does the attenuated change in CYP8B1 in MMI+T3 treated L-TRBKO mice result in measurable changes in intestinal bile acid composition compared to controls?

Other comments:

12. Figure 2: . Please specify if these mice were rendered hypothyroid before being treated with T3 as this isn't mentioned in the figure legends or corresponding results section

13. The other bile acid receptor TGR5 has been described to be crucial for GLP-1 induction and several other metabolic effects. Did you find any evidence for a role of TGR5 in this pathway? Why did you choose to focus only on FXR?
14. In general the paper is well written but the discussion would benefit from a thorough read-through to correct several grammatical errors.
15. Please specify where the MB07811 TRB agonist was purchased and include data or references that demonstrate its liver and TRB specificity.
16. Have you done any of these experiments in female mice? If so please show (in supplemental)
17. Please state whether your human subjects provided informed consent in accordance with the declaration of Helsinki.

**REVIEWER COMMENTS**

**Reviewer #1 (Remarks to the Author):**

This manuscript proposes a four-organ interaction (a thyroid-liver-ileum-pancreas axis) whereby T3
signaling increases non-12 α -hydroxylated BA production in the liver. This results in increased GLP-
1 synthesis and secretion from the ileum due to local reduction in FXR signaling by those non-
hydroxylated BAs. Ultimately, this increase in GLP-1 is responsible of the T3-induced improvement
in glucose tolerance.

This is an interesting premise that links different cellular mechanisms playing key roles in the
control of glucose homeostasis that are susceptible of drug targeting. This undeniably adds interest
to the work. However, there also significant limitations, which are explained below:

Most experiments are performed in the context of T3 replacement in pharmacologically (MMI)-
induced hypothyroidism and conclusions are drawn based on simple pair-wise comparisons between
minimally powered groups. Lack of back-to-back comparison with euthyroid controls limits the
potential relevance of the findings a pathophysiological state. There is also some lack of lack of
systematic comparability across otherwise similar experiments (e.g. groups or parameters are
selectively shown).

**Response to the Reviewer's Comments:** We thank the reviewer for the comments. We agree with
the reviewer that lack of comparison with euthyroid controls limits the potential pathophysiological
relevance. Previously, we did not include such a comparison for following reasons. 1) TRs are ligand
(T3)-dependent nuclear receptors. For positively regulated genes, in the absence of T3, unliganded
TRs (apoTRs) sit on the promoter of its target gene and repress transcription by recruiting
corepressors rather than sit there doing nothing. In the presence of T3, liganded TRs (holoTRs)
release corepressors and recruit coactivators, thereby stimulating the transcription. Thus, in the TR
field, to identify T3 target genes or T3-regulated pathways that have maximal T3 responsiveness,
mice were normally rendered hypothyroid before T3 treatment, while cultured cells were normally
pre-cultured in Td (TH-deficient) medium before T3 treatment. 2) The role of TH in metabolism is
profound, as TH normally coordinates metabolic pathways by targeting multiple enzymes and affect
systemic homeostasis via its regulatory action in different metabolic tissues. The severity or duration
of thyrotoxicosis or hormone treatment may yield inconsistent results. Based on available evidence,
we speculate that TH may exert beneficial effects on glucose homeostasis at least within the normal
range of thyroid function, while extremely high levels of TH may have deleterious effects on
metabolic homeostasis. Furthermore, because MMI may have unknown effects on metabolism
independent of its actions on TH synthesis in the thyroid, to minimize the effects of potential
confounding factors and focus on the regulatory action of T3, we employed MMI-treated mice with
detectable but low T3 levels as a control group and MMI-treated mice receiving T3 injections as an
experimental group.

As suggested, we provided the data showing the differences between hypothyroid MMI-treated mice
(MMI mice) and untreated euthyroid control mice (CT mice). Consistent with our hypothesis, MMI
mice exhibited elevated blood glucose levels, impaired intraperitoneal and oral glucose tolerance,
decreased insulin levels, and reduced GLP-1 expression and production as compared to CT mice
(Supplementary Fig. 1k-n). It is worth noting that, as the half-life of T3 in mouse serum is about 2

42 hours, MMI mice receiving daily injection of T3 are not an authentic mouse model for clinical
hyperthyroidism. Thus, we should be cautious when they were compared directly to euthyroid mice.
To avoid potential confusion and interpret the data correctly in a straightforward way, we decided
to show the results of comparison between MMI mice and CT mice (euthyroid controls) rather than
the results of comparison between T3-treated MMI mice (MMI+T3 mice) with CT mice in our
revised manuscript (Supplementary Fig. 1k-n, 4h, i, 5b-e). Similar comparisons could also be found
in Supporting data (Supporting Fig. 1e, f, 2i-k, 3a).

As the reviewer suggested, to substantiate our notion by increasing sample numbers in Fig. 1b, 1d, 1e,
and 1h (previous version), we repeated these experiments and provided the new data in our revised
manuscript (Fig. 1a, c, d, and the left panel of f). In addition, more control groups were included for
better systematic comparability and more detailed information for general phenotype of the mice
and experimental conditions were also provided in our revised manuscript (Supplementary Fig. 1i,
k-n, 2g-i, 4h, i, 5b-e and Supporting Fig. 1d-f, 2a, i-k).

Although increased GLP-1 levels are at the center of their hypothesis, conclusions are inferred out
of small changes in ambient levels of GLP-1 or Gcg expression. Considering the low number of
replicates and the small size of the effect, this is a significant weakness that challenges
reproducibility. Sometimes conclusions are derived from overinterpretation of indirect assessments.
For example, in Fig1n if the T3-improvement in tolerance were due to solely to increased GLP-1,
then the effect of the blockade of GLP1R with Ex9 between T3 (difference between green vs yellow)
the untreated groups (difference between blue vs red) should be statistically significant. Given that
additional experiments providing a more accurate determination of the GLP-1 secretion kinetics
would be desirable.

**Response to the Reviewer's Comments:** We thank the reviewer for the comments and suggestion.
Regarding the reproducibility, all animal experiments in our previously submitted manuscript had
been repeated at least two to three times. Indeed, the regulation of GLP-1 production by either T3
or MB07811 treatment can be easily detected in mice under various conditions (Fig. 1f, k, 2b, 3a, e,
i, 4j, 6d, 7b, d, g and Supplementary Fig 1i, 2h, 3b, 7c, e). As the reviewer suggested, to substantiate
our conclusion, we repeated the experiment by using a larger sample size and obtained similar
results (Fig. 1n). Since Ex-9 treatment could attenuate the T3 effect on oral glucose tolerance, we
speculate that GLP-1 is critically involved in the T3 action on glucose metabolism. As suggested,
we also compared the changes in AUC after Ex-9 treatment between MMI+T3 mice and MMI mice.
In line with the data that the GLP-1 levels were higher in MMI+T3 mice than those in MMI mice,
a significantly larger effect of Ex-9 treatment was observed in MMI+T3 mice than that in MMI
mice, further supporting our notion that GLP-1 is critically involved in the T3 action on glucose
metabolism (Fig. 1n and Supporting data Fig. 1a-c).

To be noted, as the T3 levels were decreased but still detectable after MMI treatment, therefore, we
speculate the retained T3 action might contribute to the observed effect of Ex-9 in MMI mice (Fig.
1n and Supporting data Fig. 1a). Nevertheless, since the sensitivity and specificity of different assays
vary and systemic administration of T3 would affect other metabolic tissues, we could not totally
rule out the possibility that other mechanisms might also be involved here. Indeed, in this study, we
intended not to claim that the observed beneficial effect of systemic T3 treatment were solely due
to the increased GLP-1 production. Based on our findings in this study, we prefer to hypothesize

that the beneficial effect of hepatic TR β -mediated T3 effect might be primarily attributed to the
regulation of GLP-1 production by T3 via BA-mediated FXR antagonism.

Furthermore, as the reviewer suggested, we determined the kinetics by measuring the GLP-1 and
insulin levels during the oGTT assay. We found that the GLP-1 levels were higher before glucose
oral ingestion and elevated more markedly after glucose oral ingestion in MMI+T3 mice than those
in MMI mice (Supplementary Fig. 1e). Similar results were observed for the insulin levels during
the oGTT assay (Supplementary Fig. 1f). These data indicate that the capacity of GLP-1 production
was enhanced in MMI+T3 mice as compared to MMI mice.

The T3-induced improvement in glucose tolerance were mimicked by treatment with a liver-biased
TRb agonist and were is significantly attenuated in mice lacking TRb expression in the liver
(although, and again, critical Albcre⁺ control mice were not investigated; additional details about
the generation including targeting methodology, genomic context and general phenotype of the flox
mouse should provided).

**Response to the Reviewer's Comments:** We thank the reviewer for the comments and suggestion.
Liver-selective TR β knockout (LTR β KO) mice were generated by cross-breeding of TR β flox/flox
mice with loxP sites flanking the fifth TR β exon (TR β Floxed mice), developed by Shanghai Model
Organism Center, Inc., with mice harboring Cre-recombinase under the control of albumin promoter
(Alb-Cre mice) (Supplementary Fig. 2d, e). We provided detailed information, including targeting
methodology, genetic background, and general phenotype of the TR β Floxed (TR $\beta^{f/f}$) mice in our
revised manuscript (Supplementary Fig. 2d and Page 16, Line 528-532). Briefly, TR β Floxed (TR $\beta^{f/f}$)
mice were fertile and appeared indistinguishable from TR $\beta^{+/+}$ control littermates. There was no
significant difference in body weight, food intake, glucose and insulin levels between TR $\beta^{f/f}$ mice
and TR $\beta^{+/+}$ controls (Supporting data Fig. 1d). As the reviewer suggested, we also compared the
LTR β KO mice (TR $\beta^{f/f}$, Alb-Cre⁺) with Alb-Cre⁺ control mice (TR $\beta^{+/+}$, Alb-Cre⁺). As expected, the
MB07811 effects on GLP-1, insulin, and glucose levels were abolished in LTR β KO mice lacking
hepatic TR β as compared to Alb-cre⁺ control mice (Supplementary Fig. 3b).

Nonetheless, this is an important experiment that compellingly demonstrate the role of hepatic T3
signaling in modulating glucose metabolism. There is no question that this TRb mediates some of
the benefits of T3 and MB07811 on glycemc control. However, significant unknowns arise from
this experiment that question the main hypothesis. For instance, loss of hepatic TRb3 signaling in
liver does not result in dramatic increases in Cyp8b1 levels compared to controls, as it would be
expected in light of the effect of MMI treatment. Since changes in BA composition were not
investigated either, it is hard to attribute the loss of effect of T3 in the KO to changes in BA
composition. Considering this, MMI treatment of liver-specific TRb KO mice, plus minus T3, would
shed light on the actual contribution of hepatic TRb signaling on the benefits seen with the T3
replacement in the MMI treated mice. If worsening in glycemc control (note that the TRb KO
exhibit normal glucose tolerance) is seen and that can be normalized with T3 treatment, then other
options should be considered.

**Response to the Reviewer's Comments:** We thank the reviewer for the comments and suggestion.
In our study, loss of TR β in liver only slightly increased the mRNA levels but not the protein levels
of CYP8B1 (Fig. 4i, f), suggesting compensatory mechanisms may exist. In line with these results,
no differences in ileal BA composition (percentages of 12 α -OH BAs and non-12 α -OH BAs) (Fig.

4n, 5d) and GLP-1 production (Fig. 2b, 3e and Supplementary Fig. 2h) were observed between
LTR β KO and Floxed mice. In contrast, MMI treatment led to an elevation of CYP8B1 levels,
accompanied with altered BA composition (Supplementary Fig. 4h, i) and GLP-1 production
(Supplementary Fig. 1n). This is not surprising because it has been proposed that, unlike steroid
hormone receptors, TRs can act in the absence of the ligand as aporeceptors (apoTRs), which have
an intrinsic activity rather than being silent. These apoTRs repress basal transcription of positively
regulated genes and stimulate that of negatively regulated genes. In agreement with this notion,
unliganded TRs (apoTRs) in MMI mice might act to increase the mRNA expression of CYP8B1,
thereby altering the BA composition and impairing glucose metabolism (Supplementary Fig. 1k, l,
4h, i). It is worth noting that, as a matter of fact, more severe defects were normally observed in
hypothyroid subjects than those subjects harboring TR mutations.

As the reviewer suggested, we rendered LTR β KO mice hypothyroid followed by T3 treatment. In
line with our previous findings, T3 treatment markedly repressed the mRNA expression of CYP8B1
and lowered the glucose levels in MMI-treated Floxed mice, while loss of hepatic TR β could block
these effects in MMI-treated mice, further supporting our hypothesis that hepatic TR β signaling
contributes to the benefits seen with the T3 treatment (Supporting data Fig. 1e). It has been proposed
that TH deprivation induces a strong apoTR activity, while the effect of the absence of TH can be
attenuated by the removal of TR [1]. Interestingly, the mRNA levels of CYP8B1 were not reduced
in MMI-treated LTR β KO mice compared to MMI-treated Floxed mice. Moreover, the glycemic
control was not improved after the removal of hepatic TR β in MMI-treated mice (Supporting data
Fig. 1e). These observations indicate that apoTR α but not apoTR β may play a more important role
at least in the control of CYP8B1 transcription, which agrees with the previous notion that TR β may
have no aporeceptor activity in liver [1]. Notably, our findings are similar to those observed for TSH,
another negatively regulated target gene of T3. It has been proposed that TR α but not TR β seems to
be responsible for aporeceptor-mediated activation of TSH. Because whether apoTRs are silent or
whether they have an intrinsic activity should be determined by measuring the gene activity in the
absence of TH and in the absence of individual TR isoforms, to fully understand the role of hepatic
apoTRs in metabolic regulation, a more careful examination is needed in the future. Anyway, to the
best of our knowledge and according to our data, we believe that physiological homeostasis depends
on a precise balance between apoTRs and holoTRs, the apoTRs participate in the fine-tuning of T3-
target genes, and the combination of holoTRs with active apoTRs permits a larger amplitude of
transcriptional responses to moderate variations in T3 concentrations. Because we mainly focus on
the beneficial T3 effect and the role of liganded TR β (holoTR β) in current study, we provided the
related data from MMI-treated LTR β KO mice as supporting data (Supporting data Fig. 1e, f) to
avoid confusion due to the potential contribution from apoTR.

Additionally, we noticed that lacking hepatic TR β abolished the T3 effect on CYP8B1 expression
and glucose tolerance in MMI-treated mice, which further supports our proposed model involving
the action of hepatic TR β and incretin GLP-1 (Supporting data Fig. 1e, f). We also noticed a small
reduction of glucose levels in MMI-treated LTR β KO mice after T3 treatment, although the
difference did not reach statistical significance ($p=0.06$) (Supporting data Fig. 1e, left panel). These
results indicate that systemic administration of T3 might also affect glucose metabolism via other
mechanisms, which seemed to be not as dominant as those mediated by hepatic TR β under these
experimental conditions. Based on our findings in this study, we believe we have discovered a novel

role of T3 and hepatic TR β in modulating glucose homeostasis, which involves the regulation of
GLP-1 production via BA -mediated FXR antagonism. Nevertheless, we did not intend to exclude
the possibility that other mechanism exists that also contribute the profound role of T3 in glucose
homeostasis.

Administration of T β MCA acid to MMI-treated mice recapitulated a similar effect to that of T3
replacement on glucose tolerance and Gcg expression, GLP-1 and insulin levels. Conversely, treated
with CA or the gut-bias FXR agonist Fexeramine reverted some of the effects of T3 replacement in
MMI-treated mice that are consistent with enhanced GLP-1 secretion. However, due to the concern
manifested above, this interaction between FXR signaling and T3 could be due to independent
events acting in parallel with different contributions. T3 treatment of MMI-treated, villin-cre:FXR
flox mice would be certainly more convincing. This is not unreasonable considering that other
modalities of BA signaling play a meaningful role regulating L-cell differentiation
(<https://doi.org/10.2337/db19-0764>)

**Response to the Reviewer's Comments:** We thank the reviewer for the comments and suggestion.
We agree with the reviewer that other mechanisms might also be involved in the regulation of
glucose metabolism by systemic administration of T3 in mice. Our findings in LTR β KO mice and
MB07811-treated animals support our hypothesis that hepatic TR β signalling plays a critical role in
modulating glucose homeostasis through the regulation of GLP-1 production via BA-mediated FXR
antagonism. As the reviewer suggested, to test whether intestinal FXR signalling was essential for
the T3 action observed in this study, intestine-specific FXR-null mice (IFXRKO) were employed
[24]. We found that the T3 effects on the mRNA expression of ileal proglucagon, GLP-1, insulin,
and glucose levels, and oral glucose tolerance were all abolished in IFXRKO mice in hypothyroid
state (Fig. 7a-c). Similar results were observed when MB07811 was used for treatment in IFXRKO
mice in hypothyroid state (Fig. 7d-f). We also found that the T3 effects on the GLP-1, insulin, and
glucose levels, and oral glucose tolerance were all attenuated in euthyroid IFXRKO mice
(Supplementary Fig. 7c, d). Similar data were obtained when MB07811 was used for treatment in
IFXRKO mice in euthyroid state (Supplementary Fig. 7e, f). These results indicate that intestinal
FXR is indispensable for the beneficial effect of either T3 or MB07811 on GLP-1 production and
glucose homeostasis. Additionally, consistent with our current knowledge of intestinal FXR
signalling and our working hypothesis, the GLP-1 production was increased, while the glucose
metabolism was improved in MMI-treated IFXRKO mice as compared to MMI-treated Floxed mice
(Fig. 7a-f). Notably, in line with the previous findings that intestinal FXR is not required for
maintaining the normal glucose homeostasis [2], the glucose metabolism was normal in IFXRKO
mice in the euthyroid state, suggesting that compensatory mechanisms may exist (Supplementary
Fig. 7c-f).

We also agree with the reviewer that other modalities of BA signaling might also play a role here.
We then tested whether increased T β MCA could have impact on L-cell differentiation. In cultured
enteroendocrine cells and intestinal organoids, T β MCA treatment not only increased the GLP-1
expression and secretion (Fig. 6e, g and Supplementary Fig. 6c-e) but also elevated the mRNA
expression of Ngn3, NeuroD1, and Arx, which are key genes associated with L-cell differentiation
and endocrine specification, indicating that the increased T β MCA might also be able to promote the
L-cell differentiation (Supplementary Fig. 6g). Accordingly, elevated mRNA levels of Ngn3,
NeuroD1, and Arx were observed after T3 or MB07811 treatment in the ileum of MMI mice

(Supporting data Fig. 1g). Given that paracrine GLP-1 signalling has been implicated in L-cell
differentiation [3], based on our data, we speculate that activation of hepatic TR β signalling might
increase the T β MCA levels in ileum, subsequently promoting the GLP-1 secretion in L-cells, which
would act in both endocrine and paracrine manners to modulate insulin secretion and glucose
metabolism and enhance the endocrine function of L-cell by recruiting more L-cells, respectively.

**Reviewer #2 (Remarks to the Author):**

In the current study by Yan et al., hepatic activation of T3 signaling is sufficient to promote the
insulin secretion and lower the glucose levels. Furthermore, the team identified that this is due to
reduction in CYP8B1 and increase in bile acids that are FXR antagonists. Based on one study
reported in 2015 that inhibition of FXR in L cells increase GLP1 production and secretion, the
authors concluded that the T3-mediated induction in proglucagon (precursor of GLP-1) in L cells is
due to intestinal inhibition of FXR.

While the conclusion is exciting, there are some concerns:

1. How about the regulation of T3 on other genes involved in bile acid synthesis? What about levels
of bile acids that are TGR5 activators?

**Response to the Reviewer's Comments:** We thank the reviewer for raising these questions. In line
with previous findings, the mRNA expression of other enzymes involved in BA synthesis, such as
CYP7A1, CYP27A1, and CYP7B1, was positively regulated by T3 treatment (Supporting data Fig.
2a). Based on our knowledge and our findings, we hypothesize that T3 treatment not only promotes
BA synthesis by regulating the expression of these enzymes but also modulates the BA composition
by targeting CYP8B1. Our new data demonstrate that knockdown of CYP8B1 could attenuate the
effect of T3 treatment on the GLP-1 production and glucose metabolism in MMI mice, strongly
supporting the notion that CYP8B1 might be the primary BA synthetic enzyme responsible for the
observed glucose-lowering effect of T3 (Fig. 4j-l and Supplementary Fig. 4g).

Additionally, as the reviewer suggested, we analyzed the levels of BAs with TGR5 agonist activities.
We found that T3 treatment decreased the total percentage of BAs with TGR5 agonist activities in
the ileum of MMI mice, while the percentage of DCA, which is abundant in ileum and exhibits
potent TGR5 agonist activity, was not altered (Supplementary Fig. 5f). As we did not observe an
elevation in the levels of these TGR5-agonistic BAs, we speculate that FXR-mediated pathway
rather than TGR5-mediated pathway is critically involved after T3 treatment in mice. Consistently,
in human feces, positive correlation was also not observed between the T3 level and the percentage
of potent TGR5-agonistic BAs (LCA and DCA), while a negative correlation was observed between
T3 level and the percentage of total TGR5-agonistic BAs (Supplementary Fig. 8c, d), further
indicating that TGR5 signalling might not play a considerable role here.

2. What is the effects of T3 on carbohydrate ingestion and other incretin secretion in addition to
GLP-1?

**Response to the Reviewer's Comments:** We thank the reviewer for raising this question. In order
to investigate whether T3 could impact GLP-1 production directly in intestinal L-cells, we employed
enteroendocrine STC-1 and NCI-H716 cells and mouse intestinal organoids. We found that T3 had
no effect on GLP-1 production and expression in enteroendocrine cells and intestinal organoids

(Supplementary Fig. 2a-c), suggesting that T3 might regulates GLP-1 production in a cell-
nonautonomous manner. As the reviewer suggested, we examined the expression of Sglt1 and Glut2,
two key transporters involved in carbohydrate ingestion in these enteroendocrine cells and intestinal
organoids after T3 treatment. We found that T3 treatment had no effect on Sglt1 and Glut 2 mRNA
levels (Supporting data Fig. 2b, c). Similar results were observed in the ileum of T3 or MB-treated
MMI mice (Supporting data Fig. 2d). These data suggest that glucose absorption might not play a
significant role here. We also would like to point out that the plasma levels of GLP-1 were higher
in MMI+T3-5d mice than those in MMI mice before glucose challenge in oGTT assay (0 min, Fig.
1e and Supplementary Fig. 1e). Moreover, the GLP-1 expression and plasma levels of GLP-1 were
also higher in MMI+T3-5d mice than those in MMI mice without any treatment (Fig. 1f, g, k, 4j,
6d, 7a, b, g and Supplementary Fig. 1g). These data suggest that the basal level of GLP-1 production
or the capacity of GLP-1 production has been already increased or enhanced by T3 treatment, which
is independent of glucose ingestion.

Additionally, as suggested, we investigated the T3 effect on the expression of PYY, another gut
hormone produced by L-cells. We found that the mRNA expression of PYY was not altered in the
enteroendocrine STC-1 and NCI-H716 cells, mouse intestinal organoids, as well as the ileum of
mice after T3 or MB07811 treatment as indicated (Supporting data Fig. 2d, e). We also examined
the production of another incretin, glucose-dependent insulinotropic polypeptide (GIP). We found
that, in contrast to GLP-1, the plasma GIP levels and the ileal GIP mRNA levels were not altered
after T3 or MB07811 treatment in mice (Supplementary Fig. 1o). We also could not observe any
changes in GIP mRNA expression in the intestinal organoids after T3 administration
(Supplementary Fig. 1o). These results suggest that GIP might not be involved in the regulation of
glucose metabolism by TH observed in this study. Based on these results and others including the
new data obtained from *in vitro* experiments using BAs (Fig. 6e-h and Supplementary Fig. 6c-h),
we propose that T3 regulates GLP-1 production in a cell-nonautonomous manner via BA-mediated
FXR antagonism.

3. Validation of TGR5 activation. How about comparing to report showing that FXR can induce
TGR5? How about the activation status in the intestine epithelial cells and also in L cells.

**Response to the Reviewer's Comments:** We thank the reviewer for the suggestion. As the reviewer
suggested, to test the TGR5 activation, we first measured the cAMP levels in the ileum of mice and
found that either T3 or MB07811 treatment could not affect the cAMP levels (Supplementary Fig.
5h). We also employed enteroendocrine STC-1 and NCI-H716 cells and mouse intestinal organoids
to investigate the effect of T3 and BAs on TGR5 activation. Interestingly, we found that T3
treatment had no effect on cAMP levels in these enteroendocrine cells and intestinal organoids
(Supplementary Fig. 5i), suggesting that T3 could not affect the GLP-1 production in L-cells by
directly activating TGR5 signalling. We then treated enteroendocrine cells and intestinal organoids
with T β MCA. We found that T β MCA could increase GLP-1 expression and production, and
decrease the mRNA expression of SHP, a downstream target of FXR, in STC-1 cells, NCI-H716
cells, and intestinal organoids (Fig. 6e, g and Supplementary Fig. 6c-e). Meanwhile, we found that,
in contrast to potent TGR5-agonistic BAs (DCA etc.), T β MCA treatment did not elevate the cAMP
levels in STC-1 cells, NCI-H716 cells, and intestinal organoids (Supplementary Fig. 6h), suggesting
that the increased T β MCA after T3 treatment would not activate TGR5 signalling in mice. These
results together with other findings in this study, suggest that the increased ileum T β MCA after T3

treatment might promote the GLP-1 production in L-cells through FXR inhibition but not through
TGR5 activation.

Additionally, as suggested, we checked the expression of TGR5 after FXR inhibition or activation.
In agreement with previous reports showing the regulation of TGR5 by FXR [4, 5], we found that
ileal TGR5 mRNA expression was decreased in MMI-treated IFXRKO mice (mice lacking intestinal
FXR) regardless of T3 administration (Supporting data Fig. 2f). Consistently, either CA or FEX
treatment led to an elevation of TGR5 mRNA levels in the ileum of mice (Supporting data Fig. 2g).
Also consistent with above-mentioned reports and our current findings, the TGR5 mRNA
expression was decreased by T β MCA treatment in STC-1 cells, NCI-H716 cells, and intestinal
organoids (Supporting data Fig. 2h).

4. In humans, inhibition of CYP8B1 will lead to the bile acid pool enriched in CDCA, which is the
most FXR endogenous ligand---opposite to rodents.

**Response to the Reviewer's Comments:** We thank the reviewer for the comments. We agree with
the reviewer that inhibition of CYP8B1 will lead to the BA pool enriched in CDCA in human. To
test whether BAs with FXR antagonist activities have any effects on the FXR activities in the
presence of CDCA, we determined the effect of BAs with FXR antagonist activities on the FXR
signalling and GLP-1 production in the presence of CDCA *in vitro* using human enteroendocrine
NCI-H716 cells. Interestingly, we found that treatment of FXR-antagonistic BAs could reduce the
SHP mRNA levels and increase the GLP-1 production (Fig. 6h and Supplementary Fig. 6f). These
data suggest that FXR-antagonistic BAs can antagonize the effect of CDCA on FXR activities. Thus,
based on these data, we speculate that changes in FXR-antagonistic BA levels would influence FXR
activities, thereby modulating the production of GLP-1 in L-cells.

5. How about using T4 than T3? which is more commonly used in the clinic for the treatment of
hypothyroidism.

**Response to the Reviewer's Comments:** We thank the reviewer for the comments and suggestion.
As the reviewer suggested, we tested the effect of T4. To avoid the adverse effect of hyperthyroidism
induced by T4 treatment, a low dose of T4 (at a dosage of 60ug/100g BW) was used. We found that
treatment of T4 for 5 days decreased glucose levels, increased oral glucose tolerance, and enhanced
insulin and GLP-1 production in MMI mice, which were very similar to the effects observed for T3
treatment (Supplementary Fig. 1j).

6. In fig 6, it will be interesting to have additional control groups treated with CA or FEX with or
without MMI.

**Response to the Reviewer's Comments:** We thank the reviewer for the suggestion. We provided
the data showing the effects of CA and FEX in mice with or without MMI treatment (Supporting
data Fig. 2i-k). In agreement with our working hypothesis, MMI treatment resulted in a decrease in
GLP-1 production and insulin levels but an increase in glucose levels. Either CA or FEX treatment
did not affect the levels of GLP-1, insulin, and glucose in MMI mice. These data suggest that
activating FXR by administration of endogenous or synthetic agonist would not further suppress the
GLP-1 production and worsen the glucose metabolism in MMI mice probably because FXR was
already activated due to the decreases in FXR-antagonistic BAs in enteroendocrine L-cells. In the
mice without MMI treatment (CT mice), FEX had no significant effect on these parameters, while

CA slightly decreased glucose levels, increased GLP-1 expression and production, and insulin level,
although some of the differences did not reach statistical significance. We interpret these two results
as follows.

We did not observe any FEX effect on the glucose and GLP-1 levels in normal mice, which is not
surprising (Supporting data Fig. 2i, k). It has been reported that 5 weeks of FEX treatment, which
resulted in intestinally restricted FXR activation, could affect glucose homeostasis in obese mice
but not in normal mice, which might involve the browning of adipose tissues and weight loss [6]. It
has also been reported that 7 days of FEX treatment could increase GLP-1 production and oral
glucose tolerance in mice in an either FXR- or TGR5-dependent manner, which might involve the
action of gut microbiota [5]. Notably, in the same study, 9 days of FEX treatment could increase the
GLP-1 production and improve oral glucose tolerance in obese mice, which was accompanied by a
decrease in body weight [5]. In a recent paper, 28 days of FEX treatment did not affect GLP-1
production and oral glucose tolerance in mice (6 weeks old) [7]. In line with these data, we also did
not observe the effects of FEX treatment on the levels of glucose, insulin, and GLP-1 and body
weight in normal mice (CT mice) (Supporting data Fig. 2i, k), suggesting that some of FEX effects
observed in obese mice might be attributed to the weight loss after long-term FEX treatment, which
might involve gut microbiota-mediated TGR5 signaling. Interestingly, in our study, 5 days of FEX
treatment could attenuate the T3 effects on the levels of glucose, insulin, and GLP-1 in MMI-treated
mice without changing the body weight (Fig. 7g). Based on these findings and our other data, we
speculate that inactivation of FXR signalling might be critically involved in hepatic T3 signalling-
mediated regulation of GLP-1 production and glucose metabolism, which did not require either the
weight loss or the involvement of TGR5 signaling. Nevertheless, we did not intend to exclude the
possibility that other mechanism exists that also contribute the profound role of T3 in glucose
homeostasis.

BAs function as endogenous ligands for FXR, which has a complex role in metabolic homeostasis.
FXR total knockout mice developed hyperglycemia on a normal chow diet, while they exhibited
improved glucose homeostasis on a high-fat diet [8, 9]. Although contradictory observations have
been reported using systemic FXR agonists, beneficial effects were normally observed in chow-fed
mice, whereas exacerbated glucose intolerance is seen under pathophysiological conditions such as
obesity [4, 10-12]. Consistent with a previous report showing that CA feeding could decrease fasting
glucose by approximately 50% in mice [13], in this study, we also detected a beneficial effect of CA
treatment in normal mice, as evident from reduced glucose levels and increased GLP-1 expression
and production, although some of the differences did not reach statistical significance (Supporting
data Fig. 2j). However, in MMI mice, CA lost its beneficial effect, probably due to the already
enhanced intestinal FXR signalling (Supporting data Fig. 2j). Importantly, consistent with current
knowledge for intestinal FXR, administration of CA significantly attenuated T3-induced elevation
of GLP-1 and insulin levels and the glucose-lowering effect of T3 in MMI mice (Fig. 6d). These
data further support the notion that FXR is essential for normal glucose homeostasis and inhibition
of FXR might serve as an approach in glycemic control under disease states.

**Reviewer #3 (Remarks to the Author):**

The authors demonstrate that euthyroid animals have improved glucose tolerance and increased
circulating insulin and circulating and intestinal GLP-1 levels compared to hypothyroid animals.

Using a wide range of models the authors demonstrate that liver TRB is required for (the majority
of) this effect. Intestinal FXR is shown to modulate the beneficial effect of T3 on GLP-1 levels,
plasma insulin and plasma glucose in hypothyroid mice. The authors postulate that the link between
(hepatic) thyroid hormone signaling and changes in bile acid composition, resulting in increased
GLP-1 secretion, is the enzyme CYP8B1 in liver. The data is novel, interesting and clinically
relevant given the development of liver specific TRB agonists for the treatment of metabolic disease.
Specific Comments:

1. Is hyperglycemia commonly found in hypothyroid mouse models. To this reviewers knowledge
the answer is no. What is the reason for this here. Other examples should be brought forward.

**Response to the Reviewer's Comments:** We thank the reviewer for the comments and suggestion.
As we mentioned in the introduction of our manuscript, the role of TH in glucose metabolism is
profound. Abnormal glucose homeostasis has been noticed in hypothyroid patients and animal
models of hypothyroidism, although the proposed underlying mechanisms sometimes differ among
different studies [14-20]. As the reviewer suggested, we searched literatures published recently and
listed just a few here: 1) a PNAS paper contributed by Dr. Carrasco, reviewed by Dr. Hollenberg
and Dr. Moore, showed impaired glucose tolerance in LID mice in Fig. 3A; 2) similar result could
be found in a Diabetes paper by Dr. Kieffer; 3) increased fasting glucose levels in hypothyroid rats
were shown in a paper by Dr. Ayuob; 4) in addition to animal studies, low fasting glucose levels
were observed in patients with SCH in two papers reported by Dr. Gao.

As the suggested, we employed propylthiouracil (PTU) to induce hypothyroidism in mice. We also
observed increased glucose levels, impaired oral glucose tolerance, and decreased insulin and GLP-
1 levels in these hypothyroid mice induced by PTU administration (Supporting data Fig. 3a).

2. The increased levels of GLP-1 described in Figure 1J are not clear.

**Response to the Reviewer's Comments:** We thank the reviewer for the comments. As the reviewer
suggested, we provided another image with stronger staining intensity showing the increased GLP-
1 levels (Fig. 1g). To substantiate our conclusion, we performed immunohistochemistry for GLP-1
and obtained similar results (Supplementary Fig. 1g). We have provided these data in our revised
manuscript.

3. Why was GLP-1 not given directly to hypothyroid animals?

**Response to the Reviewer's Comments:** We thank the reviewer for the comments. As suggested,
we tested whether GLP-1 administration could improve the glucose homeostasis in hypothyroid
mice (MMI mice). In agreement with our working model, GLP-1 treatment increased insulin levels,
decrease glucose levels, and enhance oral glucose tolerance in MMI mice (Fig. 1h-j), suggesting
that T3-induced elevation of GLP-1 levels had the capacity to improve glucose metabolism in
hypothyroid mice.

4. Ideally, further experiments should be provided to demonstrate KO of the TR in the liver. ie TH
mediated gene expression of classic TR target genes. The model is under-described in all aspects.

**Response to the Reviewer's Comments:** We thank the reviewer for the comments. As the reviewer
suggested, we examined the classic T3 target genes (Dio1, Me, Scd1, and Sp14) in the liver of
LTR β KO mice with or without T3 treatment (Supporting data Fig. 3b). We found that the mRNA

expression of positively regulated genes (Dio1, Me, and Sp14) was decreased in the liver of
LTR β KO mice. T3 could increase the mRNA expression of these positively regulated genes in the
liver of Floxed mice but not LTR β KO mice. These data agree with the notion that TR β is the major
TR isoform in liver. For Scd1, a T3 negatively regulated gene, the expression pattern is very similar
to that of CYP8B1 (Fig. 4i). As suggested, we provided more information for Floxed mice and
LTR β KO mice in our revised manuscript, including details about targeting methodology, genomic
context, and general phenotype (Supplementary Fig. 2d-i, 3b and Supporting data Fig. 1d, Page 6,
Line 185-188 and Page 16, Line 528-532).

5. In Figure 2 and Figure 3 the induction of hypothyroidism leads to a rise in blood glucose in both
WT and KO animals and T3 or the analog reduces it only in WT or floxed animals. Why is there an
induction of BG in hypothyroidism in the absence of the TR?

**Response to the Reviewer's Comments:** We thank the reviewer for the comments. The data shown
in Fig. 2 and Fig. 3 (previously version) were not from hypothyroid animals. In this study, we did
not observe abnormal glucose levels in LTR β KO mice (Fig. 2d, 3g and Supplementary Fig. 2i),
which was consistent with the observation that loss of TR β in liver only slightly increased the mRNA
levels but not the protein levels of CYP8B1 (Fig. 4f, i) and had no effect on ileal BA composition
(Fig. 4n, 5d) and GLP-1 production (Fig. 2b, 3e and Supplementary Fig. 2h), indicating that
compensatory mechanisms may exist. To investigate the blood glucose levels and T3 effect in
LTR β KO mice in hypothyroid state, we treated LTR β KO mice with MMI (Supporting data Fig. 1e).
Our observations are as follows: 1) In line with data obtained in wild type mice (Supplementary Fig.
1k, l, 4h), MMI treatment could increase CYP8B1 mRNA expression and glucose levels and impair
oral glucose tolerance in Floxed mice (Supporting data Fig. 1e, f). The finding that MMI treatment
had a more deleterious effect than loss of TR is not surprising, because unlike steroid hormone
receptors, TRs can act in the absence of the ligand as aporeceptors (apoTRs), which have an intrinsic
activity rather than being silent. These apoTRs repress basal transcription of positively regulated
genes and stimulate that of negatively regulated genes. We speculate that unliganded TRs (apoTRs)
in MMI mice might act to increase CYP8B1 mRNA expression, thereby altering the BA composition
and impairing glucose metabolism. 2) Consistent with our previous findings in euthyroid state (Fig.
2a-e, 4i), T3 treatment markedly repressed the mRNA expression of CYP8B1 and lowered the
glucose levels in MMI-treated Floxed mice, while loss of hepatic TR β could block these effects in
MMI-treated mice (Supporting data Fig. 1e, f), further supporting our hypothesis that hepatic TR β
signaling contributes to the benefits seen with the T3 treatment. 3) It has been proposed that TH
deprivation induces a strong apoTR activity, while the effect of the absence of TH can be attenuated
by the removal of TR. Here, we found that the mRNA levels of CYP8B1 were not reduced in MMI-
treated LTR β KO mice compared to MMI-treated Floxed mice (Supporting data Fig. 1e). Moreover,
the glycemic control was not improved after the removal of hepatic TR β in MMI-treated mice
(Supporting data Fig. 1e, f). These observations indicate that apoTR α but not apoTR β may play a
more important role at least in the control of CYP8B1 transcription, which agrees with the previous
notion that TR β may have no aporeceptor activity in liver. Since we mainly focus on the beneficial
T3 effect and the role of liganded TR β (holoTR β) in current study, we provided these data from
hypothyroid LTR β KO mice as supporting data to avoid confusion due to the potential contribution
from apoTR.

Additionally, we noticed that loss of hepatic TR β abolished the T3 effect on CYP8B1 expression

and oral glucose tolerance in MMI-treated mice, which further supports our proposed model
involving the action of hepatic TR β and incretin GLP-1 (Supporting data Fig. 1e, f). We also noticed
a small reduction of glucose levels in MMI-treated LTR β KO mice after T3 treatment, although the
difference did not reach statistical significance ($p=0.06$) (Supporting data Fig. 1e, left panel). These
data indicate that systemic administration of T3 might also affect glucose levels via other pathways
or mechanisms, which seemed to be not as dominant as those mediated by hepatic TR β under these
experimental conditions. Collectively, we believe we have discovered a novel role of T3 and hepatic
TR β in glucose homeostasis, which involves the regulation of GLP-1 production via BA-mediated
FXR antagonism. Nevertheless, we did not intend to exclude the possibility that other mechanism
exists that also contribute the profound role of T3 in glucose homeostasis.

6. In Figure 3 there is no data on the high fat diet mice ie food intake, body weight etc. It is
impossible to determine the effects of the MB compound in this setting.

**Response to the Reviewer's Comments:** We thank the reviewer for the comments. In line with
previous data reported by Dr. Erion et. al. [21], we found that MB07811 treatment had no effects on
food intake but could reduce the body weight and white fat mass in HFD-fed mice (Supplementary
Fig. 3a, c-g). We have added these data to our revised manuscript.

7. In Figure 4 where is the RNA-Seq data of the genes regulated by feeding. What does Cyp8b1 do
with increased feeding.

**Response to the Reviewer's Comments:** We thank the reviewer for the comments. As the reviewer
suggested, we provided the accession code of our microarray data for the genes regulated by feeding
(GSE184055) in our revised manuscript. CYP8B1 is highly regulated under various physiological
and pathological conditions. CYP8B1 is known to be increased by fasting and downregulated by
feeding through multiple mechanisms, including FXR-SHP/MAFG-mediated or FGFR4-mediated
negative feedback on BA *de novo* synthesis, thereby modulating the BA profile and the
hydrophobicity of BA pool to adapt to the feeding status or nutritional environment. Based on
current evidence, we speculate that fasting-induced CYP8B1 expression induces an increase in CA
production, which will facilitate the dietary lipid absorption for the next meal. On the other hand,
refeeding-induced downregulation of CYP8B1 can lead to an increase in GLP-1 production to
enhance the insulin action after the meal [22, 23]. Evidence also indicates that FoxO1 can regulate
the CYP8B1 expression in an FXR-independent manner, thereby modulating lipid homeostasis [24].
Regarding the expression of CYP8B1 upon increased feeding or HFD feeding, current evidences
are not conclusive. We speculate that it is probably due to the differences in the nutritional status (as
it is very sensitive to feeding), the protocol of HFD treatment, and the facility conditions. As the
altered expression of CYP8B1 could affect systemic insulin action by modulating GLP-1 and insulin
production, which might have either beneficial or deleterious effect on glucose and lipid metabolism
dependent on the feeding pattern or condition or disease status or stage. Nevertheless, based on
available knowledge, it has been proposed that targeting BA profile by inhibiting CYP8B1 might
be a promising therapeutic strategy for metabolic diseases, including T2D.

8. Also, in Figure 4, T3 is known to regulate many other genes involved in bile acid transport that
also regulate bile acid hydrophobicity including *cyp27a1* and *cyp3a11* and as well bile acid
transporters. These genes are not included in the analysis shown. Interestingly, *Cyp8b1* has been
identified previously as a target of TH signaling: Andersson U, Yang YZ, Bjorkhem I, Einarsson C,

Eggertsen G, Gafvels M. Thyroid hormone suppresses hepatic sterol 12 α -hydroxylase (CYP8B1)
activity and messenger ribonucleic acid in rat liver: failure to define known thyroid hormone
response elements in the gene. *Biochim Biophys Acta*. 1999;1438(2):167–174. Additionally, thyroid
hormone signaling has also been shown to alter bile acid composition previously and thus regulate
intestinal cholesterol absorption (Astapova et al, *JCI* 2014).

**Response to the Reviewer's Comments:** We thank the reviewer for the comments. We agree with
the reviewer that systemic T3 treatment can regulate many hepatic genes involved in metabolism
including BA metabolism. As the reviewer suggested, we analyzed our RNA-seq data (GSE184261)
and provided these data in our revised manuscript (Supplementary Fig. 4a) and cited related papers
[25, 26]. Indeed, as described in our manuscript, to understand the mechanism underlying the
glucose-lowering effect of hepatic TH signalling, we performed RNA-seq followed by KEGG
pathway analysis to identify hepatic genes or pathways regulated by T3 (Fig. 4a). KEGG pathway
analysis of DEGs revealed that 72 pathways were regulated by T3 treatment, including primary BA
biosynthesis (Fig. 4b). Given that intestinal GLP-1 is secreted postprandially and GLP-1-mediated
incretin effect contributes to the glucose-lowering effect of T3, we compared the T3-regulated
pathways with those pathways altered in response to oral intake of nutrients and identified 10
pathways that were regulated by both T3 administration and nutrition ingestion (Fig. 4b, c). We then
analyzed the overlapped gene sets in these 10 pathways and identified 22 genes (Supplementary Fig.
4b), including CYP8B1. As CYP8B1 has been implicated in the regulation of GLP-1 secretion and
glucose homeostasis [22]. We then speculated that CYP8B1 might be involved in the regulation of
glucose metabolism by T3. Furthermore, our new data demonstrate that AAV-mediated knockdown
of CYP8B1 could block the T3 effect on GLP-1 production, insulin and glucose levels, oral glucose
tolerance (Fig. 4j-1), further suggesting that CYP8B1 might be the primary regulator that mediates
the metabolic effect of T3 observed in this study.

We also agree with the reviewer that CYP8B1 is known to be regulated by T3. Since an early study
failed to define known TRE in the promoter region of rat CYP8B1 (about 2-kb upstream of the start
site), whether CYP8B1 is transcriptionally by TR β has been a longstanding mystery. To better
understand the regulation of CYP8B1 by T3, we analyzed recent ChIP-seq data (GSE159648)
reported by Lazar's lab [27] and identified a super-enhancer encompassing the mouse CYP8B1 gene
(Supplementary Fig. 4f). Moreover, we found that two putative TR binding sites (DR1 and DR4)
identified early are in this super-enhancer, which are in the intergenic region but not in the promoter
region of mouse CYP8B1 (Supplementary Fig. 4f). Our ChIP analysis revealed that TR β could be
recruited to the super-enhancer region containing these two binding sites (Fig. 4h). In line with our
above super-enhancer analysis, we observed H3K27 acetylation in the same region (Fig. 4h). These
results suggest that TR β might transcriptionally controls the CYP8B1 expression through a super-
enhancer-mediated mechanism. Additionally, our data obtained by using LTR β KO mice suggest that
the negative regulation of CYP8B1 by T3 requires TR β (Fig. 4f, i).

Regarding the previous studies by Astapova et al from Hollenberg's lab, they employed a unique
mouse model developed mice that express a mutant NCoR protein (L-NCoR Δ ID) that cannot
interact with the TR in the liver to explore the role of NCoR and TR. In their PNAS paper [25], they
found that positive T3 targets were up-regulated in L-NCoR Δ ID mice in the hypo- and euthyroid
state. Interestingly, 326 genes were activated in hypothyroidism (representing negatively regulated
TR/T3-target genes) in control mice, and only 3 of these genes were repressed (<1%) in hypothyroid

L-NCoRAID mice. The authors proposed that NCoR is a specific regulator of T3 action and
mediates repression by unliganded TR (apoTR) in hypothyroidism. Therefore, NCoRAID mice
might not be a suitable mouse model to study the T3 effect on negatively regulated T3 target genes
or the role of liganded TRs (holoTRs) in the negative regulation by T3. Later, in their JCI paper[26],
they found an alteration in the composition and hydrophobicity of BA pool in L-NCoRAID mice
upon 2% cholesterol feeding, accompanied with changes in the expression of genes involved in BA
synthesis (CYP27A1 and CYP3A11) and transport (ABCB11), which might eventually lead to a
decrease in cholesterol absorption. Consistent with their PNAS paper, they did not observe any
changes in CYP8B1 expression. They also showed that the mRNA expression of CYP27A1 and
CYP3A11 were downregulated in the hypothyroid state, however, unlike CYP27A1, the expression
of CYP3A11 was further suppressed by T3 treatment in hypothyroid mice. Thus, the T3 action in
BA metabolism, especially the T3 action mediated by liganded TRs (holoTRs) is still not fully
understood. Notably, although CYP27A1 is the first enzyme in the alternative BA synthetic pathway,
it also participates in classic BA synthesis. Although CYP3A11 has been proposed to participate in
alternative bile acid synthesis, loss of CYP3A11 would not affect BA composition [28]. As the
regulation by BA composition by NCoRAID had not been investigated in the absence or after
knockdown of downstream effectors, thus, whether these two enzymes are the primary players in
hypothyroidism or the major players mediating the effect of apoTR in the euthyroid state requires
further investigation.

It is also worth noting that these authors only investigated the BA profiles in L-NCoRAID mice,
which provides indirect evidence for the role of TH signalling in BA metabolism. Moreover,
previous studies mainly focused on the role of BAs in nutrition absorption, whether TH signalling
could use BAs as essential endocrine molecules to control metabolic homeostasis is unknown.
Furthermore, given that NCoR only interacts with TR in the absence of T3, NCoR may also bind to
other transcription factors, and therefore the profound T3 effects, especially those effects mediated
by liganded TRs (holoTRs) or negatively regulated target genes of T3, on BA metabolism and
corresponding physiological consequence remain unclear and require extensive investigation.

Indeed, we observed similar results for those genes reported by Astapova et al from our RNA seq
analysis, as mentioned above, we provided these data with other genes involved in BA synthesis
and transport in our revised manuscript (Supplementary Fig. 4a). Whether these genes involved in
BA transport and the regulation of BA hydrophobicity also play a role in the regulation of glucose
homeostasis by T3 requires future investigation. Nevertheless, since CYP8B1 is responsible for CA
synthesis, it is destined to be a master regulator of BA pool composition. Moreover, as we mentioned
above, CYP8B1 is regulated by nutritional status, loss of CYP8B1 has been shown to be able to
promote GLP-1 production thereby modulating glucose homeostasis, and knockdown of CYP8B1
could attenuate the T3 effect on GLP-1 production and glucose metabolism (Fig. 4j-1), we propose
that CYP8B1 might be the primary regulator that mediates the metabolic effect of T3 observed in
this study.

9. The explanation of the data in Figure 7 is not clear. TSH is used to determine TH action. Do the
TSH levels inversely correlate with the TH levels. Additionally do the T3 levels correlate with other
actions of TH on the liver ie serum cholesterol.

**Response to the Reviewer's Comments:** We thank the reviewer for the comments. We agree with

the reviewer that TSH is normally used as an indicator for the changes of function in HPT axis,
moreover, it is also the only diagnostic indicator for subclinical thyroid disease. However, the
association between TSH and THs is weak or not significant or even “unexpected” within normal
arrange, which is not surprising probably due to many compensatory mechanisms and feedback
loops that maintains the levels of each hormone. In two cohorts used in our study, we did not detect
any significant association between TSH and T3/T4 (Supporting data Fig. 3c, d, g, h). It is also
worth noting that, in normal subjects, when the metabolic regulation is normal, the association
between two hormones that are regulated reciprocally is hard to detect. For instance, the association
between glucose and insulin is normally not seen in normal subject. Although we did not observe
an association between TSH levels and the levels of GLP-1 or insulin, the levels of non-12 α -OH
BAs, or the ratios of 12 α -OH BAs to non-12 α -OH BAs in this study (Supporting data Fig. 3e, f, i,
j), we could not rule out the possibility that we would detect such an association in a larger cohort.
As suggested, we also determined the association between T3 and total cholesterol levels in two
cohorts we used in this study. As expected, we observed an inverse correlation between T3 and
cholesterol levels in the plasma in the larger cohort (Supplementary Fig. 8a). A similar trend could
be observed in the small cohort, although the association was not statistically significant probably
due to the limited sample size (Supplementary Fig. 8b).

10. Is there a CYP8B1-KO model available or specific inhibitor that would allow you to study the
effect of T3 in the absence of CYP8B1?

**Response to the Reviewer's Comments:** We thank the reviewer for the comments. As CYP8B1-
KO model is not available in the lab, knockdown of hepatic CYP8B1 was achieved by using adeno-
associated virus (AAV) expressing shRNA specific for CYP8B1. As expected, AAV-mediated
knockdown of hepatic CYP8B1 could block the T3 effect on GLP-1 production, insulin and glucose
levels, oral glucose tolerance (Fig. 4j-l and Supplementary Fig. 4g), further supporting our notion
that CYP8B1 might be the primary regulator that mediates the metabolic effect of T3 observed in
this study.

11. Does the attenuated change in CYP8B1 in MMI+T3 treated L-TRBKO mice result in measurable
changes in intestinal bile acid composition compared to controls?

**Response to the Reviewer's Comments:** Yes. To test whether the attenuated change in CYP8B1
expression levels in T3-treated LTR β KO mice, as shown in Fig. 4f, would result in measurable
changes in intestinal BA composition, we determined the ileal BA composition in these four groups
of mice. Consistent with the CYP8B1 expression levels as shown in Fig. 4f, we found that the ileal
BA composition (percentages of non-12 α -OH BAs and 12 α -OH BAs) was not altered in LTR β KO
mice as compared to Floxed mice, was changed after T3 treatment in Floxed mice, and loss of
hepatic TR β could abolished the T3 effect on BA composition (Fig. 4n, 5d).

**Other comments:**

12. Figure 2: . Please specify if these mice were rendered hypothyroid before being treated with T3
as this isn't mentioned in the figure legends or corresponding results section

**Response to the Reviewer's Comments:** We thank the reviewer for the comments. Regarding
Figure 2, these LTR β KO mice and Floxed mice were not rendered hypothyroid before T3 treatment.
We have performed new experiments and provided the data from MMI-treated LTR β KO mice with

or without T3 treatment in Supporting data Fig. 1e, f.

13. The other bile acid receptor TGR5 has been described to be crucial for GLP-1 induction and
several other metabolic effects. Did you find any evidence for a role of TGR5 in this pathway? Why
did you choose to focus only on FXR?

**Response to the Reviewer's Comments:** We thank the reviewer for the comments. As suggested,
we explored the role of TGR5 in this study. Similar to FXR, we did not detect any changes in mRNA
expression of TGR5 in the ileum of MMI mice after T3 or MB07811 treatment (Supplementary Fig.
5j). We then analyzed the levels of BAs with TGR5 agonist activities. We found that T3 treatment
decreased the total percentage of BAs with TGR5 agonist activities in the ileum of MMI mice, while
the percentage of DCA, which is abundant in ileum and exhibits potent TGR5 agonist activity, was
not altered (Supplementary Fig. 5f). As we did not observe an elevation in the levels of these TGR5-
agonistic BAs, we speculate that FXR-mediated pathway rather than TGR5-mediated pathway was
critically involved after T3 treatment in mice. Consistently, positive correlation was not observed
between the T3 level and the percentage of BAs with potent TGR5 agonist activities in human feces,
further indicating that TGR5 signalling might not play a considerable role here (Supplementary Fig.
8c, d).

To further test whether TGR5 signalling were activated, we first measured the cAMP levels in the
ileum of mice and found that either T3 or MB07811 treatment could not affect the cAMP levels
(Supplementary Fig. 5h). We also employed enteroendocrine STC-1 and NCI-H716 cells and mouse
intestinal organoids to investigate the effect of T3 and BAs on TGR5 activation. Interestingly, we
found that T3 treatment had no effect on cAMP levels in STC-1 cells, NCI-H716 cells, and mouse
intestinal organoids (Supplementary Fig. 5i), suggesting that T3 could not affect GLP-1 production
in L-cells by directly activating TGR5 signalling. Meanwhile, we found that, in contrast to potent
TGR5-agonistic BAs, T β MCA treatment did not elevate cAMP levels in these enteroendocrine cells
and intestinal organoids (Supplementary Fig. 6h), suggesting that the elevated T β MCA levels after
T3 treatment would not activate TGR5 signalling in mice. We also treated STC-1 and NCI-H716
cells and intestinal organoids with T β MCA. We found that T β MCA could increase GLP-1
expression and production (Fig. 6e, g and Supplementary Fig. 6c-e), and decrease the mRNA
expression of SHP, a downstream target of FXR, in these enteroendocrine cells and intestinal
organoids (Supplementary Fig. 6c-e). These results together with other findings in this study, further
suggest that the increased ileal T β MCA after T3 treatment might promote the GLP-1 production in
L-cells through FXR inhibition but not through TGR5 activation.

14. In general the paper is well written but the discussion would benefit from a thorough read-
through to correct several grammatical errors.

**Response to the Reviewer's Comments:** We thank the reviewer for the comments. We have
corrected a few mistakes and improved the discussion part in our revised manuscript.

15. Please specify where the MB07811 TRB agonist was purchased and include data or references
that demonstrate its liver and TRB specificity.

**Response to the Reviewer's Comments:** We thank the reviewer for the comments. MB07811 can
be either purchased from TargetMol or customized from Adooq. We recently synthesized MB07811
and its analogues with the help of Dr. Wenjun Tang from State Key Laboratory of Bio-Organic and

Natural Products Chemistry, Center for Excellence in Molecular Synthesis, Shanghai Institute of
Organic Chemistry, Chinese Academy of Sciences. We have validated its selectivity (for TR β) by
performing luciferase assay, its specificity (liver-targeting) by using THAI mice and its function
(cholesterol-lowering) by using MMI mice (Supporting data Fig. 3k-m).

16. Have you done any of these experiments in female mice? If so please show (in supplemental)

**Response to the Reviewer's Comments:** Thanks for the comments. Indeed, we had performed a
few key experiments in female mice and obtained similar results, suggesting that the regulation of
GLP-1 production and glucose homeostasis by T3 treatment can also be observed in female mice
(Supplementary Fig. 1i, 2h, i). Moreover, our recent data showed that deletion of hepatic TR β in
female mice could also diminish the T3 effects on the GLP-1 production and glucose homeostasis
(Supplementary Fig. 2h, i). Thus, we speculate that the underlying mechanism for the regulation of
glucose metabolism by T3 identified in this study is gender-independent.

17. Please state whether your human subjects provided informed consent in accordance with the
declaration of Helsinki.

**Response to the Reviewer's Comments:** We thank the reviewer for the comments. Human subjects
in this study provided informed consent in accordance with the declaration of Helsinki. We have
included this information in our revised manuscript.

**Supporting data Figure 1.**

(a and b) The data in Figure 1 n are shown separately. oGTT for MMI mice (a) and MMI+T3-5d
 mice (b) treated with saline or Ex-9 for 5 days and the AUC (n=10). (c) Δ AUC are calculated as
 follows: (AUC_{MMI+EX-9}-AUC_{MMI}) and (AUC_{MMI+T3-5d+EX-9}-AUC_{MMI+T3-5d}), according to panel a and
 b. (d) The BW, food intake, glucose and plasma insulin levels in TR β Floxed (TR $\beta^{f/f}$) mice and
 unfloxed mice (TR $\beta^{+/+}$) mice (n=8). (e) The blood glucose levels (left) and relative mRNA levels of
 CYP8B1 in the liver (right) of Floxed and LTR β KO mice treated with vehicle (CT), MMI or MMI
 and 5 days of T3 (n=5). (f) oGTT for Floxed and LTR β KO mice treated with vehicle (CT), MMI or
 MMI and 5 days of T3 (n=5) and the corresponding AUC for oGTT (n=5). (g) The relative mRNA
 levels of Ngn3, ND1, and Arx in the ileum of CT mice, MMI mice, and MMI mice after 5 days of
 T3 or MB treatment (n=5-6). Means \pm SEM are shown. *p<0.05, **p<0.01 and ***p<0.001.

Supporting data Figure 2.

(a) The relative mRNA levels of CYP7A1, CYP7B1 and CYP27A1 in the liver of CT, MMI and

MMI+T3-5d mice (n=8). (b and c) Relative mRNA levels of Sglt1 (b) and Glut2 (c) in mouse

intestinal organoids, STC-1 cells and NCI-H716 cells after T3 treatment (n=3). (d) The relative

mRNA levels of Sglt1, Glut2, and PYY in the ileum of CT mice, MMI mice, MMI mice after 5 days

of T3 or MB treatment (n=5-6). (e) Relative mRNA levels of PYY in mouse intestinal organoids,

STC-1 cells and NCI-H716 cells after T3 treatment (n=3). (f) The relative mRNA levels of TGR5

in the ileum of Floxed and IFXRKO mice treated with MMI or MMI and 5 days of T3 (n=6). (g)

The relative mRNA levels of TGR5 in the ileum of mice treated with or without CA or FEX for 5

days as indicated (n=5). (h) Relative mRNA levels of TGR5 in mouse intestinal organoids, STC-1

cells and NCI-H716 cells after TBMCA treatment (n=3). (i-k) Plasma active GLP-1 (i), blood

glucose and plasma insulin levels, relative proglucagon mRNA levels (j and k) in the ileum of CT

and MMI mice treated with or without CA or FEX for 5 days as indicated (n=5). Means \pm SEM are

shown. * $p < 0.05$, ** $p < 0.01$ and *** $p < 0.001$.

**Supporting data Figure 3.**

(a) Blood glucose levels, oGTT, plasma insulin and plasma active GLP-1 levels in mice treated with
 PTU (n=5). (b) Relative mRNA levels of Dio1, ME, Scd1 and Sp14 in the liver of Floxed and
 LTR β KO mice treated with PBS or T3 for 5 days (n=6). (c-f) Correlation between free T3 and TSH
 levels (c), free T4 and TSH levels (d), TSH levels and plasma active GLP-1 levels (e) or plasma
 insulin levels (f) in a cohort of euthyroid subjects (n=30). (g-j) Correlation between free T3 and
 TSH levels (g), between free T4 and TSH levels (h), between TSH levels and the percentage of fecal
 non-12 α -OH BAs (i), or between TSH levels and the fecal 12 α -OH/non-12 α -OH ratios (j) in another
 cohort of euthyroid subjects (n=19). (k) Luciferase assay showing that MB is an agonist with
 selectivity for TR β as compared to TR α . Pal-luc reporter plasmid was co-transfected with TR α or
 TR β plasmid in HepG2 cells followed by MB treatment. Reporter activity was analyzed and fold of
 induction by MB treatment was calculated. (l) Luciferase activity in tissue samples, including liver,
 adipose tissue (inguinal fat), kidney, and heart, from TH action indicator (THAI) mice treated with
 MB for 5 days (n=3). (m) Serum total cholesterol (TC) levels in MMI mice treated with MB for 5
 735 days (n=5). Means \pm SEM are shown. * p <0.05, ** p <0.01 and *** p <0.001.

**Materials and Methods for Supporting data**

**Mice studies.** Mice were rendered hypothyroid by adding 0.15% propylthiouracil (PTU) (T1309,
TargetMol) in their drinking water for four weeks. TH action indicator (THAI) mice were developed
previously [29], which harbor a TH-responsive luciferase reporting system. THAI mice received
MB07811 treatment at a dose of 5mg/kg/day for five days. Tissue samples were lysed in luciferase
lysis buffer, supernatant was collected after centrifugation at 14,000 g at 4°C 10 min. Luciferase
activity was determined with luciferase assay system reagent (E1910, Promega) on a Luminoskan
Ascent. Serum total cholesterol levels were determined according to the manufacturer's instructions
(294-65801, Wako).

**Cell luciferase assay.** To analyze MB07811 is an agonist with selectivity for TR β as compared to
TR α , pal-luc reporter plasmid was co-transfected with TR α or TR β expressing plasmid as indicated
in HepG2 cells followed by MB07811 treatment (100 μ M, 24h), the pRL-TK vector was used to
normalize the luciferase activity. Cells were lysed 48 h after transfection and measured luciferase
activity by using Dual-Luciferase® Reporter Assay System.

**Reference**

- 1. Chassande, O., *Do unliganded thyroid hormone receptors have physiological functions?*
*J Mol Endocrinol*, 2003. **31**(1): p. 9-20.
- 2. Xie, C., et al., *An Intestinal Farnesoid X Receptor-Ceramide Signaling Axis Modulates*
*Hepatic Gluconeogenesis in Mice*. *Diabetes*, 2017. **66**(3): p. 613-626.
- 3. Lund, M.L., et al., *L-Cell Differentiation Is Induced by Bile Acids Through GPBAR1 and*
*Paracrine GLP-1 and Serotonin Signaling*. *Diabetes*, 2020. **69**(4): p. 614-623.
- 4. Pathak, P., et al., *Farnesoid X receptor induces Takeda G-protein receptor 5 cross-talk to*
*regulate bile acid synthesis and hepatic metabolism*. *J Biol Chem*, 2017. **292**(26): p. 11055-
11069.
- 5. Pathak, P., et al., *Intestine farnesoid X receptor agonist and the gut microbiota activate G-*
*protein bile acid receptor-1 signaling to improve metabolism*. *Hepatology*, 2018. **68**(4): p.
1574-1588.
- 6. Fang, S., et al., *Intestinal FXR agonism promotes adipose tissue browning and reduces*
*obesity and insulin resistance*. *Nature Medicine*, 2015. **21**(2): p. 159-165.
- 7. Zheng, X.J., et al., *Hyocholic acid species improve glucose homeostasis through a distinct*
*TGR5 and FXR signaling mechanism*. *Cell Metabolism*, 2021. **33**(4): p. 791-+.
- 8. Sinal, C.J., et al., *Targeted disruption of the nuclear receptor FXR/BAR impairs bile acid*
*and lipid homeostasis*. *Cell*, 2000. **102**(6): p. 731-44.
- 9. Prawitt, J., et al., *Farnesoid X receptor deficiency improves glucose homeostasis in mouse*
*models of obesity*. *Diabetes*, 2011. **60**(7): p. 1861-71.
- 10. Zhang, Y., et al., *Activation of the nuclear receptor FXR improves hyperglycemia and*
*hyperlipidemia in diabetic mice*. *Proc Natl Acad Sci U S A*, 2006. **103**(4): p. 1006-11.
- 11. Watanabe, M., et al., *Lowering bile acid pool size with a synthetic farnesoid X receptor*
*(FXR) agonist induces obesity and diabetes through reduced energy expenditure*. *J Biol*
*Chem*, 2011. **286**(30): p. 26913-20.
- 12. Trabelsi, M.S., et al., *Farnesoid X receptor inhibits glucagon-like peptide-1 production by*
*enteroendocrine L cells*. *Nat Commun*, 2015. **6**: p. 7629.
- 13. Ma, K., et al., *Farnesoid X receptor is essential for normal glucose homeostasis*. *J Clin Invest*,

- 2006. **116**(4): p. 1102-9.
- 14. Ferrandino, G., et al., *Pathogenesis of hypothyroidism-induced NAFLD is driven by intra-*
*and extrahepatic mechanisms*. Proceedings of the National Academy of Sciences of the
United States of America, 2017. **114**(43): p. E9172-E9180.
- 15. Bruin, J.E., et al., *Hypothyroidism Impairs Human Stem Cell-Derived Pancreatic Progenitor*
*Cell Maturation in Mice*. Diabetes, 2016. **65**(5): p. 1297-1309.
- 16. Faddladdeen, K., et al., *Thymoquinone Preserves Pancreatic Islets Structure Through*
*Upregulation of Pancreatic beta-Catenin in Hypothyroid Rats*. Diabetes Metab Syndr
Obes, 2021. **14**: p. 2913-2924.
- 17. Song, Y., et al., *Thyroid-Stimulating Hormone Levels Are Inversely Associated with Serum*
*Total Bile Acid Levels: A Cross-Sectional Study*. Endocr Pract, 2016. **22**(4): p. 420-6.
- 18. Xu, C., et al., *Abnormal Glucose Metabolism and Insulin Resistance Are Induced via the*
*IRE1alpha/XBP-1 Pathway in Subclinical Hypothyroidism*. Front Endocrinol (Lausanne),
2019. **10**: p. 303.
- 19. Salazar, P., et al., *Induction of hypothyroidism during early postnatal stages triggers a*
*decrease in cognitive performance by decreasing hippocampal synaptic plasticity*.
Biochimica Et Biophysica Acta-Molecular Basis of Disease, 2017. **1863**(4): p. 870-883.
- 20. Biondi, B., G.J. Kahaly, and R.P. Robertson, *Thyroid Dysfunction and Diabetes Mellitus: Two*
*Closely Associated Disorders*. Endocrine Reviews, 2019. **40**(3): p. 789-824.
- 21. Erion, M.D., et al., *Targeting thyroid hormone receptor-beta agonists to the liver reduces*
*cholesterol and triglycerides and improves the therapeutic index*. Proc Natl Acad Sci U S
800 A, 2007. **104**(39): p. 15490-5.
- 22. Kaur, A., et al., *Loss of Cyp8b1 improves glucose homeostasis by increasing GLP-1*.
Diabetes, 2015. **64**(4): p. 1168-79.
- 23. Li, T., et al., *Glucose and insulin induction of bile acid synthesis: mechanisms and*
*implication in diabetes and obesity*. J Biol Chem, 2012. **287**(3): p. 1861-73.
- 24. Haeusler, R.A., et al., *Impaired generation of 12-hydroxylated bile acids links hepatic*
*insulin signaling with dyslipidemia*. Cell Metab, 2012. **15**(1): p. 65-74.
- 25. Astapova, I., et al., *The nuclear corepressor, NCoR, regulates thyroid hormone action in*
*vivo*. Proc Natl Acad Sci U S A, 2008. **105**(49): p. 19544-9.
- 26. Astapova, I., et al., *Hepatic nuclear corepressor 1 regulates cholesterol absorption through*
*a TRbeta1-governed pathway*. J Clin Invest, 2014. **124**(5): p. 1976-86.
- 27. Shabtai, Y., et al., *A coregulator shift, rather than the canonical switch, underlies thyroid*
*hormone action in the liver*. Genes Dev, 2021. **35**(5-6): p. 367-378.
- 28. Wahlström, A., et al., *Cyp3a11 is not essential for the formation of murine bile acids*.
Biochem Biophys Rep, 2017. **10**: p. 70-75.
- 29. Mohacsik, P., et al., *A Transgenic Mouse Model for Detection of Tissue-Specific Thyroid*
*Hormone Action*. Endocrinology, 2018. **159**(2): p. 1159-1171.

REVIEWER COMMENTS

Reviewer #1 (Remarks to the Author):

The manuscript has been significantly improved with this revised version. The authors have included new data from critical experiments requested by the reviewers. Issues related to comparability of effects across relevant groups has been partially mitigated by the inclusion of wild type controls treated with T3 in some of those newly added loss-of-function experiments. Those experiments are key to compellingly test the authors hypothesis and provide critical mechanistic insights. Additional plasma measurements also increase the overall confidence in the results. I have no further objections.

Reviewer #2 (Remarks to the Author):

The authors have done a very comprehensive job in addressing my questions and comments. The work is novel and very intriguing, which will provide novel insights into understanding the effects and underlying molecular mechanism of hypothyroidism on glucose homeostasis.

Reviewer #4 (Remarks to the Author):

Yan et al. have described an interesting manuscript about a novel T3-mediated pathway in which T3 transcriptionally down-regulates CYP8B1 in the liver to generate bile acids that are FXR antagonists and cause increased GLP1 synthesis in the small bowel to cause increased insulin synthesis. This is an interesting pathway which highlights potential inter-organ communication and multi-hormone regulation to control insulin synthesis. It raises the possibility that blocking CYP8B1 could be a novel drug target to increase insulin secretion.

However, my major concern is an important one and is similar to Reviewer 1's concern. In Yan's model system, the mice are rendered hypothyroid with MMI. They are then injected with T3 daily for 5 days. However, since T3 has a half-life of 2 hours in mice, the mice are transiently hyperthyroid and then become hypothyroid again over the following 24 hours before their next T3 injection. Given this fluctuation, it is hard to know the precise thyroid state of the T3-injected mice. Thus, it would be useful to see whether this pathway plays a significant role in the euthyroid and hyperthyroid states where serum ft3 levels would be relatively constant. Performing such studies also would provide more clinical relevance to their studies. In Suppl. Fig. 1, the authors compare MMI-treated and euthyroid mice, and found decreased insulin and reduced GLP1 in MMI-treated mice compared to euthyroid mice. What happens to the bile acid composition in the euthyroid mice? Does AAV sh8B or FXR KO affect GLP1 and insulin secretion in euthyroid mice? The same types of questions could be asked for hyperthyroid mice. Note that the hyperthyroid mice would need to be treated with both T4 and T3 since T4 has a much longer half-life than T3.. As such, Yan's system describes chronic daily T3 injections over 5 days that repetitively stimulate or decrease target genes daily (reminiscent of a study by Ohba et al. on T3-treated

euthyroid mice. *Endocrinology* 157:1660–1672(2016)). It is interesting that one T3 injection did not generate effects on GLP1 and insulin secretion. Does MMI+T3-4h have any effects in CYP8B1 expression in comparison to 5d shown in Fig. 4d? The study by Ohba showed that acute vs. chronic stimulation by T3 has different effects on target genes.

Fig. 4. I would like to see transcriptome analysis of euthyroid vs MMI+T3-5d mice. Comparing hyperthyroid mouse transcriptome with MMI+T3-5d also would be useful. It would be useful to look at GLP1, insulin, GTT, and bile acid composition for euthyroid and hyperthyroid mice with MMI and MMI+T3-5d mice.

It would be interesting to examine insulin response in the liver in their studies. T3 typically stimulates gluconeogenesis in the liver, so does activation of this pathway by insulin change the direction towards glycolysis or are the livers of the hypothyroid mice still insulin resistant? The authors should look at IR phosphorylation, pAkt, pmTOR signaling for evidence of insulin signaling. They also could look at expression of T3 target genes, PEPCK, G6Pase, PDK4 mRNA, in the liver. It would be important to know whether this elaborate regulatory pathway is increasing insulin effects in the liver or is stimulating insulin secretion to serve other tissues.

Fig. 6. Does T3 have separate effects on intestinal cells? Could it regulate FXR levels, or GLP1 gene expression independently of FXR? Such studies could be done in cell lines or organoids described in Fig. 6e.

**REVIEWER COMMENTS**

**Reviewer #1 (Remarks to the Author):**

The manuscript has been significantly improved with this revised version. The authors have
included new data from critical experiments requested by the reviewers. Issues related to
comparability of effects across relevant groups has been partially mitigated by the inclusion of wild
type controls treated with T3 in some of those newly added loss-of-function experiments. Those
experiments are key to compellingly test the authors hypothesis and provide critical mechanistic
insights. Additional plasma measurements also increase the overall confidence in the results. I have
no further objections.

**Response to the Reviewer's Comments:** We thank the reviewer for the valuable and insightful
comments and suggestions, which have improved our manuscript substantially.

**Reviewer #2 (Remarks to the Author):**

The authors have done a very comprehensive job in addressing my questions and comments. The
work is novel and very intriguing, which will provide novel insights into understanding the effects
and underlying molecular mechanism of hypothyroidism on glucose homeostasis.

**Response to the Reviewer's Comments:** We thank the reviewer for the detailed review and the
constructive suggestions that have been helpful to improve our manuscript.

**Reviewer #4 (Remarks to the Author):**

Yan et al. have described an interesting manuscript about a novel T3-mediated pathway in which
T3 transcriptionally down-regulates CYP8B1 in the liver to generate bile acids that are FXR
antagonists and cause increased GLP1 synthesis in the small bowel to cause increased insulin
synthesis. This is an interesting pathway which highlights potential inter-organ communication and
multi-hormone regulation to control insulin synthesis. It raises the possibility that blocking CYP8B1
could be a novel drug target to increase insulin secretion.

However, my major concern is an important one and is similar to Reviewer 1's concern. In Yan's
model system, the mice are rendered hypothyroid with MMI. They are then injected with T3 daily
for 5 days. However, since T3 has a half-life of 2 hours in mice, the mice are transiently hyperthyroid
and then become hypothyroid again over the following 24 hours before their next T3 injection.
Given this fluctuation, it is hard to know the precise thyroid state of the T3-injected mice. Thus, it
would be useful to see whether this pathway plays a significant role in the euthyroid and
hyperthyroid states where serum fT3 levels would be relatively constant. Performing such studies
also would provide more clinical relevance to their studies.

**Response to the Reviewer's Comments:** We thank the reviewer for the comments. We agree with
the reviewer that, normally, the MMI or PTU-treated mice receiving T3 daily injections for several
35 days are not considered as hyperthyroid mice [1]. Although the T3 levels are repetitively elevated
by daily T3 injections, mice are thought to be transiently hyperthyroid, which are not suitable for
modeling clinical hyperthyroidism. However, given that, after 24 hours of the last T3 injection, the
T3 normally would return to baseline levels similar to those in euthyroid mice (Supporting Fig. 1a

and TABLE 1 in a published paper by Zavacki et al. [1]), we speculate that the mice may not become
hypothyroid again over the following 24 hours before the next T3 injection.

Importantly, we'd like to point out that this canonical model is normally employed to identify genes
with maximal T3 responsiveness. In this study, instead of exploring T3-regulated genes, we used
this mouse model, aiming to discover novel metabolic pathways not only responding to the T3
treatment but also mediating the beneficial effects of T3. It is well accepted that the overall effects
of thyroid hormone (TH) on glucose metabolism are complex, some of its effects are mediated via
its action on multiple metabolic tissues, including liver, skeletal muscle, and adipose tissues, and
some of its actions on different tissues may counteract [2]. Given that glucose intolerance is
sometimes observed in thyrotoxicosis and hyperthyroidism normally worsens glycemic control, we
think a hyperthyroid mouse model may be not suitable for this study that aims to explore the
beneficial effect of T3 on glucose metabolism. Thus, we neither use the word "hyperthyroid" to
describe the state of T3-treated MMI mice nor attempt to use this canonical "replenishment" model
(MMI plus T3 treatment) to understand the dysregulated glucose metabolism in patients with
clinical hyperthyroidism.

Indeed, based on our newly obtained data from a hyperthyroid mouse model (euthyroid mice treated
with both T4 and T3, also referred to as TH-5d mice), we noticed that increased GLP-1 and insulin
levels and reduced glucose levels could not be observed (Supporting Fig. 1c), although the elevated
T3 and T4 levels in hyperthyroid mice (TH-5d mice) (Supporting Fig. 1b) could result in decreases
in the hepatic CYP8B1 expression and the ratios of 12 α -OH to non-12 α -OH BAs and increases in
the ileum FXR-antagonistic BA levels and proglucagon mRNA expression (Supporting Fig. 1d-h).
Oral glucose tolerance was not significantly impaired, although the abnormal fasting glucose levels
and higher glucose levels 15 minutes after oral glucose ingestion were noticed in hyperthyroid mice
(TH-5d mice) (Supporting Fig. 1i). These results suggest that the sustained elevated or the
pathological levels of T3 in these hyperthyroid mice may have deleterious effects. As
hyperthyroidism can cause protein catabolism and increase renal blood flow and glomerular
filtration rate [3], while kidney is thought to be a major site of GLP-1 extraction [4], we speculate
that increased glomerular filtration in the hyperthyroid state may promote the renal catabolism of
GLP-1, counteracting the effect of T3 on GLP-1 production. However, we cannot rule out the
possibility that there may be more complex mechanisms involved. Due to the technical issues, the
renal clearance of GLP-1 is difficult to study for the time being, although it deserves more attention.
Anyway, we agree with the reviewer that the regulation of GLP-1 by TH in the hyperthyroid state
requires further study, which will provide insights into the adverse events associated with clinical
hyperthyroidism. We provided some discussion and cited related papers regarding these issues
mentioned above in our revised manuscript.

In Suppl. Fig. 1, the authors compare MMI-treated and euthyroid mice, and found decreased insulin
and reduced GLP1 in MMI-treated mice compared to euthyroid mice. What happens to the bile acid
composition in the euthyroid mice? Does AAV sh8B or FXR KO affect GLP1 and insulin secretion
in euthyroid mice? The same types of questions could be asked for hyperthyroid mice.

**Response to the Reviewer's Comments:** We thank the reviewer for the comments. Probably due
to the limitation on the number of figures and the length of manuscript, we provided some data as
supplementary figures. For example, we provided the data of BA levels and composition for

euthyroid mice in our manuscript submitted previously (Supplementary Fig. 4l (4i in previous
version) and Supplementary Fig. 5b-e). We found that the decreased insulin and reduced GLP-1
levels (Supplementary Fig. 1m,n) were accompanied with increased ratios of 12 α -OH to non-12 α -
OH BAs and reduced FXR-antagonistic BA levels in MMI mice compared to euthyroid control
mice (Supplementary Fig. 4l (4i in previous version) and Supplementary Fig. 5b-e).

As the reviewer suggested, we determined the effect of AAV-sh8B on GLP-1 and insulin in
euthyroid and hyperthyroid mice. We found that AAV-sh8B infection could increase GLP-1 and
insulin levels and decrease glucose levels in euthyroid mice (Supplementary Fig. 4i), which are
similar to those observed in MMI-treated mice (Fig. 4j,k), suggesting that hepatic CYP8B1
knockdown has beneficial effects in either euthyroid or hypothyroid mice. In contrast, in
hyperthyroid mice (TH-5d mice), AAV-sh8B infection was unable to alter the levels of GLP-1,
insulin, and glucose, although it could elevate the ileum GLP-1 expression (Supporting Fig. 2a).
Notably, T4 and T3 treatment for 5 days decreased the GLP-1 and insulin levels and increased
glucose levels in mice with hepatic CYP8B1 knockdown, further supporting our notion that the
sustained elevated or pathological levels of T3 have deleterious effects (Supporting Fig. 2a), which
is likely independent of the regulation of hepatic CYP8B1 by T3.

Regarding the effect of intestine-specific FXR knockdown on GLP-1 and insulin in a euthyroid state,
we included these data in our manuscript submitted previously (Supplementary Fig. 7c). We found
that the T3 effects on the GLP-1, insulin, and glucose levels were all attenuated in euthyroid
IFXRKO mice (Supplementary Fig. 7c), which are similar to those observed in MMI-treated mice
(Fig. 7a-c). These results indicate that intestinal FXR is indispensable for the beneficial effect of T3
on GLP-1 and insulin production and glucose homeostasis. As we described in detail in our previous
rebuttal letter (Page5, Line 197-200), consistent with our current knowledge of intestinal FXR
signalling and our working hypothesis, the GLP-1 production was increased, while the glucose
metabolism was improved in MMI-treated IFXRKO mice as compared to MMI-treated Floxed mice
(Fig. 7a-c). On the other hand, in line with the previous findings that intestinal FXR is not required
for maintaining the normal glucose homeostasis [5], the glucose metabolism was normal in
IFXRKO mice in the euthyroid state, suggesting that compensatory mechanisms may exist
(Supplementary Fig. 7c,d).

As the reviewer suggested, we tested the effect of intestinal FXR deficiency in the hyperthyroid
state. Again, we neither observed the T3 on the serum levels of GLP-1, insulin, and glucose in
euthyroid Floxed control mice (Supporting Fig. 2b), consistent with the data from wild-type
C57BL/6J with (Supporting Fig. 2a) or without (Supporting Fig. 1c) AAV-CT infection, nor detected
the effect of deletion of intestinal FXR in the euthyroid group as we observed previously
(Supplementary Fig. 7c). As expected, we did not observe the effect of intestinal FXR deficiency in
mice treated with both T3 and T4 (TH-5d mice) (Supporting Fig. 2b).

Note that the hyperthyroid mice would need to be treated with both T4 and T3 since T4 has a much
longer half-life than T3. As such, Yan's system describes chronic daily T3 injections over 5 days
that repetitively stimulate or decrease target genes daily (reminiscent of a study by Ohba et al. on
T3-treated euthyroid mice. *Endocrinology* 157:1660–1672(2016)). It is interesting that one T3
injection did not generate effects on GLP1 and insulin secretion. Does MMI+T3-4h have any effects

in CYP8B1 expression in comparison to 5d shown in Fig. 4d? The study by Ohba showed that acute
vs. chronic stimulation by T3 has different effects on target genes.

**Response to the Reviewer's Comments:** We agree with the reviewer that mice need to be treated
with both T4 and T3 to induce hyperthyroidism. As mentioned above, as requested, we employed
this hyperthyroid mouse model (euthyroid mice treated with both T4 and T3, referred to as TH-5d
mice) to test whether the T3 effects observed in MMI mice or euthyroid mice could be retained in
the hyperthyroid state. We found that the elevated T3 and T4 levels could downregulate hepatic
CYP8B1, decrease the ratios of 12 α -OH BAs to non-12 α -OH BAs, and increase the ileum FXR-
antagonistic BA levels and proglucagon mRNA expression, but failed to elevate the GLP-1 and
insulin levels and lower glucose levels in TH-5d mice (Supporting Fig. 1b-g), suggesting that the
sustained elevated or pathological levels of T3 might have deleterious effects. According to our
proposed model, the beneficial effect of T3 requires the downregulation of hepatic CYP8B1 and the
alteration of BA composition to affect GLP-1 and insulin. We then speculate that it might take times
to change the BA composition, therefore, the beneficial effect of T3 could not be observed 4 hours
after one T3 injection (Supplementary Fig. 1b-d). In agreement with our hypothesis, we found that
the BA composition (12 α -OH and non-12 α -OH BAs and FXR-antagonistic BAs) were not altered
4 hours after one T3 injection (Supplementary Fig. 4m and 5f,g).

As suggested, we determined the expression of CYP8B1 four hours after one T3 injection. In
contrast to MMI+T3-5d group, the mRNA levels of CYP8B1 were not reduced four hours after one
T3 injection (Supplementary Fig. 4d). Importantly, the protein levels of CYP8B1 were not altered
four hours after one T3 injection (Supplementary Fig. 4d). These results indicate that it might also
take times to change the expression of CYP8B1 after T3 treatment. We then determined the
CYP8B1 expression at different time points after single T3 injection (acute T3) (4 hours or 1 day
after T3 injection) or daily T3 injections for 2 or 5 days (chronic T3) in euthyroid mice and found a
trend of gradual decrease in CYP8B1 after T3 treatment (Supporting Fig. 3a). As the expression
changes of CYP8B1 after acute and chronic T3 treatment did not follow the “desensitization”
pattern observed by Ohba et al.[6], we speculate that the T3 responsiveness of CYP8B1 may have
its own regulatory mechanism.

Our knowledge of the transcriptional regulation mediated by T3 and its receptor TR has been greatly
expanded recently by taking advantage of the genome wide ChIP-seq analysis [7-11]. Growing
evidence suggests that the dynamic TR binding and chromatin remodeling may be critically
involved in the transcriptional regulation by T3. To better understand the regulation of CYP8B1 by
T3, we performed ChIP analysis to investigate the TR occupancy and chromatin state after single
T3 injection (acute T3) or daily T3 injections for 2 or 5 days (chronic T3) in euthyroid mice in the
super-enhancer region we identified based on the latest data from Lazar’s lab [10], using a mouse
model newly developed in our lab with an epitope tag located at the C-terminal of TR β protein (HA-
TR β mouse), which allows us to explore the recruitment of endogenous TR β at the low
physiological levels (Supporting Fig. 3b).

In agreement with the recent hypothesis based on the genome wide ChIP-seq data, we also observed
that T3 treatment could affect the TR recruitment to chromatin, as evident by decreased TR β
occupancy around two putative TR binding sites (TRBS) (DR1 and DR4) (Supporting Fig. 3c,d).
Moreover, consistent with the notion that H3K27ac, a hallmark for active enhancers, would decrease

in a T3-dependent manner at TRBSs near down-regulated genes, we found the occupancy of
H3K27ac at the DR1 and DR4 sites was also decreased after T3 administration, suggesting a
decrease in enhancer activity for CYP8B1 after T3 treatment (Supporting Fig. 3c,d).

Furthermore, for the DR1 site, H3K27ac was decreased 4 hours after T3 injection and maintained
at a low level for at least 1 day after acute T3 treatment and during chronic T3 treatment. Decreased
TR β occupancy was only observed after chronic T3 treatment (Supporting Fig. 3c). For the DR4
site, H3K27ac was decreased until 1 day after T3 injection and maintained at a low level during
chronic T3 treatment. TR β occupancy was decreased as early as 4 hours after acute T3 treatment
and maintained at a low level for at least 1 day after acute T3 treatment and during chronic T3
treatment (Supporting Fig. 3d). As the dynamics of TR recruitment and H3K27ac occupancy at DR1
and DR4 seem to be differentially regulated by T3, we speculate that the TR binding and chromatin
remodeling at DR1 and DR4 sites may be independently regulated after T3 treatment. Although a
decrease in TR binding was always accompanied with a reduction of H3K27ac during chronic T3
treatment, we speculate that decreased TR binding may not be required for the reduction of
H3K27ac and vice versa. As the decrease in H3K27ac occurred prior to the decline in TR occupancy
for the DR1 site, while the reduction of H3K27ac occurred after the decline in TR binding for the
DR4 site, we speculate that binding site-dependent mechanisms may also exist. As the chromatin
remodeling is highly dynamic and other chromatin modifications are also involved, the epigenetic
landscape of the enhancer and promoter of CYP8B1 requires further investigation in different
thyroid states and at more time points after T3 treatment in the future.

We provided some background information and cited important papers mentioned above (including
the paper by Ohba et al.) in our revised manuscript, for a better understanding the current knowledge
of the transcriptional regulation mediated by T3 and its receptor TR, especially for people not in TR
field.

Fig. 4. I would like to see transcriptome analysis of euthyroid vs MMI+T3-5d mice. Comparing
hyperthyroid mouse transcriptome with MMI+T3-5d also would be useful. It would be useful to
look at GLP1, insulin, GTT, and bile acid composition for euthyroid and hyperthyroid mice with
MMI and MMI+T3-5d mice.

**Response to the Reviewer's Comments:** As suggested, we first compared the 4428 differential
expressed genes (DEGs) identified in MMI+T3-5d mice as compared to MMI mice and 2763 DEGs
identified in euthyroid (CT) mice as compared to MMI mice and found 1412 overlapped DEGs
(Supporting Fig. 3e) (RNA-seq data in this study, GSE184261), suggesting that the DEGs in
euthyroid (CT) and MMI-T3-5d mice as compared to MMI mice exhibit a significant degree of
overlap. The heatmap of the overlapped DEGs showed that these DEGs might be classic T3-
regulated genes, although the responsiveness or sensitivity of individual DEGs to MMI or T3
treatment was different (Supporting Fig. 3f).

Also as suggested, we compared the DEGs identified in MMI+T3-5d mice as compared to MMI
mice (RNA-seq data, GSE184261) and DEGs identified in hyperthyroid (Chronic T3) mice as
compared to euthyroid (NT, no treatment) mice (microarray data from Ohba et al.). We found that
the DEGs identified from these two datasets exhibit a significant degree of overlap (Supporting Fig.
3g), suggesting that quite a few of T3-regulated genes can be identified from both experimental
models, including CYP8B1. Interestingly, a higher degree of overlap was observed for positively-

regulated genes as compared to negatively-regulated genes (Supporting Fig. 3g). Furthermore,
KEGG pathway analysis of DEGs identified from these two datasets also revealed a large overlap,
further suggesting that the T3 effects on quite a lot of pathways can be observed in both models
(Supporting Fig. 3h).

Although there is considerable overlapping, alteration of some genes was exclusively detectable in
one model but not in the other model, suggesting that these two experimental models are not exactly
the same. Nevertheless, the negatively regulation of CYP8B1 after chronic T3 treatment could be
observed in either MMI-induced hypothyroid mice or euthyroid mice with no treatment.

As suggested, we determined the GLP1, insulin, and BA composition in hyperthyroid mice (TH-5d
mice) as compared to euthyroid mice (CT mice). We have described these results above in response
to the Reviewer's Comments (Page 2, Line 54-73). Briefly, increased GLP-1 levels and reduced
glucose levels could not be observed, although the elevated T3 and T4 levels could also decrease
the hepatic CYP8B1 expression and the ratios of 12 α -OH to non-12 α -OH BAs, and increase the
ileum FXR-antagonistic BA levels and proglucagon mRNA expression (Supporting Fig. 1b-h). Oral
glucose tolerance was not significantly impaired, although the abnormal fasting glucose levels and
higher glucose levels 15 minutes after oral glucose ingestion were noticed in hyperthyroid mice
(TH-5d) (Supporting Fig. 1i). Given that hyperthyroidism may increase renal blood flow and
glomerular filtration [3] and the kidney is a site for GLP-1 extraction [4], we speculate that increased
glomerular filtration in the hyperthyroid state may counteract the effect of T3 on GLP-1 production
by promoting the renal clearance of GLP-1. Due to the technical issues, the clearance of GLP-1 is
difficult to study for the time being, although it deserves more attention and requires further study
in the future. We thus provided some discussion and cites related papers in our revised manuscript.

It would be interesting to examine insulin response in the liver in their studies. T3 typically
stimulates gluconeogenesis in the liver, so does activation of this pathway by insulin change the
direction towards glycolysis or are the livers of the hypothyroid mice still insulin resistant? The
authors should look at IR phosphorylation, pAkt, pmTOR signaling for evidence of insulin signaling.
The also could look at expression of T3 target genes, PEPCK, G6Pase, PDK4 mRNA, in the liver.
It would be important to know whether this elaborate regulatory pathway is increasing insulin
effects in the liver or is stimulating insulin secretion to serve other tissues.

**Response to the Reviewer's Comments:** We thank the reviewer for the suggestion. As requested,
we examined the insulin signalling and the expression of insulin or T3-related genes in our mouse
models. We found that the hepatic insulin signalling was decreased in hypothyroid mice (MMI mice)
as compared to euthyroid mice (CT mice) (Supporting Fig. 4a). As acute T3 treatment (4 hours after
T3 injection) did not affect the insulin signalling in the liver of MMI mice, suggesting that the
hepatic insulin signalling remained impaired (Supporting Fig. 4a). In contrast, daily T3 injections
for 5 days increased the insulin signalling in the liver of MMI mice, suggesting that chronic T3
treatment could restore the hepatic insulin signalling (Supporting Fig. 4a). Notably, although
abnormal insulin levels were not observed in hyperthyroid mice (TH-5d mice) as compared to
euthyroid mice (CT mice) (Supporting Fig. 1c), the hepatic insulin signalling was impaired in these
hyperthyroid mice (TH-5d mice) (Supporting Fig. 4a). Collectively, we speculate that both
hypothyroid and hyperthyroid mice had impaired hepatic insulin signalling and only chronic but not

acute T3 treatment could restore the hepatic insulin action in hypothyroid mice (MMI mice) with
deficiency in TH production.

Gluconeogenic PEPCK and G6Pase are responsible for glucose anabolism, while PDK4 is a
negatively regulator for glucose catabolism, the overall intracellular effect of three genes is to
promote the glucose production and reduce the glucose usage. PEPCK, G6Pase, PDK4 are
considered as T3 target genes, which are also negatively regulated by insulin signalling. As
suggested, we examined the mRNA expression of PEPCK, G6Pase, PDK4. In agreement with the
notion that they are T3 target genes, we found that they were all downregulated in the liver of
hypothyroid mice (MMI mice) as compared to euthyroid mice (CT mice) (Supporting Fig. 4b),
while they were all upregulated 4 hours after T3 injection in the liver of MMI mice (Supporting Fig.
4c). In contrast, the mRNA expression of these T3 target genes tended to be downregulated rather
than upregulated after daily T3 injections for 5 days in the liver of MMI mice (Supporting Fig. 4d).
These results suggest that acute T3 treatment can upregulate PEPCK, G6Pase, PDK4 genes, thereby
promoting gluconeogenesis and reduce glucose usage, while the expression of PEPCK, G6Pase,
PDK4 would return to baseline expression after chronic T3 treatment.

Based on our finding that chronic but not acute T3 treatment could elevate insulin levels by
promoting GLP-1 production in this study (Fig. 1c,f) and the increases in hepatic insulin signalling
could be only observed after chronic but not acute T3 treatment (Supporting Fig. 4a), we speculate
that insulin action might not be involved in the regulation of PEPCK, G6Pase, and PDK4 after acute
T3 treatment, because the hepatic insulin signalling and insulin levels in MMI mice were not altered,
while the chronic T3 treatment-induced and GLP-1-mediated increase in insulin production (as
evident by the elevated insulin levels and the increased phosphorylation of key components of
insulin signalling pathway) could counteract the effect of T3 on the expression of PEPCK, G6Pase,
and PDK4. Therefore, rather than being elevated as observed after acute T3 treatment, the mRNA
levels of PEPCK, G6Pase, and PDK4 returned to baseline expression levels in MMI mice after
chronic T3 treatment. Collectively, our study might also provide new insight into the possible
involvement of insulin action in the regulation of hepatic glucose metabolism by chronic T3
treatment.

Additionally, further analysis of our RNA-seq data (GSE184261) revealed that, in MMI mice, acute
T3 treatment might not affect glycolysis, in contrast, chronic T3 treatment might promote glycolysis,
indicating that glycolysis might be tightly controlled by insulin but not by T3 and the regulation of
glycolysis by chronic T3 treatment might be attributed to the elevation of insulin levels. As the
insulin can lower blood glucose levels through multiple metabolic tissues via multiple mechanisms,
based on all available data, we speculate that insulin action in both liver and other tissues would
contribute to the glucose-lowering effect of insulin after chronic T3 treatment, which requires
further investigation in the future.

Fig. 6. Does T3 have separate effects on intestinal cells? Could it regulate FXR levels, or GLP1
gene expression independently of FXR? Such studies could be done in cell lines or organoids
described in Fig. 6e.

**Response to the Reviewer's Comments:** We thank the reviewer for the comments. We had indeed
explored the effect of T3 on GLP-1 production *in vitro* (Supplementary Fig. 2a). As we neither
observed the effect of T3 on GLP-1 expression and secretion or FXR expression in STC-1 cells,

NCI-H716 cells, and mouse intestinal organoids (Supplementary Fig. 2a), nor detected the T3 effect
on TGR5 mRNA levels in STC-1 cells, NCI-H716 cells, and mouse intestinal organoids (not
included in our previous version, Supporting Fig. 5), we did not further explore the potential direct
effect of T3 on these cells or organoids. As the reviewer suggested, we examined the FXR levels in
these cell lines and organoids and found that the mRNA expression of FXR was not altered by T3
treatment, which agrees with our proposed model that T3 regulates GLP-1 production via a non-cell
autonomous mechanism involving hepatic TR and the alteration of BA composition.

**Attached please find the references, supporting figures and legends, and materials and**
**methods associated with the Supporting data mentioned in this response letter.**

**Reference**

- 1. Zavacki, A.M., et al., *Type 1 iodothyronine deiodinase is a sensitive marker of peripheral*
*thyroid status in the mouse*. Endocrinology, 2005. **146**(3): p. 1568-75.
- 2. Sinha, R.A., B.K. Singh, and P.M. Yen, *Thyroid hormone regulation of hepatic lipid and*
*carbohydrate metabolism*. Trends Endocrinol Metab, 2014. **25**(10): p. 538-45.
- 3. Iglesias, P., et al., *Thyroid dysfunction and kidney disease: An update*. Rev Endocr Metab
Disord, 2017. **18**(1): p. 131-144.
- 4. Deacon, C.F., *Circulation and degradation of GIP and GLP-1*. Horm Metab Res, 2004.
**36**(11-12): p. 761-5.
- 5. Xie, C., et al., *An Intestinal Farnesoid X Receptor-Ceramide Signaling Axis Modulates*
*Hepatic Gluconeogenesis in Mice*. Diabetes, 2017. **66**(3): p. 613-626.
- 6. Ohba, K., et al., *Desensitization and Incomplete Recovery of Hepatic Target Genes After*
*Chronic Thyroid Hormone Treatment and Withdrawal in Male Adult Mice*.
Endocrinology, 2016. **157**(4): p. 1660-1672.
- 7. Chatonnet, F., et al., *Genome-wide analysis of thyroid hormone receptors shared and*
*specific functions in neural cells*. Proc Natl Acad Sci U S A, 2013. **110**(8): p. E766-75.
- 8. Grontved, L., et al., *Transcriptional activation by the thyroid hormone receptor through*
*ligand-dependent receptor recruitment and chromatin remodelling*. Nat Commun,
2015. **6**: p. 7048.
- 9. Praestholm, S.M., et al., *Multiple mechanisms regulate H3 acetylation of enhancers in*
*response to thyroid hormone*. PLoS Genet, 2020. **16**(5): p. e1008770.
- 10. Shabtai, Y., et al., *A coregulator shift, rather than the canonical switch, underlies thyroid*
*hormone action in the liver*. Genes Dev, 2021. **35**(5-6): p. 367-378.
- 11. Ramadoss, P., et al., *Novel mechanism of positive versus negative regulation by thyroid*
*hormone receptor beta1 (TRbeta1) identified by genome-wide profiling of binding sites*
*in mouse liver*. J Biol Chem, 2014. **289**(3): p. 1313-28.

**Supporting Figures and legends**

**Supporting Figure 1. CYP8B1, BA, GLP-1, insulin, and glucose levels in hyperthyroid mice.**

(a) The serum total T3 (left) and T4 (right) levels in CT, MMI, MMI+T3-4h and MMI+T3-5d mice
 (n=5) (right). (b) The serum total T3 (left) and T4 (right) levels in CT and TH-5d mice (n=5). (c)
 Plasma active GLP-1 levels, plasma insulin levels and blood glucose levels in CT and TH-5d mice
 (n=5). (d) Relative mRNA (left) and protein (right) levels of hepatic CYP8B1 in CT and TH-5d
 mice. (e) The percentage of individual BA in the ileum of CT and TH-5d mice (n=5). (f) Relative
 levels of 12 α -OH (blue) and non-12 α -OH (red) BAs in the ileum of CT and TH-5d mice (n=5). (g)
 The percentage of non-12 α -OH FXR-antagonistic BAs, including T(α / β)MCA, (T/G)UDCA and
 (T)HDCA in the ileum of CT and TH-5d mice (n=5). (h) The relative mRNA levels of proglucagon
 in CT and TH-5d mice (n=5). (i) oGTT for CT and TH-5d mice (left) and the AUC (right) (n=5).
 CT mice (euthyroid) are untreated control mice, while TH-5d mice (hyperthyroid) are euthyroid
 mice treated with both T3 and T4 as described in Materials and Methods for Supporting data. Means
 \pm SEM are shown. * p <0.05, ** p <0.01 and *** p <0.001.

**Supporting Figure 2. The effects of hepatic CYP8B1 knockdown or intestine-specific FXR**
 **knockout in euthyroid and hyperthyroid mice.**

(a) Plasma active GLP-1 levels, plasma insulin levels, blood glucose levels and relative mRNA
 levels of proglucagon in CT and TH-5d mice administered with AAV-CT or AAV-shCYP8B1 (AAV-
 sh8B) (n=5). (b) Plasma active GLP-1, plasma insulin and blood glucose levels in Floxed and
 IFXRKO mice treated with PBS or TH for 5 days (n=3). Mice were treated with both T3 and T4
 (TH) or PBS as described in Materials and Methods for Supporting data. Means ± SEM are shown.
 *p<0.05 and **p<0.01.

**Supporting Figure 3. The occupancy TRβ and H3K27ac in super-enhancer region of CYP8B1**
 **and the comparison between changes in transcriptome in different experimental models.**

(a) Relative mRNA levels of hepatic CYP8B1 in mice treated with T3 (0.25 μg/g; 0h (CT), 4h, 1d,
 2d, 5d) (n=3). (b) Schematic diagram of generating HA-TRβ mouse line that harbors a HA-tag in
 the C-terminal of *Thrβ* gene. (c and d) ChIP-PCR showing enrichment (percent input normalized to
 CT) of HA-TRβ (left) and H3K27 acetylation (right) at the putative TRβ binding sites (c, DR1) (d,
 DR4) of the super-enhancer region of CYP8B1 in the liver of HA-TRβ mice treated with T3 (0.25
 370 μg/g; 0h (CT), 4h, 1d, 2d, 5d) (n=3). (e) Venn diagrams showing overlapped DGEs between
 371 MMI+T3-5d vs MMI groups and MMI vs CT groups according to the RNA-seq data reported in
 this study. (f) Heatmaps of DEGs identified in the liver between CT, MMI and MMI+T3-5d groups
 (n=3) based on the RNA-seq data reported in this study. (g) Venn diagrams showing overlapped
 DGEs between MMI+T3-5d vs MMI groups (RNA-seq data reported in this study) and Chronic T3
 vs NT groups (microarray data reported by Ohba et al.). (h) Venn diagrams showing the overlapped
 KEGG pathways identified from the DEGs between MMI+T3-5d vs MMI groups (RNA-seq data
 reported in this study) and Chronic T3 vs NT groups (microarray data reported by Ohba et al.).
 The detailed information for the Chronic T3 treatment group and no treatment (NT) group in the study
 by Ohba et al. could also be found in the Materials and Methods for Supporting data. Means ± SEM
 are shown. *p<0.05, **p<0.01 and ***p<0.001.

**Supporting Figure 4. Hepatic insulin signalling and the mRNA expression of G6pase, PEPCK,**
 **and PDK4 in different experimental models.**

(a) Western blot analysis of key molecules of insulin signalling pathway in the liver of CT and MMI
 mice, MMI and MMI+T3-4h mice, MMI and MMI+T3-5d mice, CT and TH-5d mice. (b-d) Relative
 mRNA levels of G6Pase/PEPCK/PDK4 in the liver of CT and MMI mice, MMI and MMI+T3-4h
 mice, MMI and MMI+T3-5d mice (n=5). Means ± SEM are shown. *p<0.05, **p<0.01 and
 ***p<0.001.

**Supporting Figure 5. The effect of T3 treatment on TGR5 mRNA expression in cultured cells.**

Relative mRNA levels of TGR5 in STC-1 cells, NCI-H716 cells and mouse intestinal organoids
 after T3 treatment (n=3). Means ± SEM are shown.

**Materials and Methods for Supporting data**

**Mice study**

HA-TR β mice are mice harboring a HA-tag in the C-terminal of *Thrb* gene in C57BL/6J mice,
developed by the Genome Tagging Project (GTP) Center of Shanghai Institute of Biochemistry and
Cell Biology. Male C57BL/6J mice between ages 8 and 10 weeks were made hyperthyroid by
intraperitoneal injection of 40 μ g/100 g T4 with 4 μ g/100 g T3 for 5 days (TH-5d mice). Control
mice were injected with the same volume of PBS alone.

**Chromatin immunoprecipitation assay**

Male HA-TR β mice between ages 8 and 10 weeks received daily intraperitoneal injection of
T3 (0.25 μ g per gram BW) as indicated time in the figure. ~25mg of frozen liver material was used
404 per IP sample from 3–4 mice in each group. Livers were homogenized in PBS containing 1%
formaldehyde, incubated 10 min at room temperature and quenched with 0.125 M glycine. ChIP
assays were performed using an EZ Magna ChIP G kit (Millipore) according to the manufacturer's
protocol. 2 μ g/IP of antibody was used in HA (3724S, Cell Signaling Technology) and H3K27ac
(ab4729, Abcam) ChIP experiments.

**Analysis of mRNA and protein expression**

For qRT-PCR analysis, primers for G6Pase, PEPCK and PDK4 are show below: mG6Pase-F,
CTCTGGCCATGCCATG; mG6Pase-R, GCTGGCATTGTAGATGCC; m PEPCK -F, GAGAAAG
CATTCAACGCCA; m PEPCK -R, AGTTGTTGACCAAAGGCTTTTTTA; mPDK4-F, AGGGAG
GTCGAGCTGTTCTC; mPDK4-R, GGAGTGTTCACTAAGCGGTCA. For western blot analysis,
primary antibodies against p-Akt (9271S, Cell Signaling Technology), p-mTOR (2971S, Cell
Signaling Technology), p-IR (3024S, Cell Signaling Technology), p-FoxO1(9461T, Cell Signaling
Technology), mTOR (2983S, Cell Signaling Technology), FoxO1 (2880S, Cell Signaling
Technology), IR (23413T, Cell Signaling Technology), Akt (9272S, Cell Signaling Technology)
were used.

**Hormone measurement**

For the measurement of serum levels of total T3, a T3 (total) (Mouse/Rat) ELISA Kit (KA0925,
Abnova) was used. For the measurement of serum levels of total T4, a Thyroxine (T4) ELISA Kit
(MBS9711535, MYBiosource) was used.

**Data availability**

The liver RNA-seq data from CT, MMI and MMI+T3-5d groups generated in this study have
been deposited in the Gene Expression Omnibus database under accession code GSE184261.
Previously published liver microarray data from the Chronic T3 treatment group and no treatment
(NT) group is from Gene Expression Omnibus: GSE68867.

**Notes:**

The other materials and methods were described in the paper.

REVIEWERS' COMMENTS

Reviewer #4 (Remarks to the Author):

I am satisfied with responses to my comments. I appreciate the extra effort and experiments to address my concerns.

**REVIEWER COMMENTS**

**Reviewer #4 (Remarks to the Author):**

I am satisfied with responses to my comments. I appreciate the extra effort and experiments to
address my concerns.

**Response to the Reviewer's Comments:** We thank the reviewer for the detailed review for helping
6 us to improve our manuscript.